# Non-Convex Federated Optimization under Cost-Aware Client Selection

**Xiaowen Jiang**
Saarland University & CISPA*
`xiaowen.jiang@cispa.de`

**Anton Rodomanov**
CISPA*
`anton.rodomanov@cispa.de`

**Sebastian U. Stich**
CISPA*
`stich@cispa.de`

## ABSTRACT

Different federated optimization algorithms typically employ distinct client-selection strategies: some methods communicate only with a randomly sampled subset of clients at each round, while others need to periodically communicate with all clients or use a hybrid scheme that combines both strategies. However, existing metrics for comparing optimization methods typically do not distinguish between these strategies, which often incur different communication costs in practice. To address this disparity, we introduce a simple and natural model of federated optimization that quantifies communication and local computation complexities. This new model allows for several commonly used client-selection strategies and explicitly associates each with a distinct cost. Within this setting, we propose a new algorithm that achieves the best-known communication and local complexities among existing federated optimization methods for non-convex optimization. This algorithm is based on the inexact composite gradient method with a carefully constructed gradient estimator and a special procedure for solving the auxiliary subproblem at each iteration. The gradient estimator is based on SAGA, a popular variance-reduced gradient estimator. We first derive a new variance bound for it, showing that SAGA can exploit functional similarity. We then introduce the Recursive-Gradient technique as a general way to potentially improve the error bound of a given conditionally unbiased gradient estimator, including both SAGA and SVRG. By applying this technique to SAGA, we obtain a new estimator, RG-SAGA, which has an improved error bound compared to the original one.

## 1 INTRODUCTION

**Motivation.** Federated Learning (FL) is a distributed training paradigm in which a central server coordinates model updates across multiple remote clients—such as mobile devices or hospitals—without requiring access to their local data (McMahan et al., 2017; Kairouz et al., 2021). This framework enables collaborative learning on decentralized data, but introduces new algorithmic challenges due to the distributed nature of optimization.

A key issue in FL is the high cost of communication between the clients and the server. Clients may be intermittently available (Konečný et al., 2016) and connected over slow or unreliable networks. These constraints make it critical to design optimization algorithms that minimize communication costs, particularly in settings with partial client participation.

Various federated optimization algorithms have been proposed to address communication efficiency, each often relying on distinct client-selection strategies. Some methods communicate only with a randomly sampled subset of clients at each round, while others need to select the set of participating clients more carefully or employ hybrid schemes that combine both strategies. While prior works (Woodworth et al., 2018; Korhonen & Alistarh, 2021; Patel et al., 2022; Zhang et al., 2013; Davies et al., 2020; Scaman et al., 2019) introduced a few models for federated optimization, they do not account for the varying costs of each client-selection strategy, which can in practice differ due to factors such as client reliability, device heterogeneity, and network conditions. Consequently, existing metrics such as the number of communication rounds are not entirely fair for comparing methods in such scenario.

---

*CISPA Helmholtz Center for Information Security, Saarbrücken, Germany.

For instance, optimization methods based on SARAH (Nguyen et al., 2017; Li et al., 2021a) have been shown to be communication-efficient in finding an approximate stationary point (Mishchenko et al., 2024; Khaled & Jin, 2023). This efficiency arises from the method's ability to exploit dissimilarity ($\delta$) between local and global objectives. In many practical scenarios—such as statistical or semi-supervised learning (Chayti & Karimireddy, 2022; Karimireddy et al., 2021; Khaled & Jin, 2023)—$\delta$ is often small, leading to substantial theoretical gains in communication cost. However, SARAH-based methods require periodic full synchronization with all clients in order to compute full gradients. This can be impractical in real-world large-scale federated systems, where clients may be intermittently unavailable due to energy constraints, network issues, or user behavior.

In contrast to SARAH, methods such as SAG (Schmidt et al., 2017) and SAGA (Defazio et al., 2014) are naturally better suited to the partial participation setting in FL. These methods update the model by sampling a small subset of clients at each round and using locally stored gradients. As a result, they avoid the need for periodic full synchronization, which makes them more compatible with federated systems where only a fraction of clients may be available at any given time. Despite this advantage, the existing communication complexity of such methods depends on the individual smoothness constant $L_{\max}$ (Reddi et al., 2016; Li et al., 2021b; Karimireddy et al., 2020), which can be significantly larger than the dissimilarity constant $\delta$. Consequently, it remains unclear whether such methods are more communication-efficient than SARAH-based methods, since they rely on fundamentally different client-selection strategies with different constant dependencies.

**Contributions.** In this work, we aim to develop optimization algorithms that are efficient in both communication and local computation in the setting where client-selection strategies incur different costs. Our main contributions are as follows:

- We propose a new model formalizing the concept of federated optimization algorithms and defining information-based notions of communication and local complexities. This model associates the non-uniform costs with different client-selection strategies, enabling fair comparisons across optimization algorithms. (Section C)
- Within our new model, we propose a new gradient method that achieves the best communication and local complexities among existing first-order methods for non-convex optimization. This method is based on the inexact composite gradient method (I-CGM) with a carefully constructed gradient estimator and a special procedure for solving auxiliary subproblem at each iteration. (Section 6)
- Specifically, we first study the convergence of I-CGM for arbitrary gradient estimators and present an efficient technique for solving the auxiliary subproblem. Our technique is based on running the classical composite gradient method locally for a random number of iterations following a geometrical distribution with a carefully chosen parameter. (Section 3)
- We then analyze the SAGA estimator and establish a new variance bound for it that only depends on $\delta$ without requiring individual smoothness, improving upon previous results showing that SAGA can exploit functional similarity. We also study SVRG as another example that can be incorporated into I-CGM. (Section 4)
- Finally, we introduce the *Recursive-Gradient (RG) technique* as a general way to potentially improve the error bound for a given conditionally unbiased gradient estimator, including both SAGA and SVRG. Applying this technique to SAGA and SVRG, we obtain new RG-SAGA and RG-SVRG gradient estimators with better error bounds compared to the original ones. (Section 5)

We discuss our results in detail in the context of related work in Appendix D and summarize them in Table 1.

## 2 PROBLEM FORMULATION

We consider the following distributed minimization problem:

$$\min_{\mathbf{x} \in \mathbb{R}^d} \Big\{ f(\mathbf{x}) := \frac{1}{n} \sum_{i=1}^{n} f_i(\mathbf{x}) \Big\}, \tag{1}$$

where each $f_i \colon \mathbb{R}^d \to \mathbb{R}$ is a differentiable function which can be directly accessed only by client $i$.

**Notation.** We abbreviate $[n] := \{1, 2, \ldots, n\}$. For a finite set $A$ and an integer $1 \leq m \leq |A|$, $\binom{A}{m}$ denotes the power set comprised of all $m$-element subsets of $A$. $\|\cdot\|$ denotes the standard Euclidean

Table 1: Summary of efficiency guarantees (in BigO-notation) for finding an $\varepsilon$-stationary point. I-CGM-RG-SAGA achieves the best communication and local complexities. For the precise description of the problem classes, notations, as well as the discussions of the methods, see Appendix D.

| Method | Communication complexity | Assumption | Local complexity | VR type |
|---|---|---|---|---|
| Centralized GD | $C_A n_m \frac{L_f F^0}{\varepsilon^2}$ | FS | $n_m \frac{L_f F^0}{\varepsilon^2}$ | None |
| FedRed (Jiang et al., 2024a) | $C_A n_m \frac{\Delta_1 F^0}{\varepsilon^2}$ | 2.1 ; 2.3 | $\frac{L_1 F^0}{\varepsilon^2} + n_m \frac{\Delta_1 F^0}{\varepsilon^2}$ | None |
| FedAvg (McMahan et al., 2017) | $C_R\left(\frac{\zeta_m^2 F^0}{\varepsilon^4} + \frac{\sqrt{L_{\max}}\zeta}{\varepsilon^3} + \frac{L_{\max} F^0}{\varepsilon^2}\right)$ | IS , BGD | $\frac{\zeta_m^2 F^0}{\varepsilon^4} + \frac{\sqrt{L_{\max}}\zeta}{\varepsilon^3} + \frac{L_{\max} F^0}{\varepsilon^2}$ | None |
| FedDyn (Acar et al., 2021) | $C_A n_m + C_R n_m \frac{L_{\max} F^0}{\varepsilon^2}$ | IS | unknown | None |
| MimeMVR (Karimireddy et al., 2021) | $C_A\left(\frac{\zeta_m^2 F^0}{\varepsilon^2} + \frac{\zeta_m \Delta_{\max} F^0}{\varepsilon^3} + \frac{\Delta_{\max} F^0}{\varepsilon^2}\right)$ | IS, BGD, SD | $\frac{L_{\max}\zeta_m^2 F^0}{\Delta_{\max}\varepsilon^2} + \frac{\zeta_m L_{\max} F^0}{\varepsilon^3} + \frac{L_{\max} F^0}{\varepsilon^2}$ | None |
| CE-LGD (Patel et al., 2022) | $C_R\left(\frac{\zeta_m^2 F^0}{\varepsilon^2} + \frac{\zeta_m \Delta_{\max} F^0}{\sqrt{m}\varepsilon^3} + \frac{\Delta_{\max} F^0}{\varepsilon^2}\right)$ | IS, BGD, SD | $\frac{L_{\max}\zeta_m^2 F^0}{\Delta_{\max}\varepsilon^2} + \frac{\zeta_m L_{\max} F^0}{\sqrt{m}\varepsilon^3} + \frac{L_{\max} F^0}{\varepsilon^2}$ | None |
| Scaffold (Karimireddy et al., 2020) | $C_A n_m + C_A \frac{n_m^{2/3} L_{\max} F^0}{\varepsilon^2}$ | IS | $n_m + \frac{n_m^{2/3} L_{\max} F^0}{\varepsilon^2}$ | SAG |
| SABER-full (Mishchenko et al., 2024) | $C_A n_m + C_A \frac{(\Delta_{\max}+\sqrt{n_m}\delta_m) F^0}{\varepsilon^2}$ | SD | unknown | PAGE |
| SABER-partial (Mishchenko et al., 2024) | $C_A n_m + C_R \frac{\zeta_m^2 \Delta_{\max} F^0}{\varepsilon^2}$ | SD, BGD | unknown | SARAH |
| **I-CGM-RG-SVRG (ours)** | $C_A n_m + \frac{(C_R\Delta_1+\sqrt{C_A C_R n_m}\delta_m) F^0}{\varepsilon^2}$ | 2.1,2.2 ; 2.3 | $n_m + \frac{(L_1+\Delta_1+\sqrt{\frac{C_A}{C_R}n_m}\delta_m) F^0}{\varepsilon^2}$ | RG-SVRG |
| **I-CGM-RG-SAGA (ours)** | $C_A n_m + C_R \frac{(\Delta_1+\sqrt{n_m}\delta_m) F^0}{\varepsilon^2}$ | 2.1,2.2 ; 2.3 | $n_m + \frac{(L_1+\Delta_1+\sqrt{n_m}\delta_m) F^0}{\varepsilon^2}$ | RG-SAGA |

norm in $\mathbb{R}^d$. We use $\mathbb{E}[\cdot]$ to denote the standard (full) expectation. We write $\mathbb{E}_\xi[\cdot]$ for the expectation taken w.r.t. $\xi$. We assume that the objective function in problem (1) is bounded from below and denote its infimum by $f^\star$. We denote $F^0 := f(\mathbf{x}^0) - f^\star$ where $\mathbf{x}^0$ is the initial point.

## 2.1 FEDERATED OPTIMIZATION ALGORITHMS AND THEIR COMPLEXITY

Due to space limitations, we defer the whole definitions of federated optimizaton algorithms and their complexity metrics to Appendix C, which is encouraged to read before proceeding.

## 2.2 PROBLEM CLASS

We study optimization problem (1) in which the client objectives exhibit an underlying similarity structure. Specifically, we use the following two assumptions that relax standard smoothness assumptions. The first quantifies the deviation between the delegate function $f_1$ and $f$. For an index $i \in [n]$, we use $h_i := f - f_i$ to denote the difference function.

**Assumption 2.1.** There exists $\Delta_1 > 0$ such that for any $\mathbf{x}, \mathbf{y} \in \mathbb{R}^d$, we have:

$$\|\nabla h_1(\mathbf{x}) - \nabla h_1(\mathbf{y})\| \leq \Delta_1 \|\mathbf{x} - \mathbf{y}\|. \tag{2}$$

Alternatively, one may define a uniform dissimilarity constant $\Delta_{\max}$ (Karimireddy et al., 2020; Jiang et al., 2024a) such that for any $i \in [n]$, it holds that $\|\nabla h_i(\mathbf{x}) - \nabla h_i(\mathbf{y})\| \leq \Delta_{\max} \|\mathbf{x} - \mathbf{y}\|$. In this work, we focus on $\Delta_1$ since it can be much smaller than $\Delta_{\max}$.

The second assumption characterizes the average dissimilarity among all local functions.

**Assumption 2.2** (Khaled & Jin (2023); Jiang et al. (2024b); Lin et al. (2024); Jiang et al. (2024a); Takezawa et al. (2025)). There exists $\delta > 0$ such that for any $\mathbf{x}, \mathbf{y} \in \mathbb{R}^d$, we have:

$$\frac{1}{n}\sum_{i=1}^n \|\nabla h_i(\mathbf{x}) - \nabla h_i(\mathbf{y})\|^2 \leq \delta^2 \|\mathbf{x} - \mathbf{y}\|^2 . \tag{3}$$

The left-hand side of (3) is equal to $\frac{1}{n}\sum_{i=1}^n \|\nabla f_i(\mathbf{x}) - \nabla f_i(\mathbf{y})\|^2 - \|\nabla f(\mathbf{x}) - \nabla f(\mathbf{y})\|^2$, which can be interpreted as the variance of $\nabla f_i(x) - \nabla f_i(y)$ where $i$ is selected uniformly at random. If each $f_i$ has $L_{\max}$-Lipschitz gradient, then we have $\Delta_1 \leq 2L_{\max}$ and $\delta \leq L_{\max}$. Therefore, both conditions are weaker than assuming each $f_i$ is Lipschitz-smooth. We refer to discussions in (Jiang et al., 2024b) for more properties and details.

The previous two quantities $\delta$ and $\Delta_1$ will only affect the communication complexity of our algorithms, while the local complexity additionally depends on $L_1$ which is defined as follows.

**Assumption 2.3.** There exists $L_1 > 0$ such that for any $\mathbf{x}, \mathbf{y} \in \mathbb{R}^d$, we have:

$$\|\nabla f_1(\mathbf{x}) - \nabla f_1(\mathbf{y})\| \leq L_1 \|\mathbf{x} - \mathbf{y}\| . \tag{4}$$

## 3 INEXACT COMPOSITE GRADIENT METHOD

**Inexact Composite Gradient Method.** We first introduce the Inexact Composite Gradient Method (I-CGM), which serves as the backbone of our approach. Consider the composite reformulation of the problem 1: $f = f_1 + [f - f_1] = f_1 + h_1$. Let $\lambda > 0$ and $\mathbf{x}^0 \in \mathbb{R}^d$ be the initial point. At each iteration $t \geq 0$, I-CGM computes an approximation of the gradient $\mathbf{g}^t \approx \nabla f(\mathbf{x}^t)$ and defines the next iterate as:

$$\mathbf{x}^{t+1} \approx \underset{\mathbf{x} \in \mathbb{R}^d}{\arg\min} \left\{ F_t(\mathbf{x}) := f_1(\mathbf{x}) + h_1(\mathbf{x}^t) + \langle \mathbf{g}^t - \nabla f_1(\mathbf{x}^t), \mathbf{x} - \mathbf{x}^t \rangle + \frac{\lambda}{2} \|\mathbf{x} - \mathbf{x}^t\|^2 \right\},$$
(I-CGM)

where both the inaccuracy in solving the subproblem and the approximation error (defined below) are assumed to be sufficiently small (to be specified later):

$$F_t(\mathbf{x}^{t+1}) \leq F_t(\mathbf{x}^t), \quad e_t := \|\nabla F_t(\mathbf{x}^{t+1})\|, \quad \hat{\Sigma}_t^2 := \left\|\mathbf{g}^t - \nabla f(\mathbf{x}^t)\right\|^2 . \tag{5}$$

In the following statement, we provide the general convergence guarantee for I-CGM. The proof can be found in Section F.1 in the Appendix.

**Theorem 3.1.** *Let I-CGM be applied to Problem* (1)*. Suppose Assumption 2.1 and condition* (5) *are satisfied. Let $\lambda > \Delta_1$. Then for any $T \geq 1$, we have:*

$$\sum_{t=1}^{T} \|\nabla f(\mathbf{x}^t)\|^2 + (\lambda + \Delta_1)^2 \sum_{t=1}^{T} \|\mathbf{x}^t - \mathbf{x}^{t-1}\|^2$$

$$\leq \frac{12(\lambda + \Delta_1)^2}{\lambda - \Delta_1} F^0 + \left( \frac{12(\lambda + \Delta_1)^2}{(\lambda - \Delta_1)^2} + 4 \right) \sum_{t=0}^{T-1} \hat{\Sigma}_t^2 + 4 \sum_{t=0}^{T-1} e_t^2 .$$

We see that each subproblem can be solved inexactly without affecting the convergence rate (up to absolute constants), provided that the error term $\sum_{t=0}^{T-1} e_t^2$ is of the same order as the first two terms on the right-hand side. Moreover, if the approximation errors $\sum_{t=0}^{T-1} \hat{\Sigma}_t^2$ can also be bounded by the first two terms on the left-hand side, then the convergence of the gradient norm is guaranteed. If there exists randomness either in solving the subproblems or in constructing the estimators, then these conditions are required to hold in expectation. Specifically, we obtain the following corollary.

**Corollary 3.2.** *Following the same settings as in Theorem 3.1. If the inaccuracies in solving the subproblems satisfy:*

$$F_t(\mathbf{x}^{t+1}) \leq F_t(\mathbf{x}^t), \quad \sum_{t=0}^{T-1} \mathbb{E}[e_t^2] \leq \frac{(\lambda + \Delta_1)^2}{\lambda - \Delta_1} F^0 + \sum_{t=0}^{T-1} \Sigma_t^2 , \tag{6}$$

*and the approximation errors satisfy:*

$$\left( \frac{12(\lambda + \Delta_1)^2}{(\lambda - \Delta_1)^2} + 8 \right) \sum_{t=0}^{T-1} \Sigma_t^2 \leq \frac{1}{2} \sum_{t=1}^{T} G_t^2 + (\lambda + \Delta_1)^2 \sum_{t=1}^{T} \chi_t^2 , \tag{7}$$

*then for any $T \geq 1$, we have:*

$$\mathbb{E}[\|\nabla f(\bar{\mathbf{x}}^T)\|^2] \leq \frac{32(\lambda + \Delta_1)^2}{\lambda - \Delta_1} \frac{F^0}{T} .$$

*where $G_t^2 := \mathbb{E}[\|\nabla f(\mathbf{x}^t)\|^2]$, $\chi_t^2 := \mathbb{E}[\|\mathbf{x}^t - \mathbf{x}^{t-1}\|^2]$, $\Sigma_t^2 := \mathbb{E}[\|\mathbf{g}^t - \nabla f(\mathbf{x}^t)\|^2]$, and $\bar{\mathbf{x}}^T$ is uniformly sampled from $(\mathbf{x}^t)_{t=1}^{T}$.*

When $\mathbf{g}^t$ is the exact gradient $\nabla f(\mathbf{x}^t)$ for all $t \geq 0$, then I-CGM is reduced to CGM that is widely used for solving Problem (1), particularly because of its ability to exploit functional similarity and reduce communication costs (Hendrikx et al., 2020; Jiang et al., 2024a; Lin et al., 2024; Khaled & Jin, 2023; Jiang et al., 2024b; Mishchenko et al., 2024; Kovalev et al., 2022). Indeed, if $\lambda \simeq \Delta_1$ and the accuracy condition (6) is satisfied, then $\mathbb{E}[\|\nabla f(\bar{\mathbf{x}}^T)\|^2] \leq \varepsilon^2$ after $T = \mathcal{O}(\frac{\Delta_1 F^0}{\varepsilon^2})$ iterations.

In contrast, the iteration complexity of Gradient Descent depends on $L_f$ which can be larger than $\Delta_1$ (2.1) when $f_1$ is similar to $f$. However, CGM has sub-optimal communication complexity in terms of $n$. Indeed, let us assume, for simplicity, that $m = 1$. Then each iteration involves: 1) computing the full gradient $\nabla f(\mathbf{x}^t)$, which requires $n$ sequential communication rounds using A-CSS, and 2) an additional round using D-CSS for solving the subproblem. Consequently, the total number of communication rounds with A-CSS and D-CSS is $N_A = nT$ and $N_D = T$, respectively. The communication complexity of CGM is thus: $C_A N_A + N_D = C_A nT + T = \mathcal{O}(C_A nT) = \mathcal{O}(C_A \frac{n\Delta_1 F^0}{\varepsilon^2})$. This linear dependency on $n$ can be prohibitive in large-scale federated learning settings and is worse than the complexity of stochastic methods such as PROXSARAH (Pham et al., 2020), SPIDERBOOST (Wang et al., 2019), and PAGE (Li et al., 2021a), each of them achieving: $\mathcal{O}(C_A \frac{\sqrt{n}\bar{L} F^0}{\varepsilon^2})$, although they rely on a slightly different assumption of average smoothness [1]. Moreover, the dependence on $\frac{C_A}{\varepsilon^2}$ can become much larger in scenarios where using A-CSS is costly.

**Solving Auxiliary Subproblems.** In this section, we assume that $f_1$ is $L_1$-smooth and study how to achieve the accuracy condition (6). Recall that each subproblem $F_t$ consists of a smooth function $\phi(\mathbf{x}) = f_1(\mathbf{x})$ and a quadratic regularizer $\psi_t(\mathbf{x}) = \langle \mathbf{g}^t - \nabla f_1(\mathbf{x}^t), \mathbf{x} - \mathbf{x}^t \rangle + \frac{\lambda}{2} \|\mathbf{x} - \mathbf{x}^t\|^2$. Let us solve it using the standard composite gradient method (CGM), which proceeds as follows: For $k = 0, 1, ..., K_t - 1$,

$$
\begin{aligned}
\mathbf{y}_{k+1}^t &= \underset{\mathbf{y} \in \mathbb{R}^d}{\arg\min} \Big\{ \phi(\mathbf{y}_k^t) + \big\langle \nabla\phi(\mathbf{y}_k^t), \mathbf{y} - \mathbf{y}_k^t \big\rangle + \frac{L_1}{2} \big\|\mathbf{y} - \mathbf{y}_k^t\big\|^2 + \psi_t(\mathbf{y}) \Big\} \\
&= \frac{1}{\lambda + L_1} \big( L_1 \mathbf{y}_k^t + \lambda \mathbf{x}^t + \nabla f_1(\mathbf{x}^t) - \mathbf{g}^t - \nabla f_1(\mathbf{y}_k^t) \big) \, .
\end{aligned}
\tag{8}
$$

Each CGM step monotonically decreases the function value of $F_t$ (see Lemma F.2). Therefore, we can initialize $\mathbf{y}_0^t = \mathbf{x}^t$ and choose $\mathbf{x}^{t+1}$ to be a certain iterate of $(\mathbf{y}_k)_{k=0}^K$. Then the condition on $F_t(\mathbf{x}^{t+1}) \leq F_t(\mathbf{x}^t)$ is satisfied. We next study the number of local steps $K_t$ required to achieve the second inequality in condition (6).

**Fixed Number of Local Steps.** Let $K_t \equiv K \geq 1$ be a constant number and let $\mathbf{x}^{t+1}$ be the iterate with the minimum gradient norm of $F_t$ among $\{\mathbf{y}_k^t\}_{k=1}^K$. We use the notation $\mathbf{x}^{t+1} = \text{CGM}_{\text{const}}(\lambda, K, \mathbf{x}^t, \mathbf{g}^t)$ for this process.

The goal is to upper bound $\sum_{t=0}^{T-1} e_t^2$ where $e_t := \|\nabla F_t(\mathbf{x}^{t+1})\|$. For each $t \geq 0$, we have: $e_t^2 \lesssim \frac{L_1(F_t(\mathbf{y}_0^t) - F_t(\mathbf{y}_K^t))}{K} \lesssim \frac{L_1(f(\mathbf{x}^t) - f(\mathbf{y}_K^t) + \frac{1}{\lambda}\hat{\Sigma}_t^2)}{K}$ (see Lemma F.2 and F.1). However, since $\mathbf{y}_K^t$ and $\mathbf{x}^t$ are not necessarily the same, we cannot telescope $f(\mathbf{x}^t) - f(\mathbf{y}_K^t)$ when we sum up $e_t^2$. Instead, the "best" we can do is to upper bound $f(\mathbf{x}^t) - f(\mathbf{y}_K^t)$ by $f(\mathbf{x}^t) - f^\star$. Then by further upper-bounding the summation of $\sum_{t=0}^{T-1}[f(\mathbf{x}^t) - f^\star]$ in terms of $F^0$ and $\hat{\Sigma}_t^2$, it can be shown that we need $K \simeq \frac{L_1 T}{\lambda}$ local steps to achieve the desired accuracy condition (6). The proof can be found in Section F.2.1.

**Lemma 3.3.** *Consider I-CGM with $\mathbf{x}^{t+1} = \text{CGM}_{\text{const}}(\lambda, K, \mathbf{x}^t, \mathbf{g}^t)$ under Assumption 2.1 and 2.3. Let $T \geq 1$ be the fixed number in condition (6). Then by choosing $\lambda > \Delta_1$ and $K = K_T := \lceil \frac{8L_1 T}{\lambda - \Delta_1} \rceil$, the accuracy condition (6) is satisfied.*

**Random Number of Local Steps.** We now allow the number of local steps $K_t$ to follow a geometric distribution—a common technique used to derive last-iterate recurrences (Allen-Zhu, 2018b). When applied to solve the subproblems in I-CGM, this approach yields an algorithm that is efficient in local computation.

Let us consider CGM (8) with $K_t = \hat{K}_t + 1$ iterations where $\hat{K}_t \sim \text{Geom}(p)$, that is $\mathbb{P}(\hat{K}_t = k) = (1 - p)^k p$ for each $k \in \{0, 1, 2, ...\}$. The solution is set to be $\mathbf{x}^{t+1} = \mathbf{y}_{K_t}$. We use the notation $\mathbf{x}^{t+1} = \text{CGM}_{\text{rand}}(\lambda, \hat{K}_t, \mathbf{x}^t, \mathbf{g}^t)$ for this process.

In contrast to the convergence rate of using a deterministic $K$, we can now show that $\mathbb{E}_{\hat{K}_t}[e_t^2] \lesssim \frac{L_1^2 p}{L_1 + \lambda} \mathbb{E}_{\hat{K}_t}[F_t(\mathbf{x}^t) - F_t(\mathbf{x}^{t+1})]$. Using $\mathbb{E}_{\hat{K}_t}[F_t(\mathbf{x}^t) - F_t(\mathbf{x}^{t+1})] \lesssim \mathbb{E}_{\hat{K}_t}[f(\mathbf{x}^t) - f(\mathbf{x}^{t+1}) + \frac{1}{\lambda}\hat{\Sigma}_t^2]$, we get the telescoping term $\mathbb{E}[f(\mathbf{x}^t) - f(\mathbf{x}^{t+1})]$ after passing to the full expectation, which allows to improve the total amount of local computations.

---

[1] $\forall \mathbf{x}, \mathbf{y} \in \mathbb{R}^d$, it holds that $\frac{1}{n}\sum_{i=1}^n \|\nabla f_i(\mathbf{x}) - \nabla f_i(\mathbf{y})\|^2 \leq \bar{L}^2 \|\mathbf{x} - \mathbf{y}\|^2$ and we have $\delta \leq \bar{L}$.

**Lemma 3.4.** *Consider I-CGM with $\mathbf{x}^{t+1} = \mathrm{CGM}_{\mathrm{rand}}(\lambda, \hat{K}_t, \mathbf{x}^t, \mathbf{g}^t)$ where $\hat{K}_t \sim \mathrm{Geom}(p)$ under Assumption 2.1 and 2.3. Let $T \geq 1$ be the fixed number in condition (6). Then by choosing $\lambda > \Delta_1$ and $p = \frac{\lambda - \Delta_1}{8(L_1 + \lambda)} < 1$, the accuracy condition (6) is satisfied.*

To achieve the accuracy condition (6), the number of local first-order oracle queries required by using the random $\hat{K}_t$ at each iteration $t$ in expectation is $\mathbb{E}_{\hat{K}_t}[K_t] = \frac{1}{p} \simeq \frac{L_1}{\lambda}$, which improves upon the previous result of $\frac{L_1 T}{\lambda}$ obtained by using a fixed number of $K$.

So far, we have studied how to solve the subproblems of I-CGM such that the accuracy condition (6) is satisfied. We now turn to constructing the gradient estimator $\mathbf{g}^t$ that has the desired approximation error (7). Meanwhile, we aim to improve both the dependence on $n$ and $C_A$ in the communication complexity of CGM. The main strategy is to design a gradient estimator whose approximation error depends only on the similarity constant $\delta$ while avoiding periodic full synchronizations.

## 4 BASIC APPLICATION EXAMPLES: SAGA + SVRG

In this section, we present two algorithms that maintain an approximation of the gradient, $\mathbf{G}^t \approx \nabla f(\mathbf{x}_t)$ for $t \geq 0$. Each algorithm starts with an initial point $\mathbf{x}^0$. Then at each iteration $t \geq 0$, $\mathbf{G}^t$ is computed first, after which the next iterate $\mathbf{x}^{t+1}$ is computed. In what follows, for a set $S \in \binom{[n]}{m}$ and $m \in [n]$, we use $f_S := \frac{1}{m} \sum_{i \in S} f_i$ to denote the average function over this set.

For convenience of presentation, we use the following notations throughout the rest of the paper:

$$n_m := \frac{n}{m}, \quad q_m := \frac{n - m}{n - 1}, \quad \text{and} \quad \delta_m^2 := \frac{q_m}{m} \delta^2 . \tag{9}$$

**SAGA Estimator**. SAGA estimator is a variance-reduction technique based on incremental gradient updates, originally designed for centralized finite-sum minimization (Defazio et al., 2014). In this section, we adapt this estimator to the federated optimization scenario and study its properties.

The SAGA estimator defines:

$$\mathbf{G}^0 = \nabla f(\mathbf{x}^0), \quad \mathbf{G}^1 = \nabla f(\mathbf{x}^1), \quad \mathbf{G}^t = \mathbf{b}_{S_t}^t - \mathbf{b}_{S_t}^{t-1} + \mathbf{b}^{t-1}, \, t \geq 2 , \tag{SAGA}$$

where $S_t \in \binom{[n]}{m}$ is uniformly sampled at random without replacement, $\mathbf{b}_{S_t}^t := \frac{1}{m} \sum_{i \in S_t} \mathbf{b}_i^t$, $\mathbf{b}_{S_t}^{t-1} := \frac{1}{m} \sum_{i \in S_t} \mathbf{b}_i^{t-1}$, $\mathbf{b}^t := \frac{1}{n} \sum_{i=1}^n \mathbf{b}_i^t$, and for any $i \in [n]$, $\mathbf{b}_i^t$ is recurrently defined as:

$$\mathbf{b}_i^0 = \nabla f_i(\mathbf{x}^0), \quad \mathbf{b}_i^1 = \nabla f_i(\mathbf{x}^1), \quad \mathbf{b}_i^t = \begin{cases} \nabla f_i(\mathbf{x}^t) & \text{if } i \in S_t, \\ \mathbf{b}_i^{t-1} & \text{otherwise,} \end{cases}, \quad t \geq 2 .$$

We have the following recurrence for $\mathbf{b}^t$ (the derivation can be found in Lemma F.3):

$$\mathbf{b}^t = \mathbf{b}^{t-1} + \frac{1}{n_m}[\nabla f_{S_t}(\mathbf{x}^t) - \mathbf{b}_{S_t}^{t-1}], \quad t \geq 2 . \tag{10}$$

**Implementation.** At the beginning, when $t = 0$ and $1$, each client $i = 1, \dots, n$ computes $\nabla f_i(\mathbf{x}^t)$ and initializes $\mathbf{b}_i^t$ and sends the result to the server; the server then aggregates these results computing $\nabla f(\mathbf{x}^t)$ to initialize $\mathbf{G}^t$ and $\mathbf{b}^t$. This requires two full synchronizations ($2\lceil n_m \rceil$ communications rounds using A-CSS). At each iteration $t \geq 2$, the server contacts the randomly selected set of clients $S_t$ using R-CSS and sends $\mathbf{x}^t$ to them. Each client $i \in S_t$ computes $\mathbf{b}_i^t = \nabla f_i(\mathbf{x}^t)$ and sends $\mathbf{b}_i^t - \mathbf{b}_i^{t-1}$ back to the server. The server then updates $\mathbf{b}^t$ according to (10) and constructs the gradient estimator $\mathbf{G}^t$ using the stored $\mathbf{b}^{t-1}$ according to (SAGA).

Each client $i$ thus needs to store a single vector $\mathbf{b}_i^t$. On the server side, only the aggregated vector $\mathbf{b}^t$ and the iterate $\mathbf{x}^t$ need to be maintained. The memory overhead of the SAGA estimator is thus very small in the federated learning setting, similarly to the SAG estimator (Schmidt et al., 2017) used in SCAFFOLD (Karimireddy et al., 2020).

**Properties of SAGA.** It is not difficult to show that $\mathbf{G}^t$ is a conditionally unbiased estimator of $\nabla f(\mathbf{x}^t)$, namely, $\mathbb{E}_{S_t}[\mathbf{G}^t] = \nabla f(\mathbf{x}^t)$. We next present a new variance bound for SAGA that is controlled by the constant $\delta$. The proof can be found in Section F.3.1.

**Lemma 4.1.** *Consider the* SAGA *estimator* (SAGA) *under Assumption 2.2. Then for any $t \geq 2$,*
$\mathbb{E}_{S_t}[\mathbf{G}^t] = \nabla f(\mathbf{x}^t)$ *and for any $T \geq 1$, we have :*

$$\sum_{t=0}^{T} \sigma_t^2 \leq \frac{2n_m q_m}{m} G_1^2 + \frac{n_m - 1 + \sqrt{n_m^2 - n_m}}{(n-1)} \sum_{t=2}^{T-1} G_t^2 + 4n_m^2 \delta_m^2 \sum_{t=2}^{T} \chi_t^2 ,$$

*where $\sigma_t^2 := \mathbb{E}_{S_{[t]}}[\|\mathbf{G}^t - \nabla f(\mathbf{x}^t)\|^2]$, $G_t^2 = \mathbb{E}_{S_{[t-1]}}[\|\nabla f(\mathbf{x}^t)\|^2]$, $\chi_t^2 := \mathbb{E}_{S_{[t-1]}}[\|\mathbf{x}^t - \mathbf{x}^{t-1}\|^2]$, and $S_{[t]} := (S_2, ..., S_t)$.*

Note that this variance bound depends on $G_1^2, \ldots, G_{T-1}^2$ and $\chi_2^2, \ldots, \chi_T^2$, which aligns with the terms on the right-hand side of the desired error bound (7). However, the coefficient in front of $G_1^2$ in this bound can be larger than 1, whereas (7) requires it to be strictly less than 1. Consequently, the requirement is not met and we cannot directly incorporate the SAGA estimator into I-CGM by setting $\mathbf{g}^t = \mathbf{G}^t$. We will show in Section 5 that this error bound can be significantly improved by using the recursive gradient estimation technique.

*Remark* 4.2. Instead of computing the exact gradients $\nabla f(x^0)$ and $\nabla f(\mathbf{x}^1)$ at the beginning which requires full synchronizations, it is possible to start with an approximation $\mathbf{G}^0 \approx \nabla f(\mathbf{x}^0)$. This requires only one communication round using R-CSS. The resulting communication-complexity estimate will now additionally depend on the inexactness of the initial approximation but this strategy often works well in practice (Figure J.1). See Appendix G for detailed discussions.

**SVRG Estimator**. Another possible choice of the gradient estimator is the SVRG estimator (Johnson & Zhang, 2013). There are different variants of SVRG, and here we consider the so-called loopless-SVRG estimator (Kovalev et al., 2020) for simplicity.

The SVRG estimator defines:

$$\boxed{\mathbf{G}^0 = \nabla f(\mathbf{x}^0), \quad \mathbf{G}^t = \nabla f_{S_t}(\mathbf{x}^t) + \nabla f(\mathbf{w}^t) - \nabla f_{S_t}(\mathbf{w}^t), \ t \geq 1 ,} \tag{SVRG}$$

where $S_t \in \binom{[n]}{m}$ is uniformly sampled at random without replacement,

$$\mathbf{w}^0 = \mathbf{x}^0, \quad \mathbf{w}^t = \begin{cases} \mathbf{x}^t & \text{if } \omega_t = 1, \\ \mathbf{w}^{t-1} & \text{otherwise,} \end{cases} \quad t \geq 1 ,$$

and $\omega_t$ is a Bernoulli random variable with parameter $p_B$, i.e., $P(\omega_t = 1) = p_B \in (0, 1)$.

**Properties:** It is not difficult to show that the SVRG estimator $\mathbf{G}^t$ is a conditionally unbiased estimator of $\nabla f(\mathbf{x}^t)$, namely, $\mathbb{E}_{S_t}[\mathbf{G}^t] = \nabla f(\mathbf{x}^t)$. Moreover, the variance is controlled by $\delta$. The proof can be found in Section F.3.2 where the implementation of the estimator is also provided.

**Lemma 4.3.** *Consider the* SVRG *estimator* (SVRG) *under Assumption 2.2. Then for any $t \geq 1$, $\mathbb{E}_{S_t}[\mathbf{G}^t] = \nabla f(\mathbf{x}^t)$ and for any $T \geq 1$, we have: $\sum_{t=0}^{T} \sigma_t^2 \leq \frac{4\delta_m^2}{p_B^2} \sum_{t=1}^{T} \chi_t^2$, where $\sigma_t^2 := \mathbb{E}_{S_t, \omega_{[t]}}[\|\mathbf{G}^t - \nabla f(\mathbf{x}^t)\|^2]$, $\chi_t^2 := \mathbb{E}_{\omega_{[t-1]}}[\|\mathbf{x}^t - \mathbf{x}^{t-1}\|^2]$, and $\omega_{[t]} := (\omega_1, ..., \omega_t)$.*

We can incorporate the SVRG estimator into I-CGM by setting $\mathbf{g}^t = \mathbf{G}^t$. This requires setting $p_B \simeq \frac{1}{n_m}$ and $\lambda \simeq \Delta_1 + n_m \delta_m$ to achieve the error condition (7). The resulting communication complexity of the method is $\mathcal{O}(C_A n_m + \frac{(C_R \Delta_1 + C_A n_m \delta_m) F^0}{\varepsilon^2})$, which still has a linear dependence on $n_m$. (See Theorem F.4 with the proof that the reader can inspect if interested). Note that unbiasedness is not needed to incorporate SVRG directly into I-CGM. However, it becomes necessary later for the recursive gradient technique, which we discuss in the next section.

## 5   RECURSIVE GRADIENT ESTIMATOR + EXAMPLES (SAGA AND SVRG)

In this section, we present a general formular of the recursive gradient estimator that can potentially improve the error bound for a given conditionally unbiased gradient estimator $\mathbf{G}^t \approx \nabla f(\mathbf{x}^t)$. Formally, we consider the following setting.

**Assumption 5.1.** For any $t \geq 0$, it holds that: 1) $S_t$ is independent of $\mathbf{x}^0, \ldots, \mathbf{x}^{t+1}, \mathbf{G}^0, \ldots, \mathbf{G}^{t-1}$; 2) $\mathbb{E}_{S_t}[\mathbf{G}^t] = \nabla f(\mathbf{x}^t)$.

The recursive gradient estimator (RG) defines:

$$\boxed{\mathbf{g}^0 = \nabla f(\mathbf{x}^0), \quad \mathbf{g}^{t+1} = (1-\beta)\mathbf{g}^t + \beta\mathbf{G}^t + \nabla f_{S_t}(\mathbf{x}^{t+1}) - \nabla f_{S_t}(\mathbf{x}^t), \ t \geq 0 ,} \quad \text{(RG)}$$

where $\beta \in (0,1]$ and $S_t \in \binom{[n]}{m}$ is uniformly sampled at random without replacement.

Note that the indexing here differs from the previous ones. The algorithm starts with an initial point $\mathbf{x}^0$. At each iteration $t \geq 0$, the estimator $\mathbf{g}^t \approx \nabla f(\mathbf{x}^t)$ is computed first and it depends only on $\mathbf{G}^{t-1}$ and $S_{t-1}$. After that, the next iterate $\mathbf{x}^{t+1}$ is computed. Therefore, $\mathbf{x}^{t+1}$ is independent from $S_t$ while previously it was dependent on it (if we use the SAGA/SVRG estimator).

Inspired by previous works, the expression of $\mathbf{g}^t$ incorporates both recursive gradient update and momentum (Chayti et al., 2025; Gao et al., 2024). This expression unifies several existing methods: When $\beta = 0$, the estimator reduces to the SARAH update rule (Nguyen et al., 2017). When $\mathbf{G}^t$ is replaced with the SAGA estimator, then $\mathbf{g}^t$ recovers the structure of ZEROSARAH (Li et al., 2021b). When $\mathbf{G}^t = \nabla f_{S_t}(\mathbf{x}^{t+1})$ and $\nabla f_{S_t}(\mathbf{x}^{t+1}) - \nabla f_{S_t}(\mathbf{x}^t)$ is multiplied by $1 - \beta$, then it becomes STORM (Cutkosky & Orabona, 2019). In our formulation, $\mathbf{G}^t$ is a general similarity-aware estimator of $\nabla f(\mathbf{x}^t)$ that satisfies Assumption 5.1, allowing us to flexibly instantiate the framework with various variance-reduction techniques.

For instance, we can combine RG with SAGA or SVRG. We refer to the resulting estimators as RG-SAGA and RG-SVRG. Note that $S_t$ in the formulas for SAGA and SVRG is exactly the same random index set that is used in the RG – they share the same randomness for the sake of efficiency.

**Implementation.** At the beginning, each client $i = 1, \ldots, n$ computes $\nabla f_i(\mathbf{x}^0)$ and sends the result to the server; the server then aggregates these results, computing $\nabla f(\mathbf{x}^0)$ to initialize $\mathbf{g}^0$. This requires one full synchronization. Then $\mathbf{x}^1$ is computed based on $\mathbf{g}^0$. At each iteration $t \geq 0$, the server uses R-CSS which generates a random subset $S_t$. For RG-SAGA, the server sends $\mathbf{x}^{t+1}$, $\mathbf{x}^t$ to the clients in $S_t$. Each client $i \in S_t$ updates $\mathbf{b}_i^t = \nabla f_i(\mathbf{x}^t)$ and sends $\nabla f_i(\mathbf{x}^{t+1})$ along with $\mathbf{b}_i^t - \mathbf{b}_i^{t-1}$ (when $t \geq 2$) or $\mathbf{b}_i^t$ (when $t = 1$) to the server. For RG-SVRG, the server sends $\mathbf{x}^{t+1}$, $\mathbf{x}^t$ and $\mathbf{w}^t$ to the clients which then return the gradients evaluated at these three points. If $\omega_t = 1$, the server additionally computes the new gradient $\nabla f(\mathbf{w}^t)$ performing one full synchronization. After receiving all the vectors, the server can compute $\nabla f_{S_t}(\mathbf{x}^{t+1}), \nabla f_{S_t}(\mathbf{x}^t), \mathbf{G}^t$ and $\mathbf{g}^{t+1}$.

For RG-SAGA, each client $i$ needs to store a single vector $\mathbf{b}_i^t$ and the server needs to maintain two points $\mathbf{x}^{t+1}$ and $\mathbf{x}^t$, and two vectors $\mathbf{b}^t$ and $\mathbf{g}^t$. For RG-SVRG, clients are stateless and the server is required to maintain three points $\mathbf{x}^{t+1}$, $\mathbf{x}^t$, $\mathbf{w}^t$ and one vector $\nabla f(\mathbf{w}^t)$.

**Lemma 5.2** (Error bound for RG). *Consider the RG estimator* (RG) *under Assumptions 5.1 and 2.2. Then for any $T \geq 1$, we have:* $\sum_{t=0}^{T} \Sigma_t^2 \leq \frac{2\beta}{2-\beta} \sum_{t=0}^{T-1} \sigma_t^2 + \frac{2\delta_m^2}{2\beta-\beta^2} \sum_{t=1}^{T} \chi_t^2$ . *where* $\Sigma_t^2 := \mathbb{E}_{S_{[t-1]}}[\|\mathbf{g}^t - \nabla f(\mathbf{x}^t)\|^2]$, $\sigma_t^2 := \mathbb{E}_{S_{[t]}}[\|\mathbf{G}^t - \nabla f(\mathbf{x}^t)\|^2]$, $\chi_t^2 := \mathbb{E}_{S_{[t-2]}}[\|\mathbf{x}^t - \mathbf{x}^{t-1}\|^2]$, *and* $S_{[t]} := (S_0, \ldots, S_t)$.

The proof can be found in Section F.4.1. We next show that the error bound of both SAGA and SVRG can be improved by combining them with RG and adjusting the parameter $\beta$. For instance, by combining Lemma 5.2 and Lemma 4.1, we obtain the following result for RG-SAGA.

**Corollary 5.3.** *Consider the RG-SAGA estimator under Assumptions 5.1 and 2.2. Then for any $T \geq 1$, it holds that:*

$$\sum_{t=0}^{T} \Sigma_t^2 \leq \frac{4\beta n_m q_m}{(2-\beta)m} G_1^2 + \frac{2\beta(n_m - 1 + \sqrt{n_m^2 - n_m})}{(2-\beta)(n-1)} \sum_{t=2}^{T-1} G_t^2 + \frac{8\beta^2 n_m^2 \delta_m^2 + 2\delta_m^2}{2\beta - \beta^2} \sum_{t=1}^{T} \chi_t^2 ,$$

*where* $\Sigma_t^2 := \mathbb{E}_{S_{[t-1]}}[\|\mathbf{g}^t - \nabla f(\mathbf{x}^t)\|^2]$, $G_t^2 := \mathbb{E}_{S_{[t-2]}}[\|\nabla f(\mathbf{x}^t)\|^2]$, $\chi_t^2 := \mathbb{E}_{S_{[t-2]}}[\|\mathbf{x}^t - \mathbf{x}^{t-1}\|^2]$ *and* $S_{[t]} := (S_0, \ldots, S_t)$.

By choosing $\beta \simeq \frac{1}{n_m}$, we get $\sum_{t=0}^{T} \Sigma_t^2 \lesssim \frac{q_m}{m} G_1^2 + \frac{1}{n} \sum_{t=2}^{T-1} G_t^2 + n_m \delta_m^2 \sum_{t=1}^{T} \chi_t^2$. Compared with the original variance bound for SAGA (Lemma 4.1), the bound with RG achieves an improvement by a factor of $n_m$.

The error bound for the SVRG estimator can be improved in a similar way.

**Corollary 5.4.** *Consider the RG-SVRG estimator under Assumptions 5.1 and 2.2. Then for any $T \geq 1$, it holds that: $\sum_{t=0}^{T} \Sigma_t^2 \leq \frac{8\beta^2 \delta_m^2 / p_B^2 + 2\delta_m^2}{2\beta - \beta^2} \sum_{t=1}^{T} \chi_t^2$, where $\Sigma_t^2 := \mathbb{E}_{S_{[t-1]}, \omega_{[t-1]}}[\|\mathbf{g}^t - \nabla f(\mathbf{x}^t)\|^2]$, $\chi_t^2 := \mathbb{E}_{S_{[t-2]}, \omega_{[t-2]}}[\|\mathbf{x}^t - \mathbf{x}^{t-1}\|^2]$, $S_{[t]} := (S_0, \ldots, S_t)$ and $\omega_{[t]} := (\omega_1, \ldots, \omega_t)$.*

Compared with the original variance bound for SVRG (Lemma 4.3), the new bound achieves an improvement by a factor of $1/p_B$ by choosing $\beta \simeq p_B$.

We can now incorporate both enhanced estimators into I-CGM. It can be shown that the iterates $\{\mathbf{x}^t\}_{t=0}^{\infty}$ generated by I-CGM-RG-SAGA or I-CGM-RG-SVRG and the corresponding sequence $\{\mathbf{G}^t\}_{t=0}^{\infty}$ satisfy Assumption 5.1. (See Lemma F.5 and F.6).

## 6 COMMUNICATION AND LOCAL COMPLEXITY OF I-CGM-RG

We are ready to establish the complexity of I-CGM equipped with the RG-SAGA and RG-SVRG estimator. We first present the result for RG-SAGA. The proof can be found in Section F.5.1.

**Theorem 6.1** (I-CGM-RG-SAGA). *Let I-CGM be applied to Problem 1 under Assumptions 2.1, 2.2 and 2.3, where $\mathbf{x}^{t+1} = \text{CGM}_{\text{rand}}(\lambda, \hat{K}_t, \mathbf{x}^t, \mathbf{g}^t)$ with $\hat{K}_t \sim \text{Geom}(p)$ and $\mathbf{g}^t$ is generated by the RG-SAGA estimator. Then by choosing $\lambda = 3\Delta_1 + 113\sqrt{n_m}\delta_m$, $\beta = \frac{1}{112 n_m}$ and $p = \frac{\lambda - \Delta_1}{8(L_1 + \lambda)}$, after $T = \lceil \frac{(256(\Delta_1 + 38\sqrt{n_m}\delta_m)F^0}{\varepsilon^2} \rceil$ iterations, we have $\mathbb{E}[\|\nabla f(\bar{\mathbf{x}}^T)\|^2] \leq \varepsilon^2$, where $\bar{\mathbf{x}}^T$ is is uniformly sampled from $(\mathbf{x}^t)_{t=1}^{T}$. The communication complexity is at most $2C_A \lceil n_m \rceil + (C_R + 1) \lceil \frac{(256(\Delta_1 + 38\sqrt{n_m}\delta_m)F^0}{\varepsilon^2} \rceil$ and the local complexity is bounded by $14 + 2\lceil n_m \rceil + \frac{512(7\Delta_1 + 283\sqrt{n_m}\delta_m + 2L_1)F^0}{\varepsilon^2} + \frac{4L_1}{\Delta_1 + 28\sqrt{n_m}\delta_m}$.*

The communication complexity of I-CGM-RG-SAGA is of order $C_A n_m + C_R \frac{\Delta_1 + (\sqrt{n_m}\delta_m)F^0}{\varepsilon^2}$ and the local complexity is of order $n_m + \frac{(\Delta_1 + \sqrt{n_m}\delta_m + L_1)F^0}{\varepsilon^2}$ when $\frac{(\Delta_1 + \sqrt{n_m}\delta_m)F^0}{\varepsilon^2} \gtrsim 1$. The $n_m$ term comes from $n_m$ sequential rounds with A-CSS for computing the full gradients in the beginning.

**Comparison: RG-SVRG Estimator.** The communication complexity of I-CGM-RG-SVRG is $\mathcal{O}\left(C_A n_m + \frac{(C_R \Delta_1 + \sqrt{C_A C_R n_m}\delta_m)F^0}{\varepsilon^2}\right)$, where $C_A$ also affects the term involving $\varepsilon$ (see Appendix F.6 for details and the result of the local complexity).

## 7 NUMERICAL EXPERIMENTS

In this section, we verify the theory of the proposed methods in numerical experiments. We set $C_A = C_R = 1$ in the definition of communication complexity for all the experiments. We choose this case to demonstrate that even when A-CSS and R-CSS are equally cheap, our proposed methods already outperform several commonly used algorithms. (The study of the scenario when $C_A > C_R$ can be found in Appendix J.1.1.)

**Quadratic minimization with log-sum penalty.** Consider the problem of $f(\mathbf{x}) = \frac{1}{n} \sum_{i=1}^{n} f_i(\mathbf{x})$ with $f_i(\mathbf{x}) := \frac{1}{b} \sum_{j=1}^{b} \frac{1}{2} \langle \mathbf{A}_{i,j}(\mathbf{x} - \mathbf{b}_{i,j}), \mathbf{x} - \mathbf{b}_{i,j} \rangle + \sum_{k=1}^{d} \log(1 + \alpha |\mathbf{x}_k|)$, where $\alpha > 0$, $\mathbf{b}_{i,j} \in \mathbb{R}^d$, $\mathbf{A}_{i,j} \in \mathbb{R}^{d \times d}$ is a diagonal matrix, and $\cdot_k$ is an indexing operation of a vector. We set $\alpha = 10$, $b = 5$, $n = 100$ and $d = 1000$. Each coordinate of $\mathbf{b}_{i,j}$ is uniformly sampled from $[0, 10]$. To generate $\mathbf{A}_{i,j}$, we first sample a diagonal matrix $\bar{\mathbf{A}}$ with entries uniformly distributed in $[0, 110]$, and then add $bn$ diagonal noise matrices whose entries are sampled from $[0, 18]$. Each resulting $\mathbf{A}_{i,j}$ is clipped to the interval $[1, 100]$ on the diagonal, and some eigenvalues are further set close to zero. Consequently, the dataset satisfies $\mathbf{0} \preceq \mathbf{A}_{i,j} \preceq 100\mathbf{I}$ for any $i, j$, with $\Delta_1 \approx \delta \approx 5$ and $L_{\max} \approx 100$. We set $m = \sqrt{n}$. For I-CGM-RG, we set $p = \frac{\delta}{L}$, $\lambda = \frac{\sqrt{n}}{m}\delta + \Delta_1$, $\eta = 2L_{\max}$ and $\beta = \frac{m}{n}$. We compare two proposed algorithms against SCAFFOLD (Karimireddy et al., 2020), FEDAVG (McMahan et al., 2017) (with sampling), SABER-FULL (Mishchenko et al., 2024) (with PAGE), SABER-PARTIAL (Mishchenko et al., 2024) (only compute full gradient once) and GD (running directly on $f$). For SVRG-based methods, the expected number of communication rounds at each iteration is roughly $\frac{m}{n}n + m$, which is twice as large as other methods. From Figure 1, we observe that: 1) I-CGM-RG-SAGA is the most efficient in both communication and local computation. 2) SCAFFOLD cannot fully exploit $\delta$-similarity as its local complexity is comparable to GD (the theoretical local complexity of both

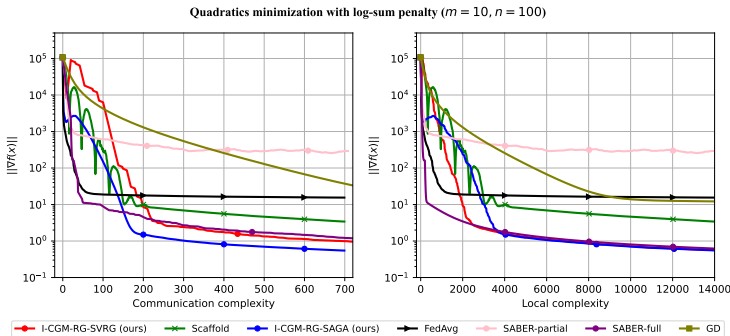

Figure 1: Comparisons of different algorithms for solving the quadratic minimization problems with non-convex log-sum penalty.

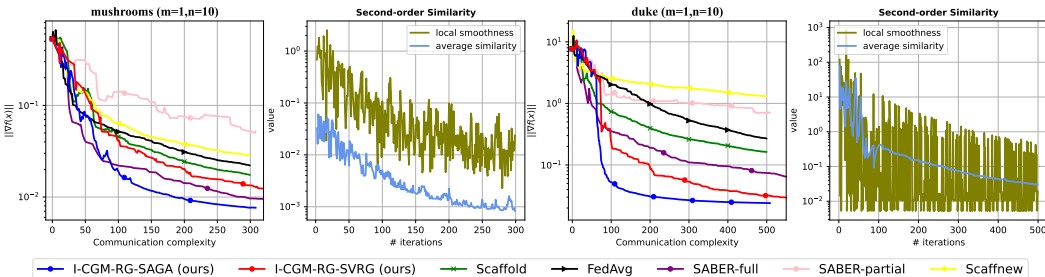

Figure 2: Comparisons of different algorithms on two LIBSVM datasets using logistic loss with non-convex regularizer.

methods depends on $L_{\max}$). Finally, I-CGM-RG-SAGA with different initialization strategies can be found in Figure J.1.

**Logistic regression with nonconvex regularizer.** We now experiment with the binary classification task on two real-world LIBSVM datasets (Chang & Lin, 2011). We use the standard regularized logistic loss: $f(\mathbf{x}) = \frac{1}{n} \sum_{i=1}^{n} f_i(\mathbf{x})$ with $f_i(\mathbf{x}) := \frac{n}{M} \sum_{j=1}^{m_i} \log(1 + \exp(-y_{i,j} \langle \mathbf{a}_{i,j}, \mathbf{x} \rangle)) + \alpha \sum_{k=1}^{d} \frac{[\mathbf{x}]_k^2}{1+[\mathbf{x}]_k^2}$ where $\alpha > 0$, $(\mathbf{a}_{i,j}, y_{i,j}) \in \mathbb{R}^{d+1}$ are feature and labels and $M := \sum_{i=1}^{n} m_i$ is the total number of data points. We use $m = 1$ and $n = 10$. We plot the local $L_1$ and $\delta$ by computing $\left\| \nabla f_1(\mathbf{x}^t) - \nabla f_1(\mathbf{x}^{t+1}) \right\| / \left\| \mathbf{x}^t - \mathbf{x}^{t+1} \right\|$ and $\sqrt{\frac{1}{n} \sum_{i=1}^{n} \left\| \nabla h_i(\mathbf{x}^t) - \nabla h_i(\mathbf{x}^{t+1}) \right\|^2 / \left\| \mathbf{x}^t - \mathbf{x}^{t+1} \right\|^2}$ along the iterates of I-CGM-RG-SAGA. From Figure 2, we observe that $\delta$ is much smaller than $L_1$ for the mushrooms dataset, while being comparable for the duke dataset. However, for both cases, I-CGM-RG-SAGA remains the most efficient in communication complexity.

**Deep learning tasks.** We defer the study of neural network training to Appendix J, where more experiments and details can be found.

## 8 CONCLUSION

We introduced a new simple model for comparing centralized distributed optimization algorithms, where different client-selection strategies are associated with non-uniform costs. Within this model, we developed a new family of algorithm based on inexact composite gradient method with recursive gradient estimator. This design enables us to exploit functional similarity among clients while supporting partial client participation—a key requirement in practical FL systems. It is efficient when full synchronizations (requiring sequential communications with all clients) are costly compared to client sampling. The key technical contribution of this work is a new variance bound for the SAGA estimator, which depends on the functional similarity constant $\delta$ rather than individual smoothness. This allows the SAGA-based variant of I-CGM-RG to outperform the previously best-known communication complexity of SARAH-based methods. Limitations and future extensions are discussed at the end of the Appendix.

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

# Appendix

## CONTENTS

## A ETHICS STATEMENT

This paper presents work that aims to advance the field of Distributed Machine Learning. There are many potential societal consequences of our work, none of which we feel must be specifically highlighted here.

## B LLM USAGE

We used LLM only for polishing the text.

## C  FEDERATED OPTIMIZATION ALGORITHMS AND THEIR COMPLEXITY

**Federated Optimization Algorithm.** We consider the standard federated optimization setting with a central server and $n$ clients. The server is the main entity that implements the optimization algorithm, but cannot directly access any of the local functions $(f_i)_{i=1}^n$. Instead, it interacts with problem (1) through communications with the clients, allowing certain information to be exchanged between them. Each client $i \in [n]$ has access to the information provided by the server and can interact with its own local function $f_i$, but only through the oracle $O_{f_i}$. An oracle is a standard notion in optimization (Nemirovsky, 1994; Nemirovsky & Yudin, 1983), which is a procedure that takes as input a point and returns certain information about the function at this point. The most commonly used oracle is the first-order oracle, which returns the function value and its gradient. In general, an oracle can be stochastic; however, in this work, we mainly consider the standard deterministic first-order oracle $O_{FO_i}(\mathbf{x}) := (f_i(\mathbf{x}), \nabla f_i(\mathbf{x}))$. Throughout the paper, we assume that the server can communicate with up to $m \in [n]$ clients simultaneously in parallel. We formalize optimization algorithms in this setting as follows.

Given the oracles $O_{f_1}, \ldots, O_{f_n}$, a *federated optimization algorithm* for a problem class $\mathcal{F}$ is a procedure that proceeds across *communication rounds*. A *problem class* is the collection of all problems of form (1) satisfying certain assumptions. (We will introduce a specific problem class considered in this work in Section 2.2.) At the beginning, the server and each client $i \in [n]$ initialize the empty information sets $\mathcal{I}^0$ and $\mathcal{H}_i^0$, respectively. At each round $r \geq 0$, the server chooses a subset of clients $S_r \subseteq [n]$ with at most $m$ elements (to be discussed later). The server then communicates with the clients in $S_r$, providing each client $i \in S_r$ with certain information $\bar{\mathcal{I}}_i^r$ constructed from the server's information set $\mathcal{I}^r$. Then it specifies a certain method $\mathcal{M}_i^r$ (often called a *local method*) for each client $i \in S_r$ to run locally. The method $\mathcal{M}_i^r$ starts with the initial information $(\bar{\mathcal{I}}_i^r, \mathcal{H}_i^r)$, and iteratively queries the oracle $O_{f_i}$, obtaining a response $\mathcal{R}_i^r$, which is then sent back to the server. (The details of this procedure are discussed in the next paragraph.) The server collects the output responses and updates the information set $\mathcal{I}^{r+1} = (\mathcal{I}^r, (\mathcal{R}_i^r)_{i \in S_r})$. At each round $r \geq 0$, the server also performs a termination test based on the current information set $\mathcal{I}^r$. After the algorithm terminates at a certain round $R \geq 0$, the server then constructs and outputs an approximate solution $\hat{\mathbf{x}}^R$ to problem (1) based on $\mathcal{I}^R$ using a certain rule specified by the algorithm. To summarize, a federated optimization algorithm is a collection of rules prescribing what to do at each communication round $r$: 1) how to select clients, 2) how to compute the information $\bar{\mathcal{I}}_i^r$ that is sent to each selected client, 3) which local method each selected client runs, 4) when to terminate, and 5) how to form the approximate solution. We allow each of these rules to be randomized. See Figure D.1 for an illustration summarizing the procedures described above.

At the beginning of each round $r$, each selected client $i \in S_r$ receives the information $\bar{\mathcal{I}}_i^r$ from the server. Using this new information, it enriches its information set $\mathcal{H}_{i,0}^r := (\mathcal{H}_i^r, \bar{\mathcal{I}}_i^r)$ and runs the specified method $\mathcal{M}_i^r$. At each step $k \geq 0$, this method first computes a point $\mathbf{x}_{i,k}^r$ based on $\mathcal{H}_{i,k}^r$, queries the oracle at this point, and then updates its local information: $\mathcal{H}_{i,k+1}^r = (\mathcal{H}_{i,k}^r, O_{f_i}(\mathbf{x}_{i,k}^r))$. At the beginning of each step $k$, the method also performs a termination test $T_{i,k}^r(\mathcal{H}_{i,k}^r)$; Once this test is satisfied at a certain step $K_i^r$, the method terminates and constructs the output $\mathcal{R}_i^r$ from the final information $\mathcal{H}_{i,K_i^r}^r$. The information sets are then updated as $\mathcal{H}_i^{r+1} := \mathcal{H}_{i,K_i^r}^r$, and remain the same ($\mathcal{H}_i^{r+1} := \mathcal{H}_i^r$) for each non-selected client $i \notin S_r$. To summarize, a local method $\mathcal{M}_i^r$ is a collection of 3 rules: 1) how to compute the next point at each step, 2) when to terminate, and 3) how to form the result. We allow each of these rules to be randomized (resulting in a *randomized* local method); if all the rules are deterministic, the local method is called *deterministic*.

Note that the above definition of a distributed optimization algorithm is rather general and only constrains how the algorithm accesses information about the optimization problem. In particular, we do not impose any restrictions on the arithmetic or memory complexity of each step of the algorithm, nor on the size of the data transmitted between the server and the clients. This general definition is sufficient to introduce the two *information-based* notions of complexity that we focus on in this work: communication and local complexities (defined below). In practice, however, both memory storage and information usage should be implemented efficiently. Typically, the accumulated information sets maintained by the clients and the server, as well as the information exchanged between them, are simply a collection of a few vectors and scalars.

**Client-Selection Strategies.** We next introduce three commonly used client-selection strategies and associate them with different costs. The distinction among them lies in how the set $S_r$ is selected.

- *Arbitrary Client Selection Strategy* (A-CSS): The set $S_r$ can be chosen in any way from $\binom{[n]}{m}$. We define the cost of this operation as $C_A$.

- *Random Client Selection Strategy* (R-CSS): The set $S_r$ is sampled uniformly at random from $\binom{[n]}{m}$. We define the cost of this operation as $C_R$.

- *Delegated Client Selection Strategy* (D-CSS): The set $S_r$ is chosen to be $S_D$, where $S_D$ is a fixed set of so-called delegate clients (to be discussed later) with $|S_D| \leq m$. We define the cost of this operation as 1.

Clearly, A-CSS is the most powerful among the three strategies, as the other two could be easily implemented in terms of A-CSS. Further, this strategy allows the server to collect information from any subset of clients. This

flexibility enables the implementation of *full synchronization*, where the algorithm needs certain information to be collected from all clients. This feature appears in many algorithms, the most basic example being the usual gradient descent (GD). Specifically, if an algorithm requires computing the full gradient $\nabla f(\mathbf{x})$ at a point $\mathbf{x}$, the server can split $[n]$ into $m$ disjoint sets and repeatedly use A-CSS to make $\lceil \frac{n}{m} \rceil$ sequential communications with each set of clients, sending to each client the point $\mathbf{x}$ and asking it to compute and return the gradient $\nabla f_i(\mathbf{x})$ (this corresponds to the simplest one-step local first-order method $\mathcal{M}_i^r$).

However, when clients are unreliable or slow to respond, using A-CSS can become costly. In cross-device settings, it is often more efficient to communicate only with a randomly sampled subset of clients at each communication round—a strategy commonly known as partial client participation (McMahan et al., 2017), which is modeled by the R-CSS. Therefore, we treat R-CSS as a cheaper strategy compared with A-CSS. Unlike A-CSS, the full-gradient computation cannot be directly implemented with R-CSS. (But it can be recovered with high probability by using R-CSS multiple times (Arjevani et al., 2020).)

In addition to the previous two strategies, there are scenarios where there exist so-called *delegate* clients that are always reliable and efficient both in communication and performing local computations. Sometimes, it is sufficient—or even preferable—to interact with these clients. With D-CSS, the server can always query information about the specific functions in the delegate set. In this work, we focus on the setting where there is one delegate client (number 1), i.e., $S_D = \{1\}$.

Based on the properties of each strategy discussed above, we assume that the above costs satisfy the following natural relations:

$$\boxed{1 \leq C_R \leq C_A}\,.$$

**Communication-and Local Complexities.** Consider a federated optimization algorithm $\mathcal{A}$ for solving a problem $f$ from the problem class $\mathcal{F}$. Let $R$ be the (possibly random) number of communication rounds made by $\mathcal{A}$ on $f$ and let $\hat{\mathbf{x}}^R$ be the corresponding output of $\mathcal{A}$. We define the *accuracy* of the algorithm $\mathcal{A}$ at the problem $f$ as:

$$\boxed{\mathrm{Accur}(\mathcal{A}, f) = \mathbb{E}[\|\nabla f(\hat{\mathbf{x}}^R)\|^2]}\,.$$

Further, let $N_A$, $N_R$, and $N_D$ be the (possibly random) total number of times that the client-selection strategies A-CSS, R-CSS, and D-CSS are used by $\mathcal{A}$ during the $R$ communication rounds, respectively. We define the *communication complexity* of $\mathcal{A}$ on $f$ as:

$$\boxed{N_f = \mathbb{E}[C_A N_A + C_R N_R + N_D]}\,,$$

and the *local complexity* of $\mathcal{A}$ on $f$ as:

$$\boxed{K_f = \mathbb{E}\left[\sum_{r=0}^{R-1} K_r\right]}\,,$$

where $K_r := \max_{i \in S_r} K_i^r$ and $K_i^r \geq 0$ is the number of queries to the oracle $O_{f_i}$ by the client $i$ at round $r$. We next define the worst-case complexities of $\mathcal{A}$ over the entire problem class $\mathcal{F}$. The communication complexity of $\mathcal{A}$ for solving the problem class $\mathcal{F}$ up to $\varepsilon$ accuracy is defined as:

$$\boxed{N_{\mathcal{F}}(\varepsilon) = \sup_{f \in \mathcal{F}}\big\{N_f \,|\, \mathrm{Accur}(\mathcal{A}, f) \leq \varepsilon^2\big\}}\,,$$

and the corresponding local complexity is defined as:

$$\boxed{K_{\mathcal{F}}(\varepsilon) = \sup_{f \in \mathcal{F}}\big\{K_f \,|\, \mathrm{Accur}(\mathcal{A}, f) \leq \varepsilon^2\big\}}\,.$$

If there exists some $f \in \mathcal{F}$ such that $\mathcal{A}$ fails to reach $\mathrm{Accur}(\mathcal{A}, f) \leq \varepsilon^2$, then both complexities $N_{\mathcal{F}}(\varepsilon)$ and $K_{\mathcal{F}}(\varepsilon)$ are defined as $+\infty$.

After fixing the desired accuracy $\varepsilon$, we consider only federated optimization algorithms that can achieve $\mathrm{Accur}(\mathcal{A}, f) \leq \varepsilon^2$ for all $f \in \mathcal{F}$. Among these algorithms, we say that the one with smaller communication complexity $N_{\mathcal{F}}(\varepsilon)$ is more efficient. If two algorithms have the same communication complexity, the one with lower local complexity $K_{\mathcal{F}}(\varepsilon)$ is preferable.

# D  RELATED WORK

## D.1  FORMALIZATION OF FEDERATED OPTIMIZATION ALGORITHMS AND THEIR COMPLEXITY

Several prior works have proposed oracle models and complexity metrics for distributed and federated optimization. These works typically consider solving the same problem as in (1), where each of the $n$ clients or

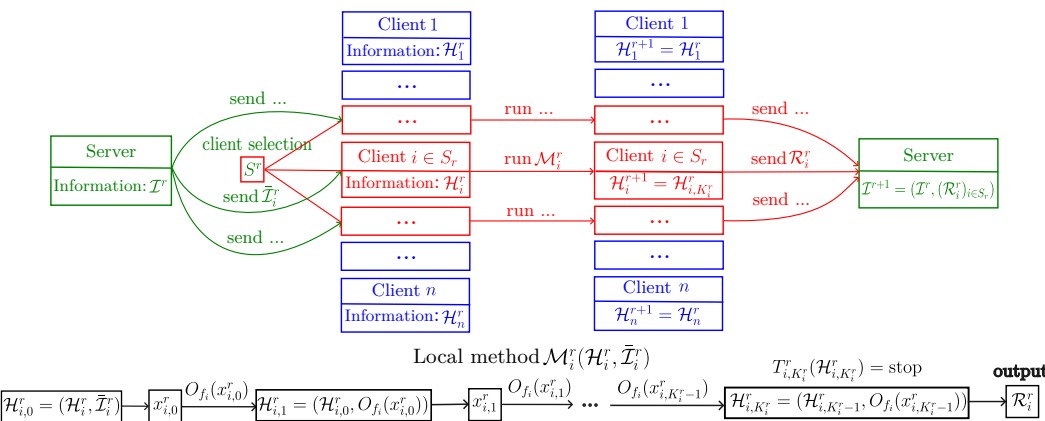

Figure D.1: Illustration of the sequence of procedures performed by a centralized distributed optimization algorithm at each communication round $r$.

workers only has access to its own local function. The primary distinctions among these models lie in how communication and computation are formalized. One line of work focuses on the settings where each worker can compute arbitrary information about its local objective, but only a limited number of bits is allowed to transmitted during each communication round (Braverman et al., 2016; Garg et al., 2014; Zhang et al., 2013). In this setting, complexity is often defined as the number of rounds required to reach a target accuracy. Alternative models remove this constraint and instead measure the total number of bits communicated over the entire optimization process (Korhonen & Alistarh, 2021). Other works impose structural restrictions on the communicated information, such as requiring exchanged vectors to lie in a certain subspace (e.g., linear combinations of local gradients) (Arjevani & Shamir, 2015; Lee et al., 2017).

The closest related model to ours is the Graph Oracle Model (GOM) (Woodworth et al., 2018; Patel et al., 2022; Woodworth, 2021). GOM introduces a computation and communication graph that determines how each device queries its local oracle and how the computed information propagates through the devices during optimization. Once the oracle and the graph structure are fixed, we obtain a specific model that allows to define the corresponding optimization algorithms. A commonly studied setting is the intermittent communication model, where $n$ devices work in parallel and synchronize after every $K$ local oracle queries. This setting becomes conceptually equivalent to our model when 1) all clients participate in every round, 2) the number of local oracle queries $K_r$ is uniformly bounded across all communication rounds, and 3) the server and the clients are allowed to send its entire accumulated information.

Beyond this scenario, there are several main differences between GOM and our proposed model. 1) GOM does not distinguish between different client-selection strategies that might have non-uniform associated costs. 2) Even when all the strategies have the same cost, GOM fixes the maximum number of local oracle queries for each round, whereas our model allows $K_r$ to vary across rounds. This flexibility enables modeling algorithms such as DANE (Shamir et al., 2014; Jiang et al., 2024a), which needs to solve each local subproblem sufficiently accurately, making $K_r$ dependent on the round $r$. 3) While partial client participation can be modeled in GOM by generating a random graph, the resulting algorithms are generally restricted to using pre-specified groups of clients at each round. This effectively enforces an offline client-selection strategy for GOM, since it may not account for the need to know the past responses of specific clients before deciding which clients to contact next. In contrast, our model fully supports online and adaptive client-selection strategies.

We believe that our model is reasonably simple and it appears to sufficiently capture how federated optimization algorithms work in practice. Even in the cases where our model is mathematically equivalent to existing ones, our model could still be more convenient to be used.

## D.2 COMPARISON WITH EXISTING FEDERATED OPTIMIZATION METHODS

**Notations in Table 1.** We denote $F^0 := f(\mathbf{x}^0) - f^\star$, $n_m := \frac{n}{m}$, $\delta_m^2 := \frac{n-m}{n-1}\frac{\delta^2}{m}$, $\zeta_m^2 := \frac{\zeta^2}{m}$, and $1 \lesssim C_R \lesssim C_A$ are the costs of communicating with a random set of $m$ clients and a specific set of $m$ clients, respectively.

In this section, we compare our proposed methods with several popular federated optimization algorithms in terms of their communication and local complexities (Section C). For simplicity and to ensure fair comparisons across algorithms, we assume that all methods use the deterministic first-order oracle locally, i.e., $O_{f_i} = O_{\mathrm{FO}_i}$, for all $i \in [n]$. We first state and discuss the assumptions under which each algorithm was analyzed in the literature. The abbreviations used in Table 1 are defined as follows.

**Assumption D.1** (FS (Function Smoothness)). There exists $L_f > 0$ such that for any $\mathbf{x}, \mathbf{y} \in \mathbb{R}^d$, we have: $\|\nabla f(\mathbf{x}) - \nabla f(\mathbf{y})\| \leq L_f \|\mathbf{x} - \mathbf{y}\|$.

**Assumption D.2** (IS (Individual Smoothness)). There exists $L_{\max} > 0$ such that for any $\mathbf{x}, \mathbf{y} \in \mathbb{R}^d$ and any $i \in [n]$, we have: $\|\nabla f_i(\mathbf{x}) - \nabla f_i(\mathbf{y})\| \leq L_{\max} \|\mathbf{x} - \mathbf{y}\|$.

**Assumption D.3** (SD (Second-order Dissimilarity) (Karimireddy et al., 2020; Jiang et al., 2024a; Gao et al., 2025)). There exists $\Delta_{\max} > 0$ such that for any $\mathbf{x}, \mathbf{y} \in \mathbb{R}^d$ and any $i \in [n]$, we have: $\|\nabla h_i(\mathbf{x}) - \nabla h_i(\mathbf{y})\| \leq \Delta_{\max} \|\mathbf{x} - \mathbf{y}\|$.

**Assumption D.4** (BGD (Bounded Gradient Dissimilarity)). There exists $\zeta > 0$ such that for any $\mathbf{x} \in \mathbb{R}^d$, we have: $\frac{1}{n} \sum_{i=1}^{n} \|\nabla f_i(\mathbf{x}) - \nabla f_i(\mathbf{y})\|^2 \leq \zeta^2$.

Note that the problem class of IS belong to SD, and SD implies Assumption 2.1 and 2.2. Moreover, any functions that satisfy Assumption 2.1 and 2.3 belong to the class of FS. Finally, the class of FS partially overlaps with the problem class defined by Assumptions 2.1 and 2.2. In Table 1, the assumption under which Centralized GD is analyzed is the most general one, and those for I-CGM-RG are the second most general. Finally, the smoothness constants satisfy the following relations:

$$\delta, \Delta_1 \lesssim \Delta_{\max} \lesssim L_{\max}, \quad L_1, L_f \lesssim L_{\max}.$$

We next briefly describe each method in Table 1 and discuss how the communication and local complexities are computed for each method. There are two operations that are commonly used in these methods. We describe them here to avoid repetition later. Denote $n_m := n/m$. The first operation is to compute the full gradient $\nabla f$ at the server at a certain point $\mathbf{x}$. As discussed in Section C, this can be implemented with $\lceil n_m \rceil$ successive communication rounds, each involving the use of A-CSS and one local gradient computation. Each such operation adds therefore $N_{\nabla f} := C_A \lceil n_m \rceil$ to the total communication complexity and $K_{\nabla f} := \lceil n_m \rceil$ to the total local complexity of an algorithm.

Another commonly used operation is to compute several mini-batch gradients $\nabla f_S$ at $b \geq 1$ points where $S \in \binom{[n]}{m}$ is sampled uniformly at random. This requires one communication round using R-CSS. The communication complexity of this operation is $N_{\nabla f_S, b} := C_R$ and the local complexity is $K_{\nabla f_S, b} := b$.

In what follows, we omit the subscript $\mathcal{F}$ in the notation for the complexities $N_{\mathcal{F}}(\varepsilon)$ and $K_{\mathcal{F}}(\varepsilon)$; the corresponding problem class is specified in Table 1 for each method.

**Centralized GD**. The method iterates:

$$\mathbf{x}^{t+1} = \mathbf{x}^t - \frac{1}{L_f} \nabla f(\mathbf{x}^t).$$

The iteration complexity of GD is $T = \mathcal{O}(\frac{L_f F^0}{\varepsilon^2})$ (Nesterov, 2018), implying that the communication complexity is $N(\varepsilon) = T N_{\nabla f} = \mathcal{O}(C_A n_m \frac{L_f F^0}{\varepsilon^2})$ and the local complexity is $K(\varepsilon) = T K_{\nabla f} = \mathcal{O}(n_m \frac{L_f F^0}{\varepsilon^2})$.

**FEDRED** (Jiang et al., 2024a). We consider FEDRED-GD, which initializes $\tilde{\mathbf{x}}_0 = \mathbf{x}_0$ and iterates:

$$\mathbf{x}_{t+1} = \arg\min_{\mathbf{x} \in \mathbb{R}^d} \left\{ f_1(\mathbf{x}_t) + \langle \nabla f_1(\mathbf{x}_t) + \nabla f(\tilde{\mathbf{x}}_t) - \nabla f_1(\tilde{\mathbf{x}}_t), \mathbf{x} \rangle + \frac{\eta}{2} \|\mathbf{x} - \mathbf{x}_t\|^2 + \frac{\lambda}{2} \|\mathbf{x} - \tilde{\mathbf{x}}_t\|^2 \right\},$$

where $\tilde{\mathbf{x}}_{t+1} = \mathbf{x}_{t+1}$ w.p. $p$ and $\tilde{\mathbf{x}}_{t+1} = \tilde{\mathbf{x}}_t$ w.p. $1 - p$. The solution of the subproblem can be computed in a closed-form. For $p \simeq \frac{\Delta_1}{L_1 + \Delta_1}$, $\eta \simeq L_1$ and $\lambda \simeq \Delta_1$, the iteration complexity of the method is $T = \mathcal{O}(\frac{L_1 F^0}{\varepsilon^2})$ (Jiang et al., 2024a). In expectation, once every $1/p$ iterations, the server computes the full gradient $\nabla f(\tilde{\mathbf{x}}_t) = \nabla f(\mathbf{x}_t)$, which adds $N_{\nabla f}$ to the total communication complexity and $K_{\nabla f}$ to the total local complexity. Then the server makes another communication round with D-CSS, sends $\nabla f(\mathbf{x}_t)$ to client 1 which then performs $1/p$ local steps in expectation and sends the result back to the server. The expected number of times the full gradient $\nabla f(\tilde{\mathbf{x}}_t)$ is computed is $pT = \mathcal{O}(\frac{\Delta_1 F^0}{\varepsilon^2})$. The expected number of communication rounds where D-CSS is used is $\mathbb{E}[N_D] = pT$. Therefore, the communication complexity is

$$N(\varepsilon) = \mathbb{E}[C_A N_A + N_D] = \mathcal{O}(pT N_{\nabla f} + pT) = \mathcal{O}\left(C_A n_m \frac{\Delta_1 F^0}{\varepsilon^2}\right).$$

The local complexity is bounded by

$$K(\varepsilon) = \mathcal{O}(pT K_{\nabla f} + T) = \mathcal{O}(pT n_m + T) = \mathcal{O}\left(n_m \frac{\Delta_1 F^0}{\varepsilon^2} + \frac{L_1 F^0}{\varepsilon^2}\right).$$

**FEDAVG** (McMahan et al., 2017). At each communication round $r \geq 0$, the server uses R-CSS to select clients, sends $\mathbf{x}^r$ to each client $i \in S_r$, which then returns an approximation solution $\mathbf{x}_i^{r+1} \approx \arg\min_{\mathbf{x}} f_i(\mathbf{x})$ by

running local GD starting at $\mathbf{x}^r$ for $K_r$ steps. Then the next iterate is defined as $\mathbf{x}^{r+1} = \frac{1}{m}\sum_{i \in S_r}\mathbf{x}_i^{r+1}$. When using local-GD, the optimal number of local steps is of order 1 (Karimireddy et al., 2020). Therefore, the local complexity is of the same order as the iteration complexity $T = \mathcal{O}(\frac{\varsigma_m^2 F^0}{\varepsilon^4} + \frac{\sqrt{L_{\max}}\varsigma}{\varepsilon^3} + \frac{L_{\max}F^0}{\varepsilon^2})$ (Karimireddy et al., 2020), and the communication complexity is $C_R T$.

**MIMEMVR** (Karimireddy et al., 2021). At each iteration $t \geq 0$, the server first uses R-CSS to get a random client set $S_t$ and computes the mini-batch gradient $\nabla f_{S_t}(\mathbf{x}^t)$. Then the server uses A-CSS to establish communication with the same set of clients $S_t$, sends $\nabla f_{S_t}(\mathbf{x}^t)$ to them which then updates:

$$\mathbf{x}_i^{t+1} \approx \arg\min_{\mathbf{x} \in \mathbb{R}^d}\{f_i(\mathbf{x}) + \langle\nabla f_{S_t}(\mathbf{x}^t) - \nabla f_i(\mathbf{x}^t), \mathbf{x}\rangle\}$$

by running momentum-based first-order methods locally for $\Theta(\frac{L_{\max}}{\Delta_{\max}})$ steps. The next iterate is defined as: $\mathbf{x}^{t+1} = \frac{1}{m}\sum_{i \in S_t}\mathbf{x}_i^{t+1}$. The communication complexity is thus $N(\varepsilon) = \mathbb{E}[C_A N_A + C_R N_R] = (C_A + C_R)T$, where $T = \mathcal{O}(\frac{\varsigma_m^2 F^0}{\varepsilon^2} + \frac{\varsigma_m \Delta_{\max}F^0}{\varepsilon^3} + \frac{\Delta_{\max}F^0}{\varepsilon^2})$ is the iteration complexity (Karimireddy et al., 2021). The local complexity is $\mathcal{O}(\frac{L_{\max}}{\Delta_{\max}}T)$.

**CE-LGD** (Patel et al., 2022). The method initializes $\mathbf{x}^{-1} = \mathbf{x}^0$ and $\mathbf{v}^{-1} \in \mathbb{R}^d$. At each iteration $t \geq 0$, the server first uses R-CSS to select clients $S_t$, sends $\mathbf{x}^{t-1}$ and $\mathbf{x}^t$ to the client $i \in S_t$ which then computes $\nabla f_i(\mathbf{x}^t)$ and $\nabla f_i(\mathbf{x}^{t-1})$ and sends them back to the server. The server computes $\mathbf{v}^t = \nabla f_{S_t}(\mathbf{x}^t) + (1 - \rho)(\mathbf{v}^{t-1} - \nabla f_{S_t}(\mathbf{x}^{t-1}))$ where $\rho \in [0, 1]$. Then the server uses R-CSS again to communicate with a random client. The client returns $\mathbf{x}^{t+1}$ by running the local SARAH method using $\mathbf{v}^t$ for $\Theta(\frac{L_{\max}}{\Delta_{\max}})$ local steps. The iteration complexity is $T = \mathcal{O}(\frac{\varsigma_m^2 F^0}{\varepsilon^2} + \frac{\varsigma_m \Delta_{\max}F^0}{\sqrt{m}\varepsilon^3} + \frac{\Delta_{\max}F^0}{\varepsilon^2})$ (Patel et al., 2022). The communication complexity is $N(\varepsilon) = \mathcal{O}(C_R T)$ and the local complexity is $K(\varepsilon) = \mathcal{O}(\frac{L_{\max}}{\Delta_{\max}}T)$.

**SCAFFOLD** (Karimireddy et al., 2020). At the beginning, each client $i = 1, \ldots, n$ computes $\mathbf{b}_i^0 = \nabla f_i(\mathbf{x}^0)$ and sends the result to the server; the server then computes $\mathbf{b}^0 = \nabla f(\mathbf{x}^0)$, which adds $N_{\nabla f}$ and $K_{\nabla f}$ to the total communication and local complexities, respectively. At each iteration $t \geq 0$, the server uses R-CSS to generate the client set $S_t$ and sends $\mathbf{x}^t$ to each client $i \in S_t$, which then computes $\mathbf{b}_i^t = \nabla f_i(\mathbf{x}^t)$ and sends $\mathbf{b}_i^t - \mathbf{b}_i^{t-1}$ back to the server. The server then updates $\mathbf{b}^t$ (SAG (Schmidt et al., 2017)) according to (10) (for $t \geq 1$). Then the server uses A-CSS to contact the clients in $S_t$ again and sends $\mathbf{b}^t$ to them. Each client $i \in S_t$ computes

$$\mathbf{x}_i^{t+1} \approx \arg\min_{\mathbf{x} \in \mathbb{R}^d}\{f_i(\mathbf{x}) + \langle\mathbf{b}^t - \nabla f_i(\mathbf{x}^t), \mathbf{x}\rangle\}$$

by running local GD for $K \simeq 1$ steps and sends the result back to the server. The server computes the next iterate as $\mathbf{x}^{t+1} = \frac{1}{m}\sum_{i \in S_t}\mathbf{x}_i^{t+1}$. The iteration complexity is $T = \mathcal{O}((\frac{n}{m})^{\frac{2}{3}}\frac{L_{\max}F^0}{\varepsilon^2})$ (Karimireddy et al., 2020). The communication complexity $N(\varepsilon) = \mathcal{O}(C_A\lceil n_m\rceil + (C_A + C_R)T)$. The local complexity $K(\varepsilon) = \lceil n_m\rceil + (1 + K)T = \mathcal{O}(n_m + T)$.

The final three algorithms do not strictly satisfy our definition of an algorithm because they do not clearly specify when to terminate the local method. However, we still present the conceptual methods and their communication complexity estimates, assuming (rather informally) that certain "local" operations can be implemented by running a certain local method for a sufficiently long time.

**FEDDYN** (Acar et al., 2021). During initialization, the server needs to collect $\mathbf{x}_i^0$ that satisfies $\nabla f_i(\mathbf{x}_i^0) = 0$ from all clients. Therefore, the communication complexity of this operation is $C_A\lceil n_m\rceil$. At each communication round $r \geq 0$, the server uses R-CSS to select clients $S_r$ and sends $\mathbf{x}^r$ to each client $i \in S_r$ which then sends

$$\mathbf{x}_i^{r+1} = \arg\min_{\mathbf{x}}\{f_i(\mathbf{x}) - \langle\nabla f_i(\mathbf{x}_{i,r}), \mathbf{x}\rangle + \frac{\lambda}{2}\|\mathbf{x} - \mathbf{x}^r\|^2\}, \quad i \in S_r,$$

back to the server. For $i \notin S_r$, $\mathbf{x}_i^{r+1} = \mathbf{x}_i^r$. Then the next iterate is updated as:

$$\mathbf{x}^{r+1} = \frac{1}{m}\sum_{i \in S_r}\mathbf{x}_i^{r+1} - \frac{1}{\lambda}\mathbf{h}^{r+1}, \quad \mathbf{h}^{r+1} = \mathbf{h}^r - \lambda\frac{1}{n}(\sum_{i \in S_r}\mathbf{x}_i^{r+1} - \mathbf{x}^r).$$

The iteration complexity is $T = \mathcal{O}(n_m\frac{L_{\max}F^0}{\varepsilon^2})$ (Acar et al., 2021) and the communication complexity is $N(\varepsilon) = \mathcal{O}(C_A n_m + C_R T)$.

**SABER-FULL** (Mishchenko et al., 2024). The method initializes $\mathbf{x}^{-1} = \mathbf{x}^0$ and $\mathbf{v}^{-1} = \mathbf{v}^0 = \nabla f(\mathbf{x}^0)$. At each iteration $t \geq 0$, w.p. $\frac{1}{n_m}$, the server updates $\mathbf{v}^t = \nabla f(\mathbf{x}^t)$ which adds $N_{\nabla f}$ and $K_{\nabla f}$ to the total communication and local complexity, respectively. With probability $1 - \frac{1}{n_m}$, the server uses R-CSS, obtains the random set $S_t$, and computes two mini-batch gradients $\nabla f_{S_t}(\mathbf{x}^t)$ and $\nabla f_{S_t}(\mathbf{x}^{t-1})$. This operation adds $N_{\nabla f_{S_t}}^2$ and $K_{\nabla f_{S_t}}^2$ to the communication and local complexity, respectively. Then the server updates

$\mathbf{v}^t = \mathbf{v}^{t-1} + \nabla f_{S_t}(\mathbf{x}^t) - \nabla f_{S_t}(\mathbf{x}^{t-1})$. (The original method samples a single index $m_t$. Here we extend it to $S_t$.) After the computation of $\mathbf{v}^t$, the server uses R-CSS, samples a random index $\tilde{m}_t \in [n]$, and sends $\mathbf{v}^t$ and $\mathbf{x}^t$ to the client $\tilde{m}_t$, which then returns

$$\mathbf{x}^{t+1} \approx \arg\min_{\mathbf{x} \in \mathbb{R}^d}\Big\{ f_{\tilde{m}_t}(\mathbf{x}) + \big\langle \mathbf{v}^t - \nabla f_{\tilde{m}_t}(\mathbf{x}^t), \mathbf{x} \big\rangle + \frac{\lambda}{2}\left\| \mathbf{x} - \mathbf{x}^t \right\|^2 \Big\}$$

back to the server. The iteration complexity of the method is $T = \mathcal{O}(\frac{\sqrt{n_m}\Delta_{\max}F^0}{\varepsilon^2})$ if each subproblem is solved exactly. The total communication complexity is

$$N(\varepsilon) = \mathbb{E}[C_A N_A + C_R N_R] = \mathcal{O}\Big( C_A n_m + \frac{1}{n_m} C_A T n_m + (1 - \frac{1}{n_m}) C_R T + C_R T \Big) = \mathcal{O}(C_A n_m + C_A T) \,.$$

**SABER-PARTIAL** (Mishchenko et al., 2024). We refer to Algorithm 2 in the original paper as SABER-PARTIAL. By Theorem 3 in that paper, the best $p$ is 1. The algorithm initializes $\mathbf{v}^0 = \nabla f(\mathbf{x}^0)$, which adds $N_{\nabla f}$ and $K_{\nabla f}$ to the total communication and local complexities, respectively. At each iteration $t \geq 1$, the method also needs to compute a mini-batch gradient $\mathbf{v}^t = \frac{1}{s}\sum_{i \in S_t'}\nabla f_i(\mathbf{x}^t)$ where $S_t'$ is sampled uniformly at random with replacement and $|S_t'| = s$ where $s$ is a parameter of the method. Since we assume that the server can communicate with at most $m$ clients at each round, implementing this operation requires $\lceil s/m \rceil$ sequential communication rounds with R-CSS. After that, the method needs to choose another random set $S_t$ with $|S_t| = s$. This adds $C_R \lceil s/m \rceil$ to the total communication complexity. The server sends $\mathbf{x}^t$ and $\mathbf{v}^t$ to the client $i \in S_t$, which then computes

$$\mathbf{x}_i^{t+1} \approx \arg\min_{\mathbf{x} \in \mathbb{R}^d}\Big\{ f_i(\mathbf{x}) + \big\langle \mathbf{v}^t - \nabla f_i(\mathbf{x}^t), \mathbf{x} \big\rangle + \frac{\lambda}{2}\left\| \mathbf{x} - \mathbf{x}^t \right\|^2 \Big\} \,,$$

and sends the result back to the server. The next iterate is updated as $\mathbf{x}^{t+1} = \frac{1}{s}\sum_{i \in S_t}\mathbf{x}_i^{t+1}$. If $\frac{\varsigma^2}{\varepsilon^2} \lesssim n$ and $s$ is chosen as $\Theta(\frac{\varsigma^2}{\varepsilon^2})$, then the method can output an $\varepsilon$-approximate stationary point after $T = \mathcal{O}(\frac{\Delta_{\max}F^0}{\sqrt{p}\varepsilon^2})$ iterations if each subproblem is solved exactly. The communication complexity is $N(\varepsilon) = \mathbb{E}[C_R N_R + C_A N_A] = \mathcal{O}(C_R T \lceil \frac{\varsigma^2}{m\varepsilon^2} \rceil + C_A n_m)$.

**Discussions**. FEDDYN, SABER-FULL and SABER-PARTIAL do not strictly satisfy our definition of an algorithm since they do not precisely specify when to terminate the local methods. The problem class for which FEDAVG, MIMEMVR, CE-LGD, and SABER-PARTIAL are analyzed is the smallest among all methods. Specifically, in addition to IS, they also assume BGD, which can be restrictive and exclude simple quadratics. Among these four, MIMEMVR and CE-LGD improve upon FEDAVG and SABER-PARTIAL in terms of their dependence on the target accuracy $\varepsilon$. Except for FEDAVG, the remaining three methods replace the dependence on $L_{\max}$ with $\Delta_{\max}$ in the communication complexity. Furthermore, compared to MIMEMVR, CE-LGD achieves a better dependence on $m$ in the term involving $\varepsilon^{-3}$.

For the remaining methods, SCAFFOLD improves the dependence on $n$ from $n_m$ (as in FEDDYN) to $n_m^{2/3}$. SAVER-FULL further reduces the dependence on $n$ to $\sqrt{n_m}$ and simultaneously improves the smoothness dependence from $L_{\max}$ to $\Delta_{\max}$. I-CGM-RG-SVRG achieves a tighter bound of $C_R \Delta_1 + \sqrt{C_A C_R n_m}\delta_m \lesssim C_A \sqrt{n_m}\Delta_{\max}$. Finally, I-CGM-RG-SAGA improves the communication cost constant from $C_A$ to $C_R$, compared to I-CGM-RG-SVRG, while maintaining the same local complexity—the best among all existing methods.

*Remark* D.5. According to Arjevani et al. (2020), one may alternatively compute the full gradient using only R-CSS. Let $m = 1$ for simplicity. Lemma 2 in (Arjevani et al., 2020) shows that w.p. $1 - \delta$, we can recover the full gradient $\nabla f(\mathbf{x})$ at a given point $\mathbf{x}$ by making $2n^2 \log(\frac{2n}{\delta})$ communication rounds with R-CSS. This can be helpful when the cost $C_A$ is extremely large. Indeed, the current communication complexity of I-CGM-RG-SAGA is of order $C_A n + C_R((\Delta_1 + \sqrt{n}\delta)F^0/\varepsilon^2)$. As soon as $C_A \gtrsim C_R((\Delta_1 + \sqrt{n}\delta)F^0/\varepsilon^2)/n$, the complexity is dominated by $C_A n$. This term arises from $2n$ sequential communication rounds with A-CSS for computing the full gradients. Now if we replace these operations with $4n^2 \log(\frac{2n}{\delta})$ sequential rounds with R-CSS, the total complexity might be reduced to $C_R(n^2 \log(n) + (\Delta_1 + \sqrt{n}\delta)F^0/\varepsilon^2)$. We leave a full theoretical development of this direction as interesting future work.

# E   TECHNICAL PRELIMINARIES

We frequently use the following lemmas for the proofs.

**Lemma E.1.** *For any* $\mathbf{x}, \mathbf{y} \in \mathbb{R}^d$ *and any* $\gamma > 0$, *we have:*

$$|\langle \mathbf{x}, \mathbf{y} \rangle| \leq \frac{\gamma}{2}\left\| \mathbf{x} \right\|^2 + \frac{1}{2\gamma}\left\| \mathbf{y} \right\|^2 , \tag{E.1}$$

$$\left\| \mathbf{x} + \mathbf{y} \right\|^2 \leq (1 + \gamma)\left\| \mathbf{x} \right\|^2 + \left(1 + \frac{1}{\gamma}\right)\left\| \mathbf{y} \right\|^2 . \tag{E.2}$$

**Lemma E.2** ((Jiang et al., 2024b), Lemma 13). *Let $\{\mathbf{g}_i\}_{i=1}^n$ be vectors in $\mathbb{R}^d$ with $n \geq 2$. Let $m \in [n]$ and let $S \in \binom{[n]}{m}$ be sampled uniformly at random without replacement. Let $\bar{\mathbf{g}} := \frac{1}{n} \sum_{i=1}^n \mathbf{g}_i$, $\sigma^2 := \frac{1}{n} \sum_{i=1}^n \|\mathbf{g}_i - \bar{\mathbf{g}}\|^2$, and $\bar{\mathbf{g}}_S := \frac{1}{m} \sum_{j \in S} \mathbf{g}_j$. Then,*

$$\mathbb{E}_S[\bar{\mathbf{g}}_S] = \bar{\mathbf{g}} \qquad and \qquad \mathbb{E}_S[\|\bar{\mathbf{g}}_S - \bar{\mathbf{g}}\|^2] = \frac{n-m}{n-1} \frac{\sigma^2}{m}. \tag{E.3}$$

**Lemma E.3** ((Allen-Zhu, 2018a), Fact 2.3). *Let $A_0, A_1, \dots$ be reals and let $K \sim \mathrm{Geom}(p)$ with $p \in (0, 1]$, that is $\mathbb{P}(K = k) = (1-p)^k p$ for each $k \in \{0, 1, 2, \dots\}$. Then it holds that: $\mathbb{E}[K] = \frac{1}{p} - 1$ and*

$$\mathbb{E}[A_K] = (1-p)\,\mathbb{E}[A_{K+1}] + pA_0. \tag{E.4}$$

*Proof.* Using the identity $\sum_{k \geq 0} k q^k = \frac{q}{(1-q)^2}$ for any $|q| < 1$, we have:

$$\mathbb{E}[K] = p \sum_{k \geq 0} k(1-p)^k = p \frac{1-p}{p^2} = \frac{1}{p} - 1.$$

To prove the second part, using the definition of $K$,

$$\mathbb{E}[A_{K+1}] = p \sum_{k \geq 0} A_{k+1}(1-p)^k = \frac{p}{1-p} \sum_{k \geq 1} A_k (1-p)^k = \frac{1}{1-p}(\mathbb{E}[A_K] - pA_0).$$

Rearranging gives the claim. $\qquad\square$

**Lemma E.4.** *Let $(A_t)_{t=0}^\infty$, $(B_t)_{t=0}^\infty$ and be two non-negative sequences such that*

$$A_{i+1} \leq (1-\alpha)A_i + B_i$$

*for any $i \geq 0$ with $\alpha \in (0, 1]$. Then for any $t \geq 1$,*

$$A_t \leq (1-\alpha)^t A_0 + \sum_{i=1}^t (1-\alpha)^{t-i} B_{i-1},$$

*and for any $T \geq 1$,*

$$\sum_{t=1}^T A_t \leq \frac{(1-\alpha)(1-(1-\alpha)^T)}{\alpha} A_0 + \sum_{t=0}^{T-1} \frac{1-(1-\alpha)^{T-t}}{\alpha} B_t \leq \frac{1-\alpha}{\alpha} A_0 + \frac{1}{\alpha} \sum_{t=0}^{T-1} B_t.$$

*Proof.* When $\alpha = 1$, the claim clearly holds. Let $0 < \alpha < 1$. Dividing both sides of the main recurrence by $(1-\alpha)^{i+1}$, we have for any $i \geq 0$:

$$\frac{A_{i+1}}{(1-\alpha)^{i+1}} \leq \frac{A_i}{(1-\alpha)^i} + \frac{B_i}{(1-\alpha)^{i+1}}.$$

Summing up from $i = 0$ to $i = t-1$, we get, for any $t \geq 1$:

$$\frac{A_t}{(1-\alpha)^t} \leq A_0 + \sum_{i=0}^{t-1} \frac{B_i}{(1-\alpha)^{i+1}} = A_0 + \sum_{i=1}^t \frac{B_{i-1}}{(1-\alpha)^i}.$$

This proves the first claim. To prove the second part, we sum up the first claim from $t = 1$ to $t = T$,

$$\sum_{t=1}^T A_t \leq \sum_{t=1}^T (1-\alpha)^t A_0 + \sum_{t=1}^T \sum_{i=1}^t (1-\alpha)^{t-i} B_{i-1}$$

$$= \frac{(1-\alpha)(1-(1-\alpha)^T)}{\alpha} A_0 + \sum_{i=1}^T \sum_{t=i}^T (1-\alpha)^{t-i} B_{i-1}$$

$$= \frac{(1-\alpha)(1-(1-\alpha)^T)}{\alpha} A_0 + \sum_{i=1}^T \frac{1-(1-\alpha)^{T-i+1}}{\alpha} B_{i-1}$$

$$= \frac{(1-\alpha)(1-(1-\alpha)^T)}{\alpha} A_0 + \sum_{t=0}^{T-1} \frac{1-(1-\alpha)^{T-t}}{\alpha} B_t. \qquad\square$$

**Lemma E.5.** *Let $p \in (0, 1)$. The minimizer of the problem $\min_{\gamma \in (0,1)} \{f(\gamma) := \frac{1-\gamma p}{\gamma(1-\gamma)}\}$ is attained at $\gamma^\star = \frac{1-\sqrt{1-p}}{p}$.*

*Proof.* Differentiate $f(\gamma)$, we have $f'(\gamma) = \frac{-1+2\gamma-p\gamma^2}{\gamma^2(1-\gamma)^2}$. Setting $f'(\gamma) = 0$ with $\gamma \in (0, 1)$ gives $\gamma^\star = \frac{1-\sqrt{1-p}}{p}$. Since $f(\gamma) \to \infty$ as $\gamma \to 0^+$ or $\gamma \to 1^-$, the critical point $\gamma^\star$ is the minimizer over $(0, 1)$. $\qquad\square$

# F  PROOFS FOR I-CGM

**Lemma F.1.** *Let I-CGM be applied to Problem* (1) *under Assumption 2.1. Let* $\lambda > \Delta_1$. *Then for any* $t \geq 0$,

$$\left\|\nabla f(\mathbf{x}^{t+1})\right\| \leq (\lambda + \Delta_1)\hat{\chi}_{t+1} + \hat{\Sigma}_t + e_t \ ,$$

*where* $\hat{\chi}_t := \|\mathbf{x}^t - \mathbf{x}^{t-1}\|$. *For any* $\mathbf{x} \in \mathbb{R}^d$, *we have:*

$$F_t(\mathbf{x}^t) - F_t(\mathbf{x}) \leq f(\mathbf{x}^t) - f(\mathbf{x}) + \frac{\hat{\Sigma}_t^2}{2(\lambda - \Delta_1)} \ .$$

*Suppose the iterates satisfy* (5)*, then the function value decreases as:*

$$f(\mathbf{x}^{t+1}) \leq f(\mathbf{x}^t) - \frac{\lambda - \Delta_1}{4}\hat{\chi}_{t+1}^2 + \frac{\hat{\Sigma}_t^2}{\lambda - \Delta_1} \ .$$

*Proof.* By the definition of $F_t$, we have:

$$\nabla F_t(\mathbf{x}^{t+1}) = \nabla f_1(\mathbf{x}^{t+1}) + \mathbf{g}^t - \nabla f_1(\mathbf{x}^t) + \lambda(\mathbf{x}^{t+1} - \mathbf{x}^t)$$
$$= \nabla f(\mathbf{x}^{t+1}) + \left(\mathbf{g}^t - \nabla f(\mathbf{x}^t)\right) + \left(\nabla h_1(\mathbf{x}^t) - \nabla h_1(\mathbf{x}^{t+1})\right) + \lambda(\mathbf{x}^{t+1} - \mathbf{x}^t) \ .$$

It follows that:

$$\left\|\nabla f(\mathbf{x}^{t+1})\right\| \leq \lambda\hat{\chi}_{t+1} + \|\nabla h_1(\mathbf{x}^t) - \nabla h_1(\mathbf{x}^{t+1})\| + \hat{\Sigma}_t + e_t$$
$$\overset{(2)}{\leq} (\lambda + \Delta_1)\hat{\chi}_{t+1} + \hat{\Sigma}_t + e_t \ ,$$

which proves the first inequality. Using the definition of $F_t$, for any $\mathbf{x} \in \mathbb{R}^d$, we have:

$$F_t(\mathbf{x}^t) - F_t(\mathbf{x})$$
$$= f_1(\mathbf{x}^t) + h_1(\mathbf{x}^t) - f_1(\mathbf{x}) - h_1(\mathbf{x}^t) - \langle \mathbf{g}^t - \nabla f_1(\mathbf{x}^t), \mathbf{x} - \mathbf{x}^t \rangle - \frac{\lambda}{2}\|\mathbf{x} - \mathbf{x}^t\|^2$$
$$= f(\mathbf{x}^t) - f(\mathbf{x}) + f(\mathbf{x}) - f_1(\mathbf{x}) - h_1(\mathbf{x}^t) - \langle \mathbf{g}^t - \nabla f_1(\mathbf{x}^t), \mathbf{x} - \mathbf{x}^t \rangle - \frac{\lambda}{2}\|\mathbf{x} - \mathbf{x}^t\|^2$$
$$= f(\mathbf{x}^t) - f(\mathbf{x}) + \left(h_1(\mathbf{x}) - h_1(\mathbf{x}^t) - \langle \nabla h_1(\mathbf{x}^t), \mathbf{x} - \mathbf{x}^t \rangle\right) - \frac{\lambda}{2}\|\mathbf{x} - \mathbf{x}^t\|^2 - \langle \mathbf{g}^t - \nabla f(\mathbf{x}^t), \mathbf{x} - \mathbf{x}^t \rangle$$
$$\overset{(2.1)}{\leq} f(\mathbf{x}^t) - f(\mathbf{x}) - \frac{\lambda - \Delta_1}{2}\|\mathbf{x} - \mathbf{x}^t\|^2 - \langle \mathbf{g}^t - \nabla f(\mathbf{x}^t), \mathbf{x} - \mathbf{x}^t \rangle \ .$$

Using (E.1), we can bound the last two terms by: $\frac{\|\mathbf{g}^t - \nabla f(\mathbf{x}^t)\|^2}{2(\lambda - \Delta_1)}$, which proves the second claim. Substituting $\mathbf{x} = \mathbf{x}^{t+1}$ and using (5), we get:

$$f(\mathbf{x}^{t+1}) \leq f(\mathbf{x}^t) - \frac{\lambda - \Delta_1}{2}\hat{\chi}_{t+1}^2 - \langle \mathbf{g}^t - \nabla f(\mathbf{x}^t), \mathbf{x}^{t+1} - \mathbf{x}^t \rangle$$
$$\overset{(E.1)}{\leq} f(\mathbf{x}^t) - \frac{\lambda - \Delta_1}{4}\hat{\chi}_{t+1}^2 + \frac{\hat{\Sigma}_t^2}{\lambda - \Delta_1} \ . \qquad \square$$

## F.1  PROOF FOR THEOREM 3.1

*Proof.* Let $t \geq 0$. By Lemma F.1, we have:

$$\frac{\lambda - \Delta_1}{4}\hat{\chi}_{t+1}^2 \leq f(\mathbf{x}^t) - f(\mathbf{x}^{t+1}) + \frac{\hat{\Sigma}_t^2}{\lambda - \Delta_1} \ .$$

Using the first claim of Lemma F.1, we have:

$$\|\nabla f(\mathbf{x}^{t+1})\|^2 \leq \left((\lambda + \Delta_1)\hat{\chi}_{t+1} + \hat{\Sigma}_t + e_t\right)^2 \leq 2(\lambda + \Delta_1)^2\hat{\chi}_{t+1}^2 + 2(\hat{\Sigma}_t + e_t)^2 \ ,$$

Adding $(\lambda + \Delta_1)^2\hat{\chi}_{t+1}^2$ to both sides of this inequality and substituting the first display, we have:

$$\|\nabla f(\mathbf{x}^{t+1})\|^2 + (\lambda + \Delta_1)^2\hat{\chi}_{t+1}^2$$
$$\leq 3(\lambda + \Delta_1)^2\left(\frac{4}{\lambda - \Delta_1}\left(f(\mathbf{x}^t) - f(\mathbf{x}^{t+1})\right) + \frac{4}{\lambda - \Delta_1}\frac{\hat{\Sigma}_t^2}{\lambda - \Delta_1}\right) + 2(\hat{\Sigma}_t + e_t)^2$$
$$\leq \frac{12(\lambda + \Delta_1)^2}{\lambda - \Delta_1}\left(f(\mathbf{x}^t) - f(\mathbf{x}^{t+1})\right) + \left(\frac{12(\lambda + \Delta_1)^2}{(\lambda - \Delta_1)^2} + 4\right)\hat{\Sigma}_t^2 + 4e_t^2 \ .$$

Summing up from $t = 0$ to $T - 1$, we get the claim. $\qquad \square$

### F.2 PROOFS OF LOCAL CGM FOR SOLVING THE SUBPROBLEMS

**Lemma F.2** (Composite gradient method). *Consider the composite problem:*

$$\min_{\mathbf{x}\in\mathbb{R}^d}\big\{F(\mathbf{x}) := \phi(\mathbf{x}) + \psi(\mathbf{x})\big\} ,$$

*where $\phi$ is $L_\phi$-smooth and $\psi$ is $\lambda_\psi$-strongly convex and simple with $\lambda_\psi \geq 0$. Let $\eta = L_\phi$. Consider the composite gradient method:*

$$\mathbf{x}_{k+1} = \arg\min_{\mathbf{x}\in\mathbb{R}^d}\big\{L_k(\mathbf{x}) := \phi(\mathbf{x}_k) + \langle\nabla\phi(\mathbf{x}_k), \mathbf{x} - \mathbf{x}_k\rangle + \frac{\eta}{2}\|\mathbf{x} - \mathbf{x}_k\|^2 + \psi(\mathbf{x})\big\} .$$

*Then for any $k \geq 0$, $F(\mathbf{x}_{k+1}) \leq F(\mathbf{x}_k)$. For any $K \geq 1$, it holds that:*

$$\|\nabla F(\mathbf{x}_K^\star)\|^2 \leq \frac{8L_\phi^2\big[F(\mathbf{x}_0) - F(\mathbf{x}_K)\big]}{(L_\phi + \lambda_\phi)K} ,$$

*where $\mathbf{x}_K^\star = \arg\min_{(\mathbf{x}_k)_{k=1}^K}\|\nabla F(\mathbf{x}_k)\|$. Furthermore, if $\hat{K} \sim \mathrm{Geom}(p)$ with $p \in (0, 1]$, then we also have:*

$$\mathbb{E}_{\hat{K}}\big[\|\nabla F(\mathbf{x}_{\hat{K}+1})\|^2\big] \leq \frac{8L_\phi^2 p}{L_\phi + \lambda_\psi}\big[F(\mathbf{x}_0) - \mathbb{E}_{\hat{K}}[F(\mathbf{x}_{\hat{K}+1})]\big] .$$

*Proof.* Let $k \geq 0$. By $(L_\phi + \lambda_\psi)$-strong convexity of $L_k$, for any $\mathbf{x} \in \mathbb{R}^d$, we have,

$$L_k(\mathbf{x}) \geq L_k(\mathbf{x}_{k+1}) + \frac{L_\phi + \lambda_\psi}{2}\|\mathbf{x} - \mathbf{x}_{k+1}\|^2 .$$

Substituting $\mathbf{x} = \mathbf{x}_k$, it follows that,

$$F(\mathbf{x}_k) \geq \phi(\mathbf{x}_k) + \langle\nabla\phi(\mathbf{x}_k), \mathbf{x}_{k+1} - \mathbf{x}_k\rangle + \frac{L_\phi}{2}\|\mathbf{x}_{k+1} - \mathbf{x}_k\|^2 + \psi(\mathbf{x}_{k+1}) + \frac{L_\phi + \lambda_\psi}{2}\|\mathbf{x}_{k+1} - \mathbf{x}_k\|^2$$

$$\geq \phi(\mathbf{x}_{k+1}) + \psi(\mathbf{x}_{k+1}) + \frac{L_\phi + \lambda_\psi}{2}\|\mathbf{x}_{k+1} - \mathbf{x}_k\|^2 = F(\mathbf{x}_{k+1}) + \frac{L_\phi + \lambda_\psi}{2}\|\mathbf{x}_{k+1} - \mathbf{x}_k\|^2 .$$

This proves that the function value of $F$ monotinically decreases. By the definition of $\mathbf{x}_{k+1}$, we get:

$$\nabla\phi(\mathbf{x}_k) + L_\phi(\mathbf{x}_{k+1} - \mathbf{x}_k) + \nabla\psi(\mathbf{x}_{k+1}) = 0 .$$

It follows that:

$$\nabla F(\mathbf{x}_{k+1}) = \nabla\phi(\mathbf{x}_{k+1}) + \nabla\psi(\mathbf{x}_{k+1}) = \nabla\phi(\mathbf{x}_{k+1}) - \nabla\phi(\mathbf{x}_k) + L_\phi(\mathbf{x}_k - \mathbf{x}_{k+1}) ,$$

and hence,

$$\|\nabla F(\mathbf{x}_{k+1})\| \leq \|\nabla\phi(\mathbf{x}_{k+1}) - \nabla\phi(\mathbf{x}_k)\| + \eta\|\mathbf{x}_{k+1} - \mathbf{x}_k\| \leq 2L_\phi\|\mathbf{x}_{k+1} - \mathbf{x}_k\| .$$

Substituting this inequality into the second display, we get, for any $k \geq 0$:

$$\|\nabla F(\mathbf{x}_{k+1})\|^2 \leq \frac{8L_\phi^2}{L_\phi + \lambda_\psi}\big[F(\mathbf{x}_k) - F(\mathbf{x}_{k+1})\big] .$$

Summing up from $k = 0$ to $K - 1$, we have:

$$\sum_{k=1}^K\|\nabla F(\mathbf{x}_k)\|^2 \leq \frac{8L_\phi^2}{L_\phi + \lambda_\phi}\big[F(\mathbf{x}_0) - F(\mathbf{x}_K)\big] .$$

Dividing both sides by $K$, we get the first claim.

For the second claim, substituting $k = \hat{K}$ with $\hat{K} \sim \mathrm{Geom}(p)$ into the last second display, passing to the expectations and applying Lemma E.3, we have:

$$\mathbb{E}_{\hat{K}}[\|\nabla F(\mathbf{x}_{\hat{K}+1})\|^2] \leq \frac{8L_\phi^2}{L_\phi + \lambda_\psi}\mathbb{E}_{\hat{K}}[F(\mathbf{x}_{\hat{K}}) - F(\mathbf{x}_{\hat{K}+1})]$$

$$\leq \frac{8L_\phi^2}{L_\phi + \lambda_\psi}\big((1 - p)\mathbb{E}_{\hat{K}}[F(\mathbf{x}_{\hat{K}+1})] + pF(\mathbf{x}_0) - \mathbb{E}_{\hat{K}}[F(\mathbf{x}_{\hat{K}+1})]\big)$$

$$= \frac{8L_\phi^2 p}{L_\phi + \lambda_\psi}\big[F(\mathbf{x}_0) - \mathbb{E}_{\hat{K}}[F(\mathbf{x}_{\hat{K}+1})]\big] . \qquad \square$$

### F.2.1 PROOF OF LEMMA 3.3

*Proof.* Applying Lemma F.2 (with $\phi(\mathbf{x}) = f_1(\mathbf{x})$ and $\psi(\mathbf{x}) = \langle \mathbf{g}^t - \nabla f_1(\mathbf{x}^t), \mathbf{x} - \mathbf{x}^t \rangle + \frac{\lambda}{2} \left\| \mathbf{x} - \mathbf{x}^t \right\|^2$), we have for any $t \geq 0$:

$$e_t^2 = \left\| \nabla F_t(\mathbf{x}^{t+1}) \right\|^2 \leq \frac{8L_1^2(F_t(\mathbf{x}^t) - F_t^\star)}{(L_1 + \lambda)K} \,,$$

where $F_t^\star := \min_{\mathbf{x} \in \mathbb{R}^d} \{F_t(\mathbf{x})\}$. Applying Lemma F.1, we get:

$$F_t(\mathbf{x}^t) - F_t^\star \leq f(\mathbf{x}^t) - f^\star + \frac{\hat{\Sigma}_t^2}{2(\lambda - \Delta_1)} \,.$$

It follows that:

$$\sum_{t=0}^{T-1} e_t^2 \leq \frac{8L_1^2}{(L_1 + \lambda)K} \Big( \sum_{t=0}^{T-1} (f(\mathbf{x}^t) - f^\star) + \sum_{t=0}^{T-1} \frac{\hat{\Sigma}_t^2}{2(\lambda - \Delta_1)} \Big) \,.$$

We next upper bound $\sum_{t=0}^{T-1} (f(\mathbf{x}^t) - f^\star)$. Applying Lemma F.2, we have for any $i \geq 0$:

$$f(\mathbf{x}^{i+1}) \leq f(\mathbf{x}^i) + \frac{\hat{\Sigma}_i^2}{2(\lambda - \Delta_1)} \,.$$

Summing up from $i = 0$ to $i = t - 1$, we have:

$$f(\mathbf{x}^t) \leq f(\mathbf{x}^0) + \sum_{i=0}^{t-1} \frac{\hat{\Sigma}_i^2}{2(\lambda - \Delta_1)} \,.$$

Hence,

$$\sum_{t=0}^{T-1} (f(\mathbf{x}^t) - f^\star) \leq T(f(\mathbf{x}^0) - f^\star) + \sum_{t=0}^{T-1} \sum_{i=0}^{t-1} \frac{\hat{\Sigma}_i^2}{2(\lambda - \Delta_1)} \leq TF^0 + T \sum_{t=0}^{T-2} \frac{\hat{\Sigma}_t^2}{2(\lambda - \Delta_1)} \,.$$

It follows that:

$$\sum_{t=0}^{T-1} e_t^2 \leq \frac{8L_1^2}{(L_1 + \lambda)K} \Big( TF^0 + (T+1) \sum_{t=0}^{T-1} \frac{\hat{\Sigma}_t^2}{2(\lambda - \Delta_1)} \Big) \,.$$

To achieve the accuracy condition 6, by the choice of $K$, we have

$$\frac{8L_1^2 T}{(L_1 + \lambda)K} \leq \frac{8L_1 T}{K} \leq \lambda - \Delta_1 \leq \frac{(\lambda + \Delta_1)^2}{\lambda - \Delta_1} \,, \quad \text{and} \quad \frac{8L_1^2}{(L_1 + \lambda)K} \frac{(T+1)}{2(\lambda - \Delta_1)} \leq \frac{8L_1 T}{(\lambda - \Delta_1)K} \leq 1 \,.$$

Passing to the full expectation, we get the claim. □

### F.2.2 PROOF OF LEMMA H.2

*Proof.* Applying Lemma F.2 and F.1, we have for any $t \geq 0$:

$$\mathbb{E}_{\hat{K}_t}[e_t^2] \leq \frac{8L_1^2 p}{L_1 + \lambda} \mathbb{E}_{\hat{K}_t}[F_t(\mathbf{x}^t) - F_t(\mathbf{x}^{t+1})] \leq \frac{8L_1^2 p}{L_1 + \lambda} \Big( \mathbb{E}_{\hat{K}_t} \left[ f(\mathbf{x}^t) - f(\mathbf{x}^{t+1}) \right] + \frac{\hat{\Sigma}_t^2}{2(\lambda - \Delta_1)} \Big) \,.$$

Taking the full expectation and summing up from $t = 0$ to $t = T - 1$, we have:

$$\sum_{t=0}^{T-1} \mathbb{E}[e_t^2] \leq \frac{8L_1^2 p}{L_1 + \lambda} \Big( f(\mathbf{x}^0) - f^\star + \frac{1}{2(\lambda - \Delta_1)} \sum_{t=0}^{T-1} \Sigma_t^2 \Big) \,.$$

By the choice of $p$, it holds that:

$$\frac{8L_1^2 p}{L_1 + \lambda} = \frac{L_1^2(\lambda - \Delta_1)}{(L_1 + \lambda)^2} \leq \lambda - \Delta_1 \leq \frac{(\lambda + \Delta_1)^2}{\lambda - \Delta_1} \,,$$

and

$$\frac{4L_1^2 p}{(L_1 + \lambda)(\lambda - \Delta_1)} = \frac{L_1^2}{2(L_1 + \lambda)^2} < 1 \,.$$

Hence, accuracy condition (6) is satisfied. □

### F.3 PROOFS OF PROPERTIES OF THE SAGA AND SVRG ESTIMATORS

**Lemma F.3.** *Consider the SAGA estimator. Then for any $t \geq 2$, it holds that:*

$$\mathbf{b}^t = \mathbf{b}^{t-1} + \frac{1}{n_m}[\nabla f_{S_t}(\mathbf{x}^t) - \mathbf{b}_{S_t}^{t-1}] .$$

*Proof.* Indeed,

$$\mathbf{b}^t = \frac{1}{n}\sum_{i=1}^{n}\mathbf{b}_i^t = \frac{1}{n}\Big[\sum_{i\notin S_t}\mathbf{b}_i^{t-1} + \sum_{i\in S_t}\nabla f_i(\mathbf{x}^t)\Big] = \frac{1}{n}\Big[\sum_{i=1}^{n}\mathbf{b}_i^{t-1} + \sum_{i\in S_t}[\nabla f_i(\mathbf{x}^t) - \mathbf{b}_i^{t-1}]\Big]$$

$$= \mathbf{b}^{t-1} + \frac{1}{n_m}[\nabla f_{S_t}(\mathbf{x}^t) - \mathbf{b}_{S_t}^{t-1}] . \qquad \square$$

#### F.3.1 PROOF OF LEMMA 4.1

*Proof.* Let $t \geq 2$. By definition, $\mathbf{G}^t = \frac{1}{m}\sum_{i\in S_t}\mathbf{G}_i^t$, where $\mathbf{G}_i^t := \nabla f_i(\mathbf{x}^t) - \mathbf{b}_i^{t-1} + \mathbf{b}^{t-1}$ and $S_t$ is independent from $\mathbf{x}^t$ and $(\mathbf{b}_i^{t-1})_{i=1}^{n}$. Therefore, according to Lemma E.2, we have:

$$\mathbb{E}_{S_t}[\mathbf{G}^t] = \frac{1}{n}\sum_{i=1}^{n}\mathbf{G}_i^t = \nabla f(\mathbf{x}^t) \quad \text{and} \quad \mathbb{E}_{S_t}[\|\mathbf{G}^t - \nabla f(\mathbf{x}^t)\|^2] = \frac{n-m}{n-1}\frac{1}{m}\hat{\sigma}_{t,1}^2 ,$$

where $\hat{\sigma}_{t,1}^2 := \frac{1}{n}\sum_{i=1}^{n}\|(\nabla f_i(\mathbf{x}^t) - \mathbf{b}_i^{t-1}) - (\nabla f(\mathbf{x}^t) - \mathbf{b}^{t-1})\|^2$. Taking the expectation w.r.t. $S_{[t-1]}$ on both sides, we get:

$$\sigma_t^2 = \frac{n-m}{n-1}\frac{1}{m}\mathbb{E}_{S_{[t-1]}}[\hat{\sigma}_{t,1}^2] := \frac{q_m}{m}\sigma_{t,1}^2 .$$

We next derive the recurrence for $\sigma_{t,1}^2$. Denote $\hat{\chi}_t := \|\mathbf{x}^t - \mathbf{x}^{t-1}\|$. For any $\alpha > 0$, we obtain:

$$\hat{\sigma}_{t+1,1}^2 = \frac{1}{n}\sum_{i=1}^{n}\|(\nabla f_i(\mathbf{x}^{t+1}) - \mathbf{b}_i^t) - (\nabla f(\mathbf{x}^{t+1}) - \mathbf{b}^t)\|^2$$

$$= \frac{1}{n}\sum_{i=1}^{n}\big\|(\nabla f_i(\mathbf{x}^t) - \mathbf{b}_i^t) - (\nabla f(\mathbf{x}^t) - \mathbf{b}^t) + [\nabla h_i(\mathbf{x}^t) - \nabla h_i(\mathbf{x}^{t+1})]\big\|^2$$

$$\overset{(E.2),(3)}{\leq} (1+\alpha)\frac{1}{n}\sum_{i=1}^{n}\|(\nabla f_i(\mathbf{x}^t) - \mathbf{b}_i^t) - (\nabla f(\mathbf{x}^t) - \mathbf{b}^t)\|^2 + \Big(1+\frac{1}{\alpha}\Big)\delta^2\hat{\chi}_{t+1}^2$$

$$= (1+\alpha)\Big[\frac{1}{n}\sum_{i=1}^{n}\|\nabla f_i(\mathbf{x}^t) - \mathbf{b}_i^t - \nabla f(\mathbf{x}^t)\|^2 - \|\mathbf{b}^t\|^2\Big] + \Big(1+\frac{1}{\alpha}\Big)\delta^2\hat{\chi}_{t+1}^2$$

$$= (1+\alpha)\Big[\frac{1}{n_m}\|\nabla f(\mathbf{x}^t)\|^2 + \frac{1}{n}\sum_{i\notin S_t}\|\nabla f_i(\mathbf{x}^t) - \mathbf{b}_i^{t-1} - \nabla f(\mathbf{x}^t)\|^2 - \|\mathbf{b}^t\|^2\Big] + \Big(1+\frac{1}{\alpha}\Big)\delta^2\hat{\chi}_{t+1}^2 ,$$

where the last second equality follows from the identity $\frac{1}{n}\sum_{i=1}^{n}[\nabla f_i(\mathbf{x}^t) - \mathbf{b}_i^t - \nabla f(\mathbf{x}^t)] = \mathbf{b}^t$, and the last equality follows from the definition of $\mathbf{b}_i^t$. Further note that

$$\mathbb{E}_{S_t}\Big[\frac{1}{n}\sum_{i\notin S_t}\|\nabla f_i(\mathbf{x}^t) - \mathbf{b}_i^{t-1} - \nabla f(\mathbf{x}^t)\|^2\Big] = \frac{1}{n}\sum_{i=1}^{n}\mathbb{P}(i\notin S_t)\|\nabla f_i(\mathbf{x}^t) - \mathbf{b}_i^{t-1} - \nabla f(\mathbf{x}^t)\|^2$$

$$= \Big(1-\frac{1}{n_m}\Big)\frac{1}{n}\sum_{i=1}^{n}\|\nabla f_i(\mathbf{x}^t) - \mathbf{b}_i^{t-1} - \nabla f(\mathbf{x}^t)\|^2 = \Big(1-\frac{1}{n_m}\Big)[\hat{\sigma}_{t,1}^2 + \|\mathbf{b}^{t-1}\|^2]$$

Taking the expectation w.r.t. $S_t$ on both sides of the last second display and plugging in this identity, we obtain:

$$\mathbb{E}_{S_t}[\hat{\sigma}_{t+1,1}^2 + (1+\alpha)\|\mathbf{b}^t\|^2] \leq (1+\alpha)\Big(1-\frac{1}{n_m}\Big)[\hat{\sigma}_{t,1}^2 + \|\mathbf{b}^{t-1}\|^2] + (1+\alpha)\frac{1}{n_m}\|\nabla f(\mathbf{x}^t)\|^2$$

$$+ \Big(1+\frac{1}{\alpha}\Big)\delta^2\mathbb{E}_{S_t}[\hat{\chi}_{t+1}^2] .$$

Taking the expectation w.r.t. $S_{[t-1]}$ on both sides and denoting $\mathbf{B}_t^2 := (1+\alpha)\mathbb{E}_{S_{[t]}}[\|\mathbf{b}^t\|^2]$, we get:

$$\sigma_{t+1,1}^2 + \mathbf{B}_t^2 \leq (1+\alpha)\Big(1-\frac{1}{n_m}\Big)[\sigma_{t,1}^2 + \mathbf{B}_{t-1}^2] + \frac{1+\alpha}{n_m}G_t^2 + (1+\frac{1}{\alpha})\delta^2\chi_{t+1}^2 .$$

Let $1 - \gamma/n_m := (1 + \alpha)(1 - 1/n_m) \in (1 - 1/n_m, 1)$. We then have: $\gamma \in (0, 1)$, $1 + \alpha = \frac{n_m - \gamma}{n_m - 1}$ and $1 + \frac{1}{\alpha} = \frac{n_m - \gamma}{1 - \gamma}$. The previous display can thus be reformulated as:

$$\sigma_{t+1,1}^2 + \mathbf{B}_t^2 \leq \left(1 - \frac{\gamma}{n_m}\right)[\sigma_{t,1}^2 + \mathbf{B}_{t-1}^2] + \frac{n_m - \gamma}{n_m(n_m - 1)}G_t^2 + \frac{n_m - \gamma}{1 - \gamma}\delta^2\chi_{t+1}^2 .$$

Let $T \geq 3$. Applying Lemma E.4 (starting from $t = 2$), we have:

$$\sum_{t=3}^{T}[\sigma_{t,1}^2 + \mathbf{B}_{t-1}^2] \leq \frac{1 - \gamma/n_m}{\gamma/n_m}[\sigma_{2,1}^2 + \mathbf{B}_1^2] + \frac{1}{\gamma/n_m}\sum_{t=2}^{T-1}\left[\frac{n_m - \gamma}{n_m(n_m - 1)}G_t^2 + \frac{n_m - \gamma}{1 - \gamma}\delta^2\chi_{t+1}^2\right] .$$

Adding $\sigma_{2,1}^2$ to both sides and dropping the non-negative $\mathbf{B}_t^2$, we obtain:

$$\sum_{t=2}^{T}\sigma_{t,1}^2 \leq \frac{n_m}{\gamma}\sigma_{2,1}^2 + \frac{n_m - \gamma}{\gamma}\mathbf{B}_1^2 + \frac{n_m - \gamma}{\gamma(n_m - 1)}\sum_{t=2}^{T-1}G_t^2 + \frac{n_m(n_m - \gamma)}{\gamma(1 - \gamma)}\delta^2\sum_{t=2}^{T-1}\chi_{t+1}^2 .$$

Recall that $\mathbf{b}_i^1 = \nabla f_i(\mathbf{x}^1)$ for all $i \in [n]$. It holds that:

$$\sigma_{2,1}^2 = \hat{\sigma}_{2,1}^2 = \frac{1}{n}\sum_{i=1}^{n}[\|(\nabla f_i(\mathbf{x}^2) - \nabla f_i(\mathbf{x}^1)) - (\nabla f(\mathbf{x}^2) - \nabla f(\mathbf{x}^1))\|^2 \overset{(2.2)}{\leq} \delta^2\hat{\chi}_2^2 = \delta^2\chi_2^2, \quad \mathbf{B}_1^2 = \frac{n_m - \gamma}{n_m - 1}G_1^2 .$$

It follows that:

$$\sum_{t=2}^{T}\sigma_{t,1}^2 \leq \frac{(n_m - \gamma)^2}{\gamma(n_m - 1)}G_1^2 + \frac{n_m - \gamma}{\gamma(n_m - 1)}\sum_{t=2}^{T-1}G_t^2 + \frac{n_m(n_m - \gamma)}{\gamma(1 - \gamma)}\delta^2\sum_{t=1}^{T-1}\chi_{t+1}^2 .$$

Let us choose $\gamma$ which minimizes the coefficient in front of $\sum_{t=1}^{T-1}\chi_{t+1}^2$ over $(0, 1)$. By Lemma E.5, we get $\gamma^\star = n_m - \sqrt{n_m^2 - n_m}$. Substituting $\gamma = \gamma^\star$, we have:

$$\sum_{t=2}^{T}\sigma_{t,1}^2 \leq (\sqrt{n_m^2 - n_m} + n_m)G_1^2 + \left(1 + \frac{\sqrt{n_m}}{\sqrt{n_m - 1}}\right)\sum_{t=2}^{T-1}G_t^2 + n_m(\sqrt{n_m} + \sqrt{n_m - 1})^2\delta^2\sum_{t=1}^{T-1}\chi_{t+1}^2$$

$$\leq 2n_m G_1^2 + \left(1 + \frac{\sqrt{n_m}}{\sqrt{n_m - 1}}\right)\sum_{t=2}^{T-1}G_t^2 + 4n_m^2\delta^2\sum_{t=1}^{T-1}\chi_{t+1}^2 .$$

Multiplying both sides by $\frac{q_m}{m}$, substituting the identity $\sigma_t^2 = \frac{q_m}{m}\sigma_{t,1}^2$ and $\frac{q_m}{m}\left(1 + \frac{\sqrt{n_m}}{\sqrt{n_m - 1}}\right) = \frac{n - m}{n - 1}\frac{1}{m} + \frac{\sqrt{n - m}\sqrt{n}}{m(n - 1)}$, we obtain:

$$\sum_{t=2}^{T}\sigma_t^2 \leq \frac{2n_m q_m}{m}G_1^2 + \frac{n_m - 1 + \sqrt{n_m^2 - n_m}}{(n - 1)}\sum_{t=2}^{T-1}G_t^2 + 4n_m^2\delta_m^2\sum_{t=2}^{T}\chi_t^2 .$$

Adding $\sigma_0^2 = 0$ and $\sigma_1^2 = 0$ to both sides, we prove the variance bound for $T \geq 3$, since $\mathbf{G}^0 = \nabla f(\mathbf{x}^0)$ and $\mathbf{G}^1 = \nabla f(\mathbf{x}^1)$. The same inequality also holds for $T = 1$ and $T = 2$, since $\sigma_0^2 = \sigma_1^2 = 0$ and $\sigma_2^2 = \frac{q_m}{m}\sigma_{2,1}^2 \leq \frac{q_m}{m}\delta^2\chi_2^2$.

$\square$

### F.3.2 PROOF OF LEMMA 4.3

*Proof.* Let $t \geq 1$. By definition, $\mathbf{G}^t = \frac{1}{m}\sum_{i \in S_t}\mathbf{G}_i^t$, where $\mathbf{G}_i^t := \nabla f_i(\mathbf{x}^t) - \nabla f_i(\mathbf{w}^t) + \nabla f(\mathbf{w}^t)$ and $S_t$ is independent from $\mathbf{x}^t$ and $\mathbf{w}^t$. Therefore, according to Lemma E.2, we have:

$$\mathbb{E}_{S_t}[\mathbf{G}^t] = \frac{1}{n}\sum_{i=1}^{n}\mathbf{G}_i^t = \nabla f(\mathbf{x}^t) \quad \text{and} \quad \mathbb{E}_{S_t}[\|\mathbf{G}^t - \nabla f(\mathbf{x}^t)\|^2] = \frac{n - m}{n - 1}\frac{1}{m}\hat{\sigma}_{t,1}^2 ,$$

where $\hat{\sigma}_{t,1}^2 := \frac{1}{n}\sum_{i=1}^{n}\|\nabla h_i(\mathbf{x}^t) - \nabla h_i(\mathbf{w}^t)\|^2$. Since $\omega_{t+1}$ is independent of $\mathbf{x}^{t+1}$ and $\mathbf{w}^t$, we have for any $\alpha > 0$:

$$\mathbb{E}_{\omega_{t+1}}[\hat{\sigma}_{t+1,1}^2] = (1 - p_B)\frac{1}{n}\sum_{i=1}^{n}\|\nabla h_i(\mathbf{x}^{t+1}) - \nabla h_i(\mathbf{w}^t)\|^2$$

$$\overset{(3),(E.2)}{\leq} (1 - p_B)(1 + \alpha)\hat{\sigma}_{t,1}^2 + (1 - p_B)\left(1 + \frac{1}{\alpha}\right)\delta^2\hat{\chi}_{t+1}^2 .$$

where $\hat{\chi}_{t+1} := \|\mathbf{x}^{t+1} - \mathbf{x}^t\|$. Let $1 - \gamma p_B := (1 - p_B)(1 + \alpha) \in (1 - p_B, 1)$. We then have $\gamma \in (0, 1)$ and $1 + 1/\alpha = \frac{1 - p_B\gamma}{p_B(1 - \gamma)}$. Therefore, the previous display can be reformulated as:

$$\mathbb{E}_{\omega_{t+1}}[\hat{\sigma}_{t+1,1}^2] \leq (1 - \gamma p_B)\hat{\sigma}_{t,1}^2 + \frac{(1 - p_B)(1 - p_B\gamma)}{p_B(1 - \gamma)}\delta^2\hat{\chi}_{t+1}^2 .$$

Taking the expectation w.r.t, $\omega_{[t]}$ on both sides and denoting $\sigma_{t,1}^2 := \mathbb{E}_{\omega_{[t]}}[\hat{\sigma}_{t,1}^2]$, we have:

$$\sigma_{t+1,1}^2 \leq (1 - \gamma p_B)\sigma_{t,1}^2 + \frac{(1 - p_B)(1 - p_B\gamma)}{p_B(1 - \gamma)}\delta^2\chi_{t+1}^2 .$$

Let $T \geq 2$. Applying Lemma E.4 (starting from $t = 1$), we obtain:

$$\sum_{t=2}^T \sigma_{t,1}^2 \leq \frac{1 - \gamma p_B}{\gamma p_B}\sigma_{1,1}^2 + \frac{(1 - p_B)(1 - p_B\gamma)}{p_B^2\gamma(1 - \gamma)}\delta^2 \sum_{t=1}^{T-1} \chi_{t+1}^2 .$$

Adding $\sigma_{1,1}^2$ to both sides and using $\sigma_{1,1}^2 = \mathbb{E}_{\omega_1}[\hat{\sigma}_{1,1}^2] \leq (1 - p_B)\delta^2\hat{\chi}_1^2 = (1 - p_B)\delta^2\chi_1^2$, we obtain:

$$\sum_{t=1}^T \sigma_{t,1}^2 \leq \frac{1}{\gamma p_B}(1 - p_B)\delta^2\chi_1^2 + \frac{(1 - p_B)(1 - p_B\gamma)}{p_B^2\gamma(1 - \gamma)}\delta^2 \sum_{t=1}^{T-1} \chi_{t+1}^2$$

$$\leq \frac{(1 - p_B)(1 - p_B\gamma)}{p_B^2\gamma(1 - \gamma)}\delta^2 \sum_{t=0}^{T-1} \chi_{t+1}^2 .$$

According to Lemma E.5, the minimizer of $\frac{(1 - p_B)(1 - p_B\gamma)}{p_B^2\gamma(1 - \gamma)}$ over $\gamma \in (0, 1)$ is $\gamma^\star = \frac{1 - \sqrt{1 - p_B}}{p_B}$. Substituting $\gamma = \gamma^\star$, we get:

$$\sum_{t=1}^T \sigma_{t,1}^2 \leq \frac{(1 - p_B)\delta^2}{(1 - \sqrt{1 - p_B})^2} \sum_{t=1}^T \chi_t^2 .$$

Multiplying both sides by $\frac{q_m}{m}$ and using the identity $\sigma_t^2 = \mathbb{E}_{S_t, \omega_{[t]}}[\|\mathbf{G}^t - \nabla f(\mathbf{x}^t)\|^2] = \frac{q_m}{m}\mathbb{E}_{\omega_{[t]}}[\hat{\sigma}_{t,1}^2] = \frac{q_m}{m}\sigma_{t,1}^2$, we have:

$$\sum_{t=1}^T \sigma_t^2 \leq \frac{(1 - p_B)\delta_m^2}{(1 - \sqrt{1 - p_B})^2} \sum_{t=1}^T \chi_t^2 \leq \frac{4\delta_m^2}{p_B^2} \sum_{t=1}^T \chi_t^2 .$$

Adding $\sigma_0^2 = \|\mathbf{G}^0 - \nabla f(\mathbf{x}^0)\|^2 = 0$ to both sides, we get the variance bound for $T \geq 2$. The same bound holds for $T = 1$ since $\sigma_0^2 = 0$ and $\sigma_1^2 = \frac{q_m}{m}\sigma_{1,1}^2 \leq (1 - p_B)\delta_m^2\chi_1^2$. □

**Theorem F.4.** *Let I-CGM be applied to Problem 1 with the SVRG estimator under Assumption 2.1 and 2.2. Suppose the inaccuracies in solving the subproblems satisfy (6). Then by choosing $\lambda = 3\Delta_1 + 16\delta_m/p_B$, after $T = \lceil \frac{(256(\Delta_1 + 6\delta_m/p_B)F^0}{\varepsilon^2} \rceil$ iterations, we have $\mathbb{E}[\|\nabla f(\bar{\mathbf{x}}^T)\|^2] \leq \varepsilon^2$, where $\bar{\mathbf{x}}^T$ is uniformly sampled from $(\mathbf{x}^t)_{t=1}^T$. By choosing $p_B = \frac{C_R}{C_A\lceil n_m \rceil}$ The communication complexity is at most $C_A\lceil n_m \rceil + (2C_R + 1)\lceil \frac{(256(\Delta_1 + 6\delta_m C_A\lceil n_m \rceil/C_R)F^0}{\varepsilon^2} \rceil$.*

The proof strategy is the same as the one for Theorem F.7.

**Implementation of SVRG.** At the beginning when $t = 0$, each client $i = 1, \ldots, n$ computes $\nabla f_i(\mathbf{x}^0)$ and sends the result to the server; the server then aggregates these results, computing $\nabla f(\mathbf{x}^0)$ to initialize $\mathbf{G}^0$. This requires one full synchronization. At each iteration $t \geq 1$, the server uses R-CSS and sends $\mathbf{w}^t$ and $\mathbf{x}^t$ to the clients in $S_t$ and then receives the gradient difference $\nabla f_i(\mathbf{x}^t) - \nabla f_i(\mathbf{w}^t)$ from them. If $\omega_t = 1$, the server computes the new gradient $\nabla f(\mathbf{w}^t)$ performing one full synchronization and stores it in memory; otherwise, it continues with $\nabla f(\mathbf{w}^t) = \nabla f(\mathbf{w}^{t-1})$ which is already stored in memory. In total, the server needs to maintain two points $\mathbf{x}^t$ and $\mathbf{w}^t$ and one vector $\nabla f(\mathbf{w}^t)$ and the clients are so-called stateless.

## F.4 PROPERTIES OF THE RG ESTIMATOR

### F.4.1 PROOF OF LEMMA 5.2

*Proof.* Let $t \geq 0$. By the definition of RG, we obtain:

$$\hat{\Sigma}_{t+1}^2 = \|\mathbf{g}^{t+1} - \nabla f(\mathbf{x}^{t+1})\|^2$$

$$= \left\|(1 - \beta)\mathbf{g}^t + \beta\mathbf{G}^t + \nabla f_{S_t}(\mathbf{x}^{t+1}) - \nabla f_{S_t}(\mathbf{x}^t) - \nabla f(\mathbf{x}^{t+1})\right\|^2$$

$$= \left\|(1 - \beta)(\mathbf{g}^t - \nabla f(\mathbf{x}^t)) + \beta(\mathbf{G}^t - \nabla f(\mathbf{x}^t)) + (\nabla h_{S_t}(\mathbf{x}^t) - \nabla h_{S_t}(\mathbf{x}^{t+1}))\right\|^2$$

$$= (1 - \beta)^2\hat{\Sigma}_t^2 + \left\|\beta(\mathbf{G}^t - \nabla f(\mathbf{x}^t)) + (\nabla h_{S_t}(\mathbf{x}^t) - \nabla h_{S_t}(\mathbf{x}^{t+1}))\right\|^2$$

$$+ 2(1 - \beta)\left\langle \mathbf{g}^t - \nabla f(\mathbf{x}^t), \beta(\mathbf{G}^t - \nabla f(\mathbf{x}^t)) + (\nabla h_{S_t}(\mathbf{x}^t) - \nabla h_{S_t}(\mathbf{x}^{t+1}))\right\rangle .$$

By Assumption 5.1, $S_t$ is independent of $\mathbf{x}^t$, $\mathbf{x}^{t+1}$ and $\mathbf{G}^{t-1}$. Furthermore, since $\mathbf{g}^t$ is a deterministic function of $\mathbf{G}^{t-1}$, $\mathbf{x}^t$, $\mathbf{x}^{t-1}$, $S_{t-1}$ and $\mathbf{g}^{t-1}$, by induction, $S_t$ is also independent of $\mathbf{g}^t$. Hence, it holds that:

$$\mathbb{E}_{S_t}[\langle \mathbf{g}^t - \nabla f(\mathbf{x}^t), \beta(\mathbf{G}^t - \nabla f(\mathbf{x}^t)) + \nabla h_{S_t}(\mathbf{x}^t) - \nabla h_{S_t}(\mathbf{x}^{t+1})\rangle]$$
$$= \langle \mathbf{g}^t - \nabla f(\mathbf{x}^t), \beta\,\mathbb{E}_{S_t}[\mathbf{G}^t - \nabla f(\mathbf{x}^t)] + \mathbb{E}_{S_t}[\nabla h_{S_t}(\mathbf{x}^t) - \nabla h_{S_t}(\mathbf{x}^{t+1})]\rangle \ .$$

By Assumption 5.1, we have $\mathbb{E}_{S_t}[\mathbf{G}^t] = \nabla f(\mathbf{x}^t)$. Using Lemma E.2, we have $\mathbb{E}_{S_t}[\nabla h_{S_t}(\mathbf{x}^t) - \nabla h_{S_t}(\mathbf{x}^{t+1})] = \frac{1}{n}\sum_{i=1}^n[\nabla h_i(\mathbf{x}^t) - \nabla h_i(\mathbf{x}^{t+1})] = 0$ and

$$\mathbb{E}_{S_t}[\|\nabla h_{S_t}(\mathbf{x}^t) - \nabla h_{S_t}(\mathbf{x}^{t+1})\|^2] \stackrel{(E.3)}{=} \frac{q_m}{m}\frac{1}{n}\sum_{i=1}^n \|\nabla h_i(\mathbf{x}^t) - \nabla h_i(\mathbf{x}^{t+1})\|^2 \stackrel{(2.2)}{\leq} \delta_m^2 \hat{\chi}_{t+1}^2 \ .$$

Taking the expectation w.r.t. $S_t$ on both sides of the first display, we get:

$$\mathbb{E}_{S_t}[\hat{\Sigma}_{t+1}^2]$$
$$= (1-\beta)^2\hat{\Sigma}_t^2 + \mathbb{E}_{S_t}[\|\beta(\mathbf{G}^t - \nabla f(\mathbf{x}^t)) + (\nabla h_{S_t}(\mathbf{x}^t) - \nabla h_{S_t}(\mathbf{x}^{t+1}))\|^2]$$
$$\leq (1-\beta)^2\hat{\Sigma}_t^2 + 2\beta^2\,\mathbb{E}_{S_t}[\|\mathbf{G}^t - \nabla f(\mathbf{x}^t)\|^2] + 2\delta_m^2\hat{\chi}_{t+1}^2 \ .$$

Taking the expectation w.r.t. $S_{[t-1]}$ on both sides and substituting the notations, we get:

$$\Sigma_{t+1}^2 \leq (1-\beta)^2\Sigma_t^2 + 2\beta^2\sigma_t^2 + 2\delta_m^2\chi_{t+1}^2 \ .$$

Applying Lemma E.4, we get for any $T \geq 1$:

$$\sum_{t=1}^T \Sigma_t^2 \leq \frac{(1-\beta)^2}{2\beta - \beta^2}\Sigma_0^2 + \frac{2\beta}{2-\beta}\sum_{t=0}^{T-1}\sigma_t^2 + \frac{2\delta_m^2}{2\beta - \beta^2}\sum_{t=0}^{T-1}\chi_{t+1}^2 \ . \tag{F.1}$$

This proves the claim since $\mathbf{g}^0 = \nabla f(\mathbf{x}^0)$ and so $\Sigma_0^2 = 0$. $\qquad \square$

### F.4.2 Proof of Corollary 5.3.

*Proof.* Let $T \geq 1$. Note that, under Assumption 5.1, we have $\mathbb{E}_{S_{[t-1]}}[\|\mathbf{x}^t - \mathbf{x}^{t-1}\|^2] = \mathbb{E}_{S_{[t-2]}}[\|\mathbf{x}^t - \mathbf{x}^{t-1}\|^2] = \chi_t^2$ and $\mathbb{E}_{S_{[t-1]}}[\|\nabla f(\mathbf{x}^t)\|^2] = \mathbb{E}_{S_{[t-2]}}[\|\nabla f(\mathbf{x}^t)\|^2] = G_t^2$. Applying Lemma 4.1, we have:

$$\sum_{t=0}^T \sigma_t^2 \leq \frac{2n_m q_m}{m}G_1^2 + \frac{n_m - 1 + \sqrt{n_m^2 - n_m}}{(n-1)}\sum_{t=2}^{T-1}G_t^2 + 4n_m^2\delta_m^2\sum_{t=2}^T\chi_t^2 \ .$$

Applying Lemma 5.2, we obtain:

$$\sum_{t=0}^T \Sigma_t^2 \leq \frac{2\beta}{2-\beta}\sum_{t=0}^{T-1}\sigma_t^2 + \frac{2\delta_m^2}{2\beta - \beta^2}\sum_{t=1}^T\chi_t^2 \ .$$

Combining the previous two displays, we have:

$$\sum_{t=0}^T \Sigma_t^2 \leq \frac{4\beta n_m q_m}{(2-\beta)m}G_1^2 + \frac{2\beta(n_m - 1 + \sqrt{n_m^2 - n_m})}{(2-\beta)(n-1)}\sum_{t=2}^{T-1}G_t^2 + \frac{8\beta^2 n_m^2\delta_m^2 + 2\delta_m^2}{2\beta - \beta^2}\sum_{t=1}^T\chi_t^2 \ . \qquad \square$$

### F.4.3 Proof of Corollary 5.4.

*Proof.* Let $T \geq 1$. Applying Lemma 4.3, we get:

$$\sum_{t=0}^T \mathbb{E}_{S_t,\omega_{[t]}}[\|\mathbf{G}^t - \nabla f(\mathbf{x}^t)\|^2] \leq \frac{4\delta_m^2}{p_B^2}\sum_{t=1}^T \mathbb{E}_{\omega_{[t-1]}}[\|\mathbf{x}^t - \mathbf{x}^{t-1}\|^2] \ .$$

Note that, under Assumption 5.1, $S_{t-1}$ is independent of $\mathbf{x}^t$ and $\mathbf{x}^{t-1}$. Moreover, the first iterate that might depend on $\omega_{t-1}$ is $\mathbf{x}^{t+1}$ since $\mathbf{g}^t$ is computed using $\mathbf{x}^t$ and $\mathbf{G}^{t-1}$ which is a function of $\omega_{t-1}$. Therefore, $\omega_{t-1}$ is also independent of $\mathbf{x}^t$ and $\mathbf{x}^{t-1}$. Hence, we have $\mathbb{E}_{S_{[t-1]},\omega_{[t-1]}}[\|\mathbf{x}^t - \mathbf{x}^{t-1}\|^2] = \mathbb{E}_{S_{[t-2]},\omega_{[t-2]}}[\hat{\chi}_t^2] = \chi_t^2$. Taking the expectation w.r.t. $S_{[t-1]}$ on both sides of the first display, we get:

$$\sum_{t=0}^T \sigma_t^2 \leq \frac{4\delta_m^2}{p_B^2}\sum_{t=1}^T\chi_t^2 \ .$$

where $\sigma_t^2 := \mathbb{E}_{S_{[t]}, \omega_{[t]}}[\|\mathbf{G}^t - \nabla f(\mathbf{x}^t)\|^2]$. Applying Lemma 5.2, taking the expectation w.r.t. $\omega_{[t]}$ and substituting the identity $\mathbb{E}_{S_{[t-1]}, \omega_{[t]}}[\hat{\Sigma}_t^2] = \mathbb{E}_{S_{[t-1]}, \omega_{[t-1]}}[\hat{\Sigma}_t^2] = \Sigma_t^2$ and $\mathbb{E}_{S_{[t-2]}, \omega_{[t]}}[\hat{\chi}_t^2] = \mathbb{E}_{S_{[t-2]}, \omega_{[t-2]}}[\hat{\chi}_t^2] = \chi_t^2$, we obtain:

$$\sum_{t=0}^{T} \Sigma_t^2 \leq \frac{2\beta}{2-\beta} \sum_{t=0}^{T-1} \sigma_t^2 + \frac{2\delta_m^2}{2\beta - \beta^2} \sum_{t=1}^{T} \chi_t^2 .$$

Combining the previous two displays, we have:

$$\sum_{t=0}^{T} \Sigma_t^2 \leq \frac{8\beta^2 \delta_m^2 / p_B^2 + 2\delta_m^2}{2\beta - \beta^2} \sum_{t=1}^{T} \chi_t^2 . \qquad \square$$

## F.5 PROOFS FOR I-CGM WITH RG-SAGA

**Lemma F.5.** *Let $\mathbf{x}^t$ be the iterates of I-CGM-RG-SAGA and let $\mathbf{G}^t$ be the-SAGA estimator for all $t \geq 0$. Let $\zeta_t$ denote the randomness generated during the process of solving the subproblem $F_{t-1}$ in I-CGM for any $t \geq 1$. Assume that $\{\zeta_t\}_{t=1}^{\infty}$ are mutually independent across t. Then the iterates $\{\mathbf{x}^t\}_{t=0}^{\infty}$ and the estimators $\{\mathbf{G}^t\}_{t=0}^{\infty}$ satisfy Assumption 5.1.*

*Proof.* The equation $\mathbb{E}_{S_t}[\mathbf{G}^t] = \nabla f(\mathbf{x}^t)$ has been proved in Lemma 4.1. We next verify the dependency of randomness. Let $t \geq 1$ and denote $S_{[t]} := (S_0, \ldots, S_t)$ and $\zeta_{[t]} := (\zeta_1, \ldots, \zeta_t)$. Assume that $\mathbf{x}^t$ is a deterministic function of $(S_{[t-2]}, \zeta_{[t]})$. Then $\mathbf{G}^t$ is a deterministic function of $(S_{[t]}, \zeta_{[t]})$ since $\mathbf{G}^t$ depends only on $\mathbf{x}_{[t]} := (\mathbf{x}^0, \mathbf{x}^1, ..., \mathbf{x}^t)$ and $S_t$. Next observe that $\mathbf{g}^{t-1}$ is a function of $S_{t-2}, \mathbf{x}^{t-1}, \mathbf{x}^{t-2}, \mathbf{G}^{t-2}$ and $\mathbf{g}^{t-2}$. Therefore, $\mathbf{g}^{t-1}$ is a deterministic function of $S_{[t-2]}$ and $\zeta_{[t-1]}$. Finally, from the update rule of I-CGM, $\mathbf{x}^t$ is a deterministic function of $\mathbf{g}^{t-1}, \zeta_t$ and $\mathbf{x}^{t-1}$. Therefore, the assumption that $\mathbf{x}^t$ is determinsitic conditioned on $(S_{[t-2]}, \zeta_{[t]})$ is satisfied. This implies that $S_t$ is independent of $\mathbf{x}_{[t+1]}, \mathbf{G}^{t-1}, \ldots, \mathbf{G}^0$. $\qquad \square$

### F.5.1 PROOF OF THEOREM 6.1.

*Proof.* According to Lemma H.2, by choosing $p = \frac{\lambda - \Delta_1}{8(L_1 + \lambda)}$, the accuracy condition (6) for solving the subproblems is satisfied. Applying Corollary 5.3 and taking the full expectation, for any $T \geq 1$, we have:

$$\sum_{t=0}^{T} \Sigma_t^2 \leq \frac{4\beta q_m n_m}{(2-\beta)m} G_1^2 + \frac{2\beta(n_m - 1 + \sqrt{n_m^2 - n_m})}{(2-\beta)(n-1)} \sum_{t=2}^{T-1} G_t^2 + \frac{8\beta^2 n_m^2 \delta_m^2 + 2\delta_m^2}{2\beta - \beta^2} \sum_{t=1}^{T} \chi_t^2 ,$$

where $\Sigma_t^2$, $G_t^2$ and $\chi_t^2$ are defined in Corollary 3.2. Using $\frac{1}{2-\beta} \leq 1$, $\frac{q_m}{m} \leq 1$, $\frac{n_m - 1}{n-1} \leq 1 \leq n_m$, and $\frac{\sqrt{n_m^2 - n_m}}{n-1} \leq n_m$ as $n \geq 2$, we get:

$$\sum_{t=0}^{T} \Sigma_t^2 \leq 4\beta n_m \sum_{t=1}^{T-1} G_t^2 + \left(8\beta n_m^2 \delta_m^2 + \frac{2\delta_m^2}{\beta}\right) \sum_{t=1}^{T} \chi_t^2 .$$

Let $\lambda = \frac{1}{a}\Delta_1 + b\sqrt{n_m}\delta_m$ and $\beta = \frac{1}{cn_m}$ where $0 < a < 1$ and $b, c > 0$. To achieve the error condition (7), the constants should satisfy:

$$\left(\frac{12(\lambda + \Delta_1)^2}{(\lambda - \Delta_1)^2} + 8\right)4\beta n_m \leq \left(\frac{12(1+a)^2}{(1-a)^2} + 8\right)\frac{4}{c} \leq \frac{1}{2} ,$$

and

$$\left(\frac{12(\lambda + \Delta_1)^2}{(\lambda - \Delta_1)^2} + 8\right)\left(8\beta n_m^2 \delta_m^2 + \frac{2\delta_m^2}{\beta}\right) \leq \left(\frac{12(1+a)^2}{(1-a)^2} + 8\right)(8/c + 2c)n_m \delta_m^2 \leq b^2 n_m \delta_m^2 \leq (\lambda + \Delta_1)^2 ,$$

which gives:

$$\left(\frac{12(1+a)^2}{(1-a)^2} + 8\right) \leq \frac{c}{2}, \quad \left(\frac{12(1+a)^2}{(1-a)^2} + 8\right)(8/c + 2c) \leq b^2 . \tag{F.2}$$

Let $a, b, c$ satisfy (F.2). We can apply Corollary 3.2 and obtain:

$$\mathbb{E}[\|\nabla f(\bar{\mathbf{x}}^T)\|^2] \leq \frac{32(\lambda + \Delta_1)^2}{\lambda - \Delta_1} \frac{F^0}{T} \leq \frac{32(1+a)^2}{1-a}\left(\frac{1}{a}\Delta_1 + b\sqrt{n_m}\delta_m\right)\frac{F^0}{T} .$$

Minimizing the coefficient in front of $\Delta_1$ gives $a^\star = \frac{1}{3}$. Choosing $b = 113$ and $c = 112$, the condition (F.2) is satisfied and we have:

$$\mathbb{E}[\|\nabla f(\bar{\mathbf{x}}^T)\|^2] \leq \frac{256(\Delta_1 + 38\sqrt{n_m}\delta_m)F^0}{T} .$$

Therefore, to achieve $\mathbb{E}[\|\nabla f(\bar{\mathbf{x}}^T)\|^2] \leq \varepsilon^2$, we need at most $T = \lceil \frac{(256(\Delta_1 + 38\sqrt{n_m}\delta_m)F^0}{\varepsilon^2} \rceil$ iterations. We next compute the communication and local complexity. At the beginning when $t = 0$ and $t = 1$, we need $2\lceil n_m \rceil$ communication rounds with A-CSS to compute two full gradients and the associated local complexity is 1 for each round. Additionally, 2 communication rounds with D-CSS are needed to compute $\mathbf{x}^1$ and $\mathbf{x}^2$, where the local complexity is $\frac{1}{p}$ for each round. For subsequent iterations $t \geq 2$, one communication round with R-CSS is needed for updating $\mathbf{g}^t$ and its associated local complexity is 2 since each client in $S_{t-1}$ needs to compute $\nabla f_i(\mathbf{x}^t)$ and $\nabla f_i(\mathbf{x}^{t-1})$. Then another round with D-CSS is required to compute the next iterate, where the local complexity is $\frac{1}{p}$. Therefore, the total communication complexity is at most:

$$
\begin{aligned}
N(\varepsilon) &= \mathbb{E}[C_A N_A + C_R N_R + N_D] \\
&\leq 2C_A \lceil n_m \rceil + C_R T + T \\
&= 2C_A \lceil n_m \rceil + (C_R + 1) \left\lceil \frac{(256(\Delta_1 + 38\sqrt{n_m}\delta_m)F^0}{\varepsilon^2} \right\rceil .
\end{aligned}
$$

The local complexity is bounded by:

$$
\begin{aligned}
K(\varepsilon) &= \mathbb{E}[N_A + N_D/p + 2N_R] \\
&\leq 2\lceil n_m \rceil + \frac{1}{p}T + 2T \\
&= 2\lceil n_m \rceil + \left(2 + \frac{8(L_1 + \lambda)}{\lambda - \Delta_1}\right)T \\
&\leq 2\lceil n_m \rceil + \frac{28\Delta_1 + 1130\sqrt{n_m}\delta_m + 8L_1}{2\Delta_1 + 113\sqrt{n_m}\delta_m}\left(\frac{(256(\Delta_1 + 38\sqrt{n_m}\delta_m)F^0}{\varepsilon^2} + 1\right) \\
&\leq 2\lceil n_m \rceil + \frac{512(7\Delta_1 + 283\sqrt{n_m}\delta_m + 2L_1)F^0}{\varepsilon^2} + 14 + \frac{4L_1}{\Delta_1 + 28\sqrt{n_m}\delta_m} . \qquad \square
\end{aligned}
$$

## F.6 Proofs for I-CGM with RG-SVRG

**Lemma F.6.** *Let $\mathbf{x}^t$ be the iterates of I-CGM-RG-SVRG and let $\mathbf{G}^t$ be the-SVRG estimator for all $t \geq 0$. Let $\zeta_t$ denote the randomness generated during the process of solving the subproblem $F_{t-1}$ in I-CGM for any $t \geq 1$. Assume that $\{\zeta_t\}_{t=1}^{\infty}$ are mutually independent across $t$. Then the iterates $\{\mathbf{x}^t\}_{t=0}^{\infty}$ and the estimators $\{\mathbf{G}^t\}_{t=0}^{\infty}$ satisfy Assumption 5.1.*

*Proof.* The equation $\mathbb{E}_{S_t}[\mathbf{G}^t] = \nabla f(\mathbf{x}^t)$ has been proved in Lemma 4.3. We next verify the dependency of randomness. Let $t \geq 1$ and denote $\mathbf{x}_{[t]} = (\mathbf{x}^0, .., \mathbf{x}^t)$, $\omega_{[t]} = (\omega_1, ..., \omega_t)$ and $\zeta_{[t]} = (\zeta_1, ..., \zeta_t)$. Assume that $\mathbf{x}^t$ is a deterministic function of $(S_{[t-2]}, \omega_{[t]}, \zeta_{[t]})$. It follows that $\mathbf{w}^t$ is a deterministic function of $(S_{[t-2]}, \omega_{[t]}, \zeta_{[t]})$ since $\mathbf{w}^t$ depends only on $\mathbf{x}^t$, $\mathbf{w}^{t-1}$ and $\omega_t$. Then $\mathbf{G}^t$ is deterministic conditioned on $(S_{[t]}, \omega_{[t]}, \zeta_{[t]})$ since $\mathbf{G}^t$ is a function of $\mathbf{x}_t$, $\mathbf{w}_t$ and $S_t$. Next observe that $\mathbf{g}^{t-1}$ is a function of $S_{t-2}$, $\mathbf{x}^{t-1}$, $\mathbf{x}^{t-2}$, $\mathbf{G}^{t-2}$ and $\mathbf{g}^{t-2}$. Hence, $\mathbf{g}^{t-1}$ is a deterministic function of $(S_{[t-2]}, \omega_{[t-2]}, \zeta_{[t-1]})$. Finally, from the update rule of I-CGM, $\mathbf{x}^t$ is a deterministic function of $\mathbf{g}^{t-1}$, $\zeta_t$ and $\mathbf{x}^{t-1}$. Therefore, the assumption that $\mathbf{x}^t$ is deterministic conditioned on $(S_{[t-2]}, \omega_{[t-2]}, \zeta_{[t]})$ is satisfied. This implies that $S_t$ is independent of $\mathbf{x}_{[t+1]}$, $\mathbf{G}^0, ..., \mathbf{G}^{t-1}$. $\square$

**Theorem F.7** (I-CGM-RG-SVRG). *Let I-CGM be applied to Problem 1 under Assumptions 2.1, 2.2 and 2.3, where $\mathbf{x}^{t+1} = \text{CGM}_{\text{rand}}(\lambda, \hat{K}_t, \mathbf{x}^t, \mathbf{g}^t)$ with $\hat{K}_t \sim \text{Geom}(p)$ and $\mathbf{g}^t$ is generated by the RG-SVRG estimator. Then by choosing $\lambda = 3\Delta_1 + 22\delta_m/\sqrt{p_B}$, $\beta = \frac{p_B}{2}$, and $p = \frac{\lambda - \Delta_1}{8(L_1 + \lambda)}$, after $T = \lceil \frac{(256(\Delta_1 + 8\delta_m/\sqrt{p_B})F^0}{\varepsilon^2} \rceil$ iterations, we have $\mathbb{E}[\|\nabla f(\bar{\mathbf{x}}^T)\|^2] \leq \varepsilon^2$, where $\bar{\mathbf{x}}^T$ is is uniformly sampled from $(\mathbf{x}^t)_{t=1}^T$. Further let $p_B = \frac{C_R}{C_A \lceil n_m \rceil}$. The communication complexity is at most $C_A \lceil n_m \rceil + (2C_R + 1) \lceil \frac{(256(\Delta_1 + 8\delta_m\sqrt{C_A \lceil n_m \rceil/C_R}F^0}{\varepsilon^2} \rceil$ and the local complexity is bounded by $16 + \lceil n_m \rceil + \frac{1024(L_1 + 4\Delta_1 + 33\sqrt{C_A \lceil n_m \rceil/C_R}\delta_m)F^0}{\varepsilon^2} + \frac{4L_1}{\Delta_1 + 11\sqrt{C_A \lceil n_m \rceil/C_R}\delta_m}$.*

*Proof.* According to Lemma H.2, by choosing $p = \frac{\lambda - \Delta_1}{8(L_1 + \lambda)}$, the accuracy condition (6) for solving the subproblems is satisfied. Applying Corollary 5.4 and taking the full expectation, for any $T \geq 1$, we have:

$$
\sum_{t=0}^{T} \Sigma_t^2 \leq \frac{8\beta^2 \delta_m^2/p_B^2 + 2\delta_m^2}{2\beta - \beta^2} \sum_{t=1}^{T} \chi_t^2 \leq \left(\frac{8\beta\delta_m^2}{p_B^2} + \frac{2\delta_m^2}{\beta}\right) \sum_{t=1}^{T} \chi_t^2 .
$$

Let $\lambda = \frac{1}{a}\Delta_1 + b\delta_m/\sqrt{p_B}$ and $\beta = \frac{p_B}{c}$ where $0 < a < 1$ and $b, c > 0$. To achieve the error condition (7), the constants should satisfy:

$$\Big(\frac{12(\lambda+\Delta_1)^2}{(\lambda-\Delta_1)^2}+8\Big)\Big(\frac{8\beta\delta_m^2}{p_B^2}+\frac{2\delta_m^2}{\beta}\Big) \le \Big(\frac{12(1+a)^2}{(1-a)^2}+8\Big)(8/c+2c)\delta_m^2/p_B \le b^2\delta_m^2/p_B \le (\lambda+\Delta_1)^2 \ ,$$

which gives:

$$\Big(\frac{12(1+a)^2}{(1-a)^2}+8\Big)(8/c+2c) \le b^2 \ . \tag{F.3}$$

Let $a, b, c$ satisfy (F.3). We can apply Corollary 3.2 and obtain:

$$\mathbb{E}[\|\nabla f(\bar{\mathbf{x}}^T)\|^2] \le \frac{32(\lambda+\Delta_1)^2}{\lambda-\Delta_1}\frac{F^0}{T} \le \frac{32(1+a)^2}{1-a}\Big(\frac{1}{a}\Delta_1+b\delta_m/\sqrt{p_B}\Big)\frac{F^0}{T} \ .$$

Minimizing the coefficient in front of $\Delta_1$ gives $a^\star = \frac{1}{3}$. Choosing $b = 22$ and $c = 2$, the condition (F.3) is satisfied and we have:

$$\mathbb{E}[\|\nabla f(\bar{\mathbf{x}}^T)\|^2] \le \frac{256(\Delta_1+8\delta_m/\sqrt{p_B})F^0}{T} \ .$$

Therefore, to achieve $\mathbb{E}[\|\nabla f(\bar{\mathbf{x}}^T)\|^2] \le \varepsilon^2$, we need at most $T = \lceil\frac{(256(\Delta_1+8\delta_m/\sqrt{p_B})F^0}{\varepsilon^2}\rceil$ iterations. We next compute the communication and local complexity. At iteration $t = 0$, the full gradient $\nabla f(\mathbf{x}^0)$ is computed which requires $\lceil n_m\rceil$ communication rounds with A-CSS. At each iteration $t \ge 1$, with probability $p_B$, the full gradient is computed, which requires $\lceil n_m\rceil$ rounds with A-CSS. The expected total number of rounds where A-CSS is used is thus bounded by: $\lceil n_m\rceil + \lceil n_m\rceil p_B T$ . The associated local complexity for each round with A-CSS is always 1. For $t \ge 1$, one communication round with R-CSS is needed for updating $\mathbf{g}^t$ and its associated local complexity is 3 since the client $i \in S_{t-1}$ needs to compute $\nabla f_i(\mathbf{x}^{t-1})$, $\nabla f_i(\mathbf{w}^{t-1})$ and $\nabla f_i(\mathbf{x}^t)$. Then another round with D-CSS is established, which has the local complexity of $1/p$. Therefore, the communication complexity is bounded by:

$$N(\varepsilon) = \mathbb{E}[C_A N_A + C_R N_R + N_D] \le C_A(\lceil n_m\rceil+\lceil n_m\rceil p_B T)+C_R T+T = C_A\lceil n_m\rceil+(C_A\lceil n_m\rceil p_B+C_R+1)T \ .$$

Let $C_A\lceil n_m\rceil p_B = C_R$. We have

$$N(\varepsilon) \le C_A\lceil n_m\rceil + (2C_R+1)\left\lceil\frac{(256(\Delta_1+8\delta_m\sqrt{C_A\lceil n_m\rceil/C_R})F^0}{\varepsilon^2}\right\rceil \ .$$

The local complexity $K(\varepsilon)$ is bounded by:

$$\begin{aligned}
\mathbb{E}[N_A + N_D/p + 3N_R] &= \lceil n_m\rceil + \lceil n_m\rceil p_B T + T/p + 3T \\
&\le \lceil n_m\rceil + (\frac{8(L_1+\lambda)}{\lambda-\Delta_1}+4)T \\
&= \lceil n_m\rceil + \frac{8L_1+32\Delta_1+264\delta_m/\sqrt{p_B}}{2\Delta_1+22\delta_m/\sqrt{p_B}}\Big(\frac{(256(\Delta_1+8\delta_m/\sqrt{p_B})F^0}{\varepsilon^2}+1\Big) \\
&\le \lceil n_m\rceil + 128\frac{(8L_1+32\Delta_1+264\delta_m/\sqrt{p_B})F^0}{\varepsilon^2}+16+\frac{4L_1}{\Delta_1+11\delta_m/\sqrt{p_B}} \ . \ \square
\end{aligned}$$

## G  I-CGM-RG-SAGA WITH INEXACT INITIALIZATION

In the main paper, we consider the RG- SAGA estimator with two full gradient computation during initialization. This allows to satisfy the desired error condition (7) without incurring additional error. In this section, we discuss the case where only one or no full gradient is computed for this estimator. Let us introduce the parameter $t_0 \in \{0, 1, 2\}$ to determine how many times the full gradient is computed at the beginning. We have the following definition for the general SAGA estimator:

$$\boxed{\mathbf{G}^0 = \nabla f_{S_0}(\mathbf{x}^0), t_0 = 0, \quad \mathbf{G}^t = \nabla f(\mathbf{x}^t), \ 0 \le t \le t_0-1, \quad \mathbf{G}^t = \mathbf{b}_{S_t}^t - \mathbf{b}_{S_t}^{t-1}+\mathbf{b}^{t-1}, \ t \ge \max(t_0,1)} \ ,$$

$$\tag{G.1}$$

where $S_t \in \binom{[n]}{m}$ is uniformly sampled at random without replacement, $\mathbf{b}_{S_t}^t := \frac{1}{m}\sum_{i\in S_t}\mathbf{b}_i^t$, $\mathbf{b}_{S_t}^{t-1} := \frac{1}{m}\sum_{i\in S_t}\mathbf{b}_i^{t-1}$, $\mathbf{b}^t := \frac{1}{n}\sum_{i=1}^n\mathbf{b}_i^t$ for all $t \ge \max(t_0,1)$, and for any $i \in [n]$, $\mathbf{b}_i^t$ is recurrently defined as:

$$\mathbf{b}_i^0 = \begin{cases}\nabla f_i(\mathbf{x}^0) & \text{if } i \in S_0, \\ \mathbf{0} & \text{otherwise,}\end{cases}, t_0 = 0, \qquad \mathbf{b}_i^t = \nabla f_i(\mathbf{x}^t), \ 0 \le t \le t_0-1, \forall i \in [n],$$

and

$$\mathbf{b}_i^t = \begin{cases} \nabla f_i(\mathbf{x}^t) & \text{if } i \in S_t, \\ \mathbf{b}_i^{t-1} & \text{otherwise,} \end{cases}, \quad t \geq \max(t_0, 1) .$$

We have the following recurrence for $\mathbf{b}^t$:

$$\mathbf{b}^0 = \begin{cases} \nabla f_{S_0}(\mathbf{x}^0) & \text{if } t_0 = 0, \\ \frac{1}{n}\sum_{i=1}^n \mathbf{b}_i^0 & \text{otherwise,} \end{cases} \quad \mathbf{b}^t = \mathbf{b}^{t-1} + \frac{1}{n_m}[\nabla f_{S_t}(\mathbf{x}^t) - \mathbf{b}_{S_t}^{t-1}], \quad t \geq \max(t_0, 1) .$$

**Lemma G.1.** *Consider the* SAGA *estimator* (G.1) *under Assumption 2.2. Let* $t_0 \in \{0, 1\}$*. Then for any* $t \geq 0$*,* $\mathbb{E}_{S_t}[\mathbf{G}^t] = \nabla f(\mathbf{x}^t)$ *and for any* $T \geq 1$*, we have :*

$$\sum_{t=0}^T \sigma_t^2 \leq \frac{2n_m q_m}{m}\|\nabla f(\mathbf{x}^0)\|^2 + \frac{n_m - 1 + \sqrt{n_m^2 - n_m}}{(n-1)}\sum_{t=1}^{T-1} G_t^2 + 4n_m^2\delta_m^2 \sum_{t=1}^T \chi_t^2 , \quad (t_0 = 1)$$

$$\sum_{t=0}^T \sigma_t^2 \leq \left(12\sqrt{n_m^2 - n_m} + \frac{q_m}{m}\right)\zeta_0^2 + 8\sqrt{n_m^2 - n_m}\|\nabla f(\mathbf{x}^0)\|^2$$

$$+ \frac{n_m - 1 + \sqrt{n_m^2 - n_m}}{(n-1)}\sum_{t=1}^{T-1} G_t^2 + 4n_m^2\delta_m^2 \sum_{t=1}^T \chi_t^2 , \quad (t_0 = 0)$$

*where* $\sigma_t^2 := \mathbb{E}_{S_{[t]}}[\|\mathbf{G}^t - \nabla f(\mathbf{x}^t)\|^2]$*,* $G_t^2 = \mathbb{E}_{S_{[t-1]}}[\|\nabla f(\mathbf{x}^t)\|^2]$*,* $\chi_t^2 := \mathbb{E}_{S_{[t-1]}}[\|\mathbf{x}^t - \mathbf{x}^{t-1}\|^2]$*,* $S_{[t]} := (S_{t_0}, ..., S_t)$ *and* $\zeta_0^2 := \frac{1}{n}\sum_{i=1}^n \|\nabla f_i(\mathbf{x}^0) - \nabla f(\mathbf{x}^0)\|^2$*.*

*Proof.* When $t_0 = 0$, we have $\mathbb{E}_{S_0}[\mathbf{G}^0] = \mathbb{E}_{S_0}[\nabla f_{S_0}] = \nabla f(\mathbf{x}^0)$. When $t_0 = 1$, we have $\mathbf{G}^0 = \nabla f(\mathbf{x}^0)$. For $t \geq 1$, $\mathbf{G}^t = \mathbf{b}_{S_t}^t - \mathbf{b}_{S_t}^{t-1} + \mathbf{b}^{t-1}$ and the unbiasedness has been proven in Lemma 4.1. We next study the variance bound. Following the proof in Section F.3.1, we have for any $t \geq 1$:

$$\sigma_{t+1,1}^2 + \mathbf{B}_t^2 \leq \left(1 - \frac{\gamma}{n_m}\right)[\sigma_{t,1}^2 + \mathbf{B}_{t-1}^2] + \frac{n_m - \gamma}{n_m(n_m - 1)}G_t^2 + \frac{n_m - \gamma}{1 - \gamma}\delta^2\chi_{t+1}^2 ,$$

where $\sigma_{t,1}^2 := \mathbb{E}_{S_{[t-1]}}[\frac{1}{n}\sum_{i=1}^n \|(\nabla f_i(\mathbf{x}^t) - \mathbf{b}_i^{t-1}) - (\nabla f(\mathbf{x}^t) - \mathbf{b}^{t-1})\|^2]$, $\mathbf{B}_t^2 := \frac{n_m - \gamma}{n_m - 1}\mathbb{E}_{S_{[t]}}[\|\mathbf{b}^t\|^2]$ and $\gamma \in (0, 1)$. Let $T \geq 2$. Applying Lemma E.4 (starting from $t = 1$), we have:

$$\sum_{t=2}^T [\sigma_{t,1}^2 + \mathbf{B}_{t-1}^2] \leq \frac{1 - \gamma/n_m}{\gamma/n_m}[\sigma_{1,1}^2 + \mathbf{B}_0^2] + \frac{1}{\gamma/n_m}\sum_{t=1}^{T-1}\left[\frac{n_m - \gamma}{n_m(n_m - 1)}G_t^2 + \frac{n_m - \gamma}{1 - \gamma}\delta^2\chi_{t+1}^2\right] .$$

Adding $\sigma_{1,1}^2$ to both sides and dropping the non-negative $\mathbf{B}_{t-1}^2$, we obtain:

$$\sum_{t=1}^T \sigma_{t,1}^2 \leq \frac{n_m}{\gamma}\sigma_{1,1}^2 + \frac{n_m - \gamma}{\gamma}\mathbf{B}_0^2 + \frac{n_m - \gamma}{\gamma(n_m - 1)}\sum_{t=1}^{T-1} G_t^2 + \frac{n_m(n_m - \gamma)}{\gamma(1 - \gamma)}\delta^2\sum_{t=1}^{T-1}\chi_{t+1}^2 . \quad (G.2)$$

Suppose $t_0 = 1$. Then we have $\mathbf{b}_i^0 = \nabla f_i(\mathbf{x}^0)$ for all $i \in [n]$. It holds that:

$$\sigma_{1,1}^2 = \frac{1}{n}\sum_{i=1}^n [\|(\nabla f_i(\mathbf{x}^1) - \nabla f_i(\mathbf{x}^0)) - (\nabla f(\mathbf{x}^1) - \nabla f(\mathbf{x}^0))\|^2 \overset{(2.2)}{\leq} \delta^2\chi_1^2, \quad \mathbf{B}_0^2 = \frac{n_m - \gamma}{n_m - 1}\|\nabla f(\mathbf{x}^0)\|^2 .$$

It follows that:

$$\sum_{t=1}^T \sigma_{t,1}^2 \leq \frac{(n_m - \gamma)^2}{\gamma(n_m - 1)}\|\nabla f(\mathbf{x}^0)\|^2 + \frac{n_m - \gamma}{\gamma(n_m - 1)}\sum_{t=1}^{T-1} G_t^2 + \frac{n_m(n_m - \gamma)}{\gamma(1 - \gamma)}\delta^2\sum_{t=0}^{T-1}\chi_{t+1}^2 .$$

Substituting $\gamma = \gamma^\star = n_m - \sqrt{n_m^2 - n_m}$ that minimizes the coefficient in front of $\sum_{t=0}^{T-1}\chi_{t+1}^2$ over $(0, 1)$, multiplying both sides by $\frac{q_m}{m}$, substituting the identity $\sigma_t^2 = \frac{q_m}{m}\sigma_{t,1}^2$, we have:

$$\sum_{t=1}^T \sigma_t^2 \leq \frac{2n_m q_m}{m}\|\nabla f(\mathbf{x}^0)\|^2 + \frac{n_m - 1 + \sqrt{n_m^2 - n_m}}{(n-1)}\sum_{t=1}^{T-1} G_t^2 + 4n_m^2\delta_m^2 \sum_{t=1}^T \chi_t^2 .$$

Adding $\sigma_0^2 = 0$ to both sides, we prove the variance bound for $T \geq 2$, since $\mathbf{G}^0 = \nabla f(\mathbf{x}^0)$. The same inequality also holds for $T = 1$, since $\sigma_1^2 = \frac{q_m}{m}\sigma_{1,1}^2 \leq \frac{q_m}{m}\delta^2\chi_1^2$.

We next prove the case for $t_0 = 0$. Going back to (G.2), substituting $\gamma = \gamma^\star = n_m - \sqrt{n_m^2 - n_m}$ that minimizes the coefficient in front of $\sum_{t=0}^{T-1} \chi_{t+1}^2$ over $(0,1)$, multiplying both sides by $\frac{q_m}{m}$, substituting the identity $\sigma_t^2 = \frac{q_m}{m} \sigma_{t,1}^2$, we have:

$$\sum_{t=1}^{T} \sigma_t^2 \le \frac{2q_m n_m}{m} \sigma_{1,1}^2 + \frac{2q_m n_m}{m} \mathbf{B}_0^2 + \frac{n_m - 1 + \sqrt{n_m^2 - n_m}}{(n-1)} \sum_{t=1}^{T-1} G_t^2 + 4 n_m^2 \delta_m^2 \sum_{t=2}^{T} \chi_t^2 \ .$$

Let $\alpha$ be such that $(1+\alpha)(1 - 1/n_m) = 1 - \gamma^*/n_m$ as specified in Lemma 4.1. By the definition of $\mathbf{b}_i^0$, we have:

$$\hat{\sigma}_{1,1}^2 = \frac{1}{n} \sum_{i=1}^{n} \|(\nabla f_i(\mathbf{x}^1) - \mathbf{b}_i^0) - (\nabla f(\mathbf{x}^1) - \mathbf{b}^0)\|^2$$

$$\overset{(E.2),(2.2)}{\le} (1+\alpha) \frac{1}{n} \sum_{i=1}^{n} \|(\nabla f_i(\mathbf{x}^0) - \mathbf{b}_i^0) - (\nabla f(\mathbf{x}^0) - \mathbf{b}^0)\|^2 + \left(1 + \frac{1}{\alpha}\right) \delta^2 \hat{\chi}_1^2$$

$$= (1+\alpha) \frac{1}{n} \Big[ \sum_{i \in S_0} \|\nabla f(\mathbf{x}^0) - \nabla f_{S_0}(\mathbf{x}^0)\|^2 + \sum_{i \notin S_0} \|\nabla f_i(\mathbf{x}^0) - \nabla f(\mathbf{x}^0) + \nabla f_{S_0}(\mathbf{x}^0)\|^2 \Big] + \left(1 + \frac{1}{\alpha}\right) \delta^2 \hat{\chi}_1^2$$

$$= (1+\alpha) \frac{1}{n} \Big[ m\|\nabla f(\mathbf{x}^0) - \nabla f_{S_0}(\mathbf{x}^0)\|^2 + \sum_{i \notin S_0} \|\nabla f_i(\mathbf{x}^0) - \nabla f(\mathbf{x}^0)\|^2 + m\langle \nabla f(\mathbf{x}^0), \nabla f_{S_0}(\mathbf{x}^0) \rangle$$

$$+ (n - 2m)\|\nabla f_{S_0}(\mathbf{x}^0)\|^2 \Big] + \left(1 + \frac{1}{\alpha}\right) \delta^2 \hat{\chi}_1^2 \ .$$

It follows that:

$$\sigma_{1,1}^2 = \mathbb{E}_{S_0}[\hat{\sigma}_{1,1}^2]$$

$$\overset{(E.3)}{\le} (1+\alpha) \frac{1}{n} \Big[ q_m \zeta_0^2 + (n-m) \zeta_0^2 + m\|\nabla f(\mathbf{x}^0)\|^2 + (n - 2m)\|\nabla f(\mathbf{x}^0)\|^2 + (n - 2m) \frac{q_m}{m} \zeta_0^2 \Big] + \left(1 + \frac{1}{\alpha}\right) \delta^2 \chi_1^2$$

$$= (1+\alpha) \left(1 + \frac{q_m}{m} - \frac{q_m}{n} - \frac{1}{n_m}\right) \zeta_0^2 + (1+\alpha)(1 - 1/n_m)\|\nabla f(\mathbf{x}^0)\|^2 + \left(1 + \frac{1}{\alpha}\right) \delta^2 \chi_1^2 \ .$$

$$= \frac{n_m - \gamma^*}{n_m - 1} \Big[ (1 - 1/n_m + q_m/m - q_m/n) \zeta_0^2 + (1 - 1/n_m)\|\nabla f(\mathbf{x}^0)\|^2 \Big] + \frac{n_m - \gamma^*}{1 - \gamma^*} \delta^2 \chi_1^2$$

$$\le \sqrt{\frac{n_m}{n_m - 1}} \Big[ (1 - 1/n_m + q_m/m - q_m/n) \zeta_0^2 + (1 - 1/n_m)\|\nabla f(\mathbf{x}^0)\|^2 \Big] + 2 n_m \delta^2 \chi_1^2 \ .$$

By the definition of $\mathbf{B}_0^2$, we get:

$$\mathbf{B}_0^2 = \frac{n_m - \gamma^*}{n_m - 1} \mathbb{E}_{S_0}[\|\mathbf{b}^0\|^2] \overset{(E.3)}{=} \sqrt{\frac{n_m}{n_m - 1}} \Big[ \|\nabla f(\mathbf{x}^0)\|^2 + \frac{q_m}{m} \zeta_0^2 \Big] \ .$$

Substituting the bound of $\sigma_{1,1}^2$ and $\mathbf{B}_0^2$, we have:

$$\sum_{t=1}^{T} \sigma_t^2 \le a_0 \zeta_0^2 + b_0 G_0^2 + \frac{n_m - 1 + \sqrt{n_m^2 - n_m}}{(n-1)} \sum_{t=1}^{T-1} G_t^2 + 4 n_m^2 \delta_m^2 \sum_{t=1}^{T} \chi_t^2 \ ,$$

where $a_0 = \frac{2q_m n_m}{m} \sqrt{\frac{n_m}{n_m - 1}} (1 - 1/n_m + 2q_m/m - q_m/n)$ and $b_0 = \frac{2q_m n_m}{m} \sqrt{\frac{n_m}{n_m - 1}} (2 - 1/n_m)$. Using $\frac{2q_m n_m}{m} \sqrt{\frac{n_m}{n_m - 1}} = \frac{2n}{n-1} \sqrt{n_m^2 - n_m} \le 4\sqrt{n_m^2 - n_m}$ since $n \ge 2$, we have $a_0 \le 12\sqrt{n_m^2 - n_m}$ and $b_0 \le 8\sqrt{n_m^2 - n_m}$. By the definition of $\sigma_0^2$, we have:

$$\sigma_0^2 = \mathbb{E}_{S_0}[\|\mathbf{G}_0 - \nabla f(\mathbf{x}^0)\|^2] = \mathbb{E}_{S_0}[\|\nabla f_{S_0}(\mathbf{x}^0) - \nabla f(\mathbf{x}^0)\|^2] \overset{(E.3)}{=} \frac{q_m}{m} \zeta_0^2 \ .$$

Adding $\sigma_0^2$ to both sides of the previous display, we conclude the proof. $\qquad\square$

We next consider the RG estimator with inexact initialization.

$$\boxed{\mathbf{g}^0 \approx \nabla f(\mathbf{x}^0), \quad \mathbf{g}^{t+1} = (1 - \beta)\mathbf{g}^t + \beta \mathbf{G}^t + \nabla f_{S_t}(\mathbf{x}^{t+1}) - \nabla f_{S_t}(\mathbf{x}^t), \ t \ge 0 \ ,} \tag{G.3}$$

Let us now combine RG (G.3) and SAGA (G.1). They share the same randomness $S_t$ starting from $t \ge 0$.

**Lemma G.2.** *Consider the RG-SAGA estimator (G.3)-(G.1) under Assumptions 5.1 and 2.2. Let $T \geq 1$. Suppose $t_0 = 1$, by setting $\mathbf{g}^0 = \mathbf{G}^0 = \nabla f(\mathbf{x}^0)$, we have:*

$$\sum_{t=0}^{T} \Sigma_t^2 \leq \frac{4\beta n_m q_m}{(2-\beta)m} \|\nabla f(\mathbf{x}^0)\|^2 + \frac{2\beta(n_m - 1 + \sqrt{n_m^2 - n_m})}{(2-\beta)(n-1)} \sum_{t=1}^{T-1} G_t^2 + \frac{8\beta^2 n_m^2 \delta_m^2 + 2\delta_m^2}{2\beta - \beta^2} \sum_{t=1}^{T} \chi_t^2 \ (t_0 = 1) \ ,$$

*If $t_0 = 0$, then by setting $\mathbf{g}^0 = \nabla f_{S_{-1}}(\mathbf{x}^0)$ where $S_{-1} \in \binom{[n]}{m}$ is sampled uniformly at random without replacement, we have:*

$$\sum_{t=0}^{T} \Sigma_t^2 \leq \Big( \big( \frac{(1-\beta)^2}{2\beta - \beta^2} + 1 + \frac{2\beta}{2-\beta} \big) \frac{q_m}{m} + \frac{24\beta\sqrt{n_m^2 - n_m}}{2 - \beta} \Big) \zeta_0^2 + \frac{16\beta}{2-\beta} \sqrt{n_m^2 - n_m} G_0^2$$

$$+ \frac{2\beta(n_m - 1 + \sqrt{n_m^2 - n_m})}{(2-\beta)(n-1)} \sum_{t=1}^{T-1} G_t^2 + \frac{8\beta^2 n_m^2 \delta_m^2 + + 2\delta_m^2}{2\beta - \beta^2} \sum_{t=1}^{T} \chi_t^2 \ ,$$

*where $\Sigma_t^2 := \mathbb{E}_{S_{[t-1]}}[\|\mathbf{g}^t - \nabla f(\mathbf{x}^t)\|^2]$, $G_t^2 := \mathbb{E}_{S_{[t-2]}}[\|\nabla f(\mathbf{x}^t)\|^2]$, $\chi_t^2 := \mathbb{E}_{S_{[t-2]}}[\|\mathbf{x}^t - \mathbf{x}^{t-1}\|^2]$, $S_{[t]} := (S_{-1}, S_0, \ldots, S_t)$ and $\zeta_0^2 := \frac{1}{n} \sum_{i=1}^{n} \|\nabla f_i(\mathbf{x}^0) - \nabla f(\mathbf{x}^0)\|^2$.*

*Proof.* Suppose $t_0 = 1$. Then $\mathbf{g}^0 = \nabla f(\mathbf{x}^0)$. Applying Lemma 5.2 and using the assumption that $\mathbf{x}^t$ is independent of $S_{[t-1]}$, we have for any $T \geq 1$:

$$\sum_{t=0}^{T} \Sigma_t^2 \leq \frac{2\beta}{2-\beta} \sum_{t=0}^{T-1} \sigma_t^2 + \frac{2\delta_m^2}{2\beta - \beta^2} \sum_{t=1}^{T} \chi_t^2 \ .$$

Applying Lemma G.1 with $t_0 = 1$, we have:

$$\sum_{t=0}^{T} \sigma_t^2 \leq \frac{2n_m q_m}{m} \|\nabla f(\mathbf{x}^0)\|^2 + \frac{n_m - 1 + \sqrt{n_m^2 - n_m}}{(n-1)} \sum_{t=1}^{T-1} G_t^2 + 4n_m^2 \delta_m^2 \sum_{t=1}^{T} \chi_t^2 \ .$$

Combining the previous two displays, we get the first claim.

We next prove the case for $t_0 = 0$. Since $\mathbb{E}_{S_t}[\mathbf{G}_t] = \nabla f(\mathbf{x}^t)$ for any $t \geq 0$ and Assumption 5.1 is assumed, inequality (5.1) is thus satisfied. Taking the expectation w.r.t. $S_{-1}$ on both sides of (5.1), we have for any $T \geq 1$:

$$\sum_{t=1}^{T} \Sigma_t^2 \leq \frac{(1-\beta)^2}{2\beta - \beta^2} \Sigma_0^2 + \frac{2\beta}{2-\beta} \sum_{t=0}^{T-1} \sigma_t^2 + \frac{2\delta_m^2}{2\beta - \beta^2} \sum_{t=0}^{T-1} \chi_{t+1}^2$$

Adding $\Sigma_0^2 = \mathbb{E}_{S_{-1}}[\|\mathbf{g}^0 - \nabla f(\mathbf{x}^0)\|^2] = \frac{q_m}{m} \zeta_0^2$ to both sides and applying Lemma G.1 with $t_0 = 0$, we conclude the proof. $\square$

**Theorem G.3.** *Let I-CGM be applied to Problem 1 with RG-SAGA (G.3)-(G.1) estimator under Assumption 2.1 and 2.2. Let $\mathbf{g}^0 = \nabla f(\mathbf{x}^0)$ if $t_0 = 1$ and $\mathbf{g}^0 = \nabla f_{S_{-1}}(\mathbf{x}^0)$ if $t_0 = 0$, where $S_{-1} \in \binom{[n]}{m}$ is uniformly sampled at random without replacement. Suppose the inaccuracies in solving the subproblems satisfy (6). Then by choosing $\lambda = 3\Delta_1 + 113\delta_m/p_B$ and $\beta = \frac{1}{112 n_m}$, the total communication complexity $N(\varepsilon)$ required to find an $\varepsilon$-approximate stationary point is at most:*

$$C_A \lceil n_m \rceil + (C_R + 1) \Big\lceil \frac{256(\Delta_1 + 38\sqrt{n_m}\delta_m) F^0}{\varepsilon^2} + \frac{8q_m G_0^2}{m\varepsilon^2} \Big\rceil \quad (t_0 = 1) \ ,$$

*and*

$$(C_R + 1) \Big\lceil \frac{256(\Delta_1 + 38\sqrt{n_m}\delta_m) F^0}{\varepsilon^2} + (112 q_m/m + 28\sqrt{1 - 1/n_m}) \zeta_0^2 + 16\sqrt{1 - 1/n_m} G_0^2 \Big\rceil \quad (t_0 = 0),$$

*where $G_0^2 := \|\nabla f(\mathbf{x}^0)\|^2$ and $\zeta_0^2 := \frac{1}{n} \sum_{i=1}^{n} \|\nabla f_i(\mathbf{x}^0) - \nabla f(\mathbf{x}^0)\|^2$.*

*Proof.* Let $T \geq 1$. Applying Theorem 3.1, taking the full expectation and using condition (6), we have:

$$\sum_{t=1}^{T} G_t^2 + (\lambda + \Delta_1)^2 \sum_{t=1}^{T} \chi_t^2 \leq \frac{16(\lambda + \Delta_1)^2}{\lambda - \Delta_1} F^0 + \Big( \frac{12(\lambda + \Delta_1)^2}{(\lambda - \Delta_1)^2} + 8 \Big) \sum_{t=0}^{T-1} \Sigma_t^2 \ ,$$

where $G_t^2$, $\chi_t^2$ and $\Sigma_t^2$ are defined in Corollary 3.2. Let us first assume $t_0 = 1$. Applying Lemma G.2 with $t_0 = 1$, we get:

$$\sum_{t=0}^{T} \Sigma_t^2 \leq \frac{4\beta n_m q_m}{(2-\beta)m}\|\nabla f(\mathbf{x}^0)\|^2 + \frac{2\beta(n_m - 1 + \sqrt{n_m^2 - n_m})}{(2-\beta)(n-1)}\sum_{t=1}^{T-1} G_t^2 + \frac{8\beta^2 n_m^2 \delta_m^2 + 2\delta_m^2}{2\beta - \beta^2}\sum_{t=1}^{T} \chi_t^2$$

$$\leq \frac{4\beta n_m q_m}{m} G_0^2 + 4\beta n_m \sum_{t=1}^{T-1} G_t^2 + \Big(8\beta n_m^2 \delta_m^2 + \frac{2\delta_m^2}{\beta}\Big)\sum_{t=1}^{T} \chi_t^2 ,$$

where we used $\frac{1}{2-\beta} \leq 1$, $\frac{q_m}{m} \leq 1$, $\frac{n_m - 1}{n-1} \leq 1 \leq n_m$, and $\frac{\sqrt{n_m^2 - n_m}}{n-1} \leq n_m$ as $n \geq 2$. Using the same choice of parameters as used in Theorem 6.1, it holds that:

$$\Big(\frac{12(\lambda + \Delta_1)^2}{(\lambda - \Delta_1)^2} + 8\Big)4\beta n_m \leq \frac{1}{2}, \ \Big(\frac{12(\lambda + \Delta_1)^2}{(\lambda - \Delta_1)^2} + 8\Big)\Big(8\beta n_m^2 \delta_m^2 + \frac{2\delta_m^2}{\beta}\Big) \leq (\lambda + \Delta_1)^2 .$$

It follows that:

$$\frac{1}{2}\sum_{t=1}^{T} G_t^2 \leq \frac{16(\lambda + \Delta_1)^2}{\lambda - \Delta_1} F^0 + \Big(\frac{12(\lambda + \Delta_1)^2}{(\lambda - \Delta_1)^2} + 8\Big)\frac{4\beta n_m q_m}{m} G_0^2$$

$$\leq 128(\Delta_1 + 38\sqrt{n_m}\delta_m)F^0 + \frac{4q_m}{m} G_0^2 .$$

Therefore, to achieve $\mathbb{E}[\|\nabla f(\bar{\mathbf{x}}^T)\|^2] \leq \varepsilon^2$, we need at most $T = \lceil \frac{256(\Delta_1 + 38\sqrt{n_m}\delta_m)F^0}{\varepsilon^2} + \frac{8q_m G_0^2}{m\varepsilon^2}\rceil$ iterations. We next compute the communication complexity. At the beginning when $t = 0$, we need $\lceil n_m \rceil$ communication rounds with A-CSS to compute one full gradient and the associated local complexity is 1. Additionally, one communication round with D-CSS is needed to compute $\mathbf{x}^1$. For subsequent iterations $t \geq 1$, one communication round with R-CSS is needed for updating $\mathbf{g}^t$. Then another round with D-CSS is required to compute the next iterate. Therefore, the total communication complexity is at most:

$$N(\varepsilon) = \mathbb{E}[C_A N_A + C_R N_R + N_D] \leq C_A \lceil n_m \rceil + C_R T + T .$$

We next consider the case where $t_0 = 0$. We follow the same reasoning strategy as for $t_0 = 1$. Applying Lemma G.2 with $t_0 = 0$, we get:

$$\sum_{t=0}^{T} \Sigma_t^2 \leq \Big(\big(\frac{(1-\beta)^2}{2\beta - \beta^2} + 1 + \frac{2\beta}{2-\beta}\big)\frac{q_m}{m} + \frac{24\beta\sqrt{n_m^2 - n_m}}{2-\beta}\Big)\zeta_0^2 + \frac{16\beta}{2-\beta}\sqrt{n_m^2 - n_m}G_0^2$$

$$+ \frac{2\beta(n_m - 1 + \sqrt{n_m^2 - n_m})}{(2-\beta)(n-1)}\sum_{t=1}^{T-1} G_t^2 + \frac{8\beta^2 n_m^2 \delta_m^2 + +2\delta_m^2}{2\beta - \beta^2}\sum_{t=1}^{T} \chi_t^2$$

$$\leq (4q_m/m + 24\beta\sqrt{n_m^2 - n_m})\zeta_0^2 + 16\beta\sqrt{n_m^2 - n_m}G_0^2 + 4\beta n_m \sum_{t=1}^{T-1} G_t^2 + \Big(8\beta n_m^2 \delta_m^2 + \frac{2\delta_m^2}{\beta}\Big)\sum_{t=1}^{T} \chi_t^2 .$$

Using the same choice of parameters as used in Theorem 6.1, it follows that:

$$\frac{1}{2}\sum_{t=1}^{T} G_t^2 \leq \frac{16(\lambda + \Delta_1)^2}{\lambda - \Delta_1} F^0 + \Big(\frac{12(\lambda + \Delta_1)^2}{(\lambda - \Delta_1)^2} + 8\Big)\big((4q_m/m + 24\beta\sqrt{n_m^2 - n_m})\zeta_0^2 + 16\beta\sqrt{n_m^2 - n_m}G_0^2\big)$$

$$\leq 128(\Delta_1 + 38\sqrt{n_m}\delta_m)F^0 + \big(56q_m/m + 14\sqrt{1 - 1/n_m}\big)\zeta_0^2 + 8\sqrt{1 - 1/n_m}G_0^2 .$$

Therefore, to achieve $\mathbb{E}[\|\nabla f(\bar{\mathbf{x}}^T)\|^2] \leq \varepsilon^2$, we need at most $T = \lceil \frac{256(\Delta_1 + 38\sqrt{n_m}\delta_m)F^0}{\varepsilon^2} + (112q_m/m + 28\sqrt{1 - 1/n_m})\zeta_0^2 + 16\sqrt{1 - 1/n_m}G_0^2\rceil$ iterations. We next compute the communication complexity. At the beginning when $t = 0$, we need one communication round with R-CSS to compute $\mathbf{g}^0 = \nabla f_{S_{-1}}(\mathbf{x}^0)$ and the associated local complexity is 1. Additionally, one communication round with D-CSS is needed to compute $\mathbf{x}^1$. For subsequent iterations $t \geq 1$, one communication round with R-CSS is needed for updating $\mathbf{g}^t$. Then another round with D-CSS is required to compute the next iterate. Therefore, the total communication complexity is at most:

$$N(\varepsilon) = \mathbb{E}[C_A N_A + C_R N_R + N_D] \leq C_R T + T . \qquad \square$$

**Summary**. The communication complexity of I-CGM-RG-SAGA with one full gradient computation ($t_0 = 1$) is $N(\varepsilon) \lesssim C_A n_m + C_R \frac{(\Delta_1 + \sqrt{n_m}\delta_m)F^0}{\varepsilon^2} + C_R \frac{q_m G_0^2}{m\varepsilon^2}$. Compared to the case where $t_0 = 2$, this complexity has an additional error term depending on $\|\nabla f(\mathbf{x}^0)\|^2$. Furthermore, if $t_0 = 0$ (no full synchonization is needed), the resulting communication complexity is $N(\varepsilon) \lesssim C_R \frac{(\Delta_1 + \sqrt{n_m}\delta_m)F^0}{\varepsilon^2} + C_R(q_m/m + \sqrt{1 - n_m})\frac{\zeta_0^2}{\varepsilon^2} + C_R\sqrt{1 - n_m}G_0^2$. This quantity does not depend on $C_A$ but has two additional error terms depending on $\frac{1}{n}\sum_{i=1}^{n}\|\nabla f_i(\mathbf{x}^0) - \nabla f(\mathbf{x}^0)\|^2$ and $\|\nabla f(\mathbf{x}^0)\|^2$, due to inexact initialization. Note that when $m \to n$, these error terms eventually disappear.

## H    SOLVING THE SUBPROBLEMS WITH LOCAL STOCHASTIC CGM

In this section, we discuss how to achieve the inaccuracy condition (6) by running stochastic CGM locally.

**Lemma H.1** (Stochastic composite gradient method). *Consider the composite problem:*

$$\min_{\mathbf{x}\in\mathbb{R}^d}\big\{F(\mathbf{x}) := \phi(\mathbf{x}) + \psi(\mathbf{x})\big\} ,$$

*where $\phi$ is $L_\phi$-smooth and $\psi(\mathbf{x}) := \frac{\lambda_\psi}{2}\|\mathbf{x} - \tilde{\mathbf{x}}\|^2$ where $\tilde{\mathbf{x}} \in \mathbb{R}^d$ is a fixed point and $\lambda_\psi \geq 0$. Suppose we have access to an unbiased stochastic gradient oracle $\mathbf{g}^\phi$ for $\nabla\phi$ such that:*

$$\mathbb{E}_\zeta[\mathbf{g}^\phi(\mathbf{x};\zeta)] = \nabla\phi(\mathbf{x}), \quad \mathbb{E}_\zeta[\|\mathbf{g}^\phi(\mathbf{x};\zeta) - \nabla\phi(\mathbf{x})\|^2] \leq \sigma^2, \quad \forall \mathbf{x} \in \mathbb{R}^d .$$

*Consider the stochastic composite gradient method:*

$$\mathbf{x}_{k+1} = \arg\min_{\mathbf{x}\in\mathbb{R}^d}\big\{L_k(\mathbf{x}) := \phi(\mathbf{x}_k) + \langle\mathbf{g}^\phi(\mathbf{x}_k;\zeta_k),\mathbf{x}-\mathbf{x}_k\rangle + \frac{\eta}{2}\|\mathbf{x}-\mathbf{x}_k\|^2 + \psi(\mathbf{x})\big\} .$$

*Let $\eta \geq L_\phi$ and $\hat{K} \sim \mathrm{Geom}(p)$ with $p \in (0,1]$. Then we have:*

$$(1-p)\,\mathbb{E}\big[\|\nabla F(\mathbf{x}_{\hat{K}+1})\|^2\big] + p\|\nabla F(\mathbf{x}_0)\|^2 \leq \frac{2(\eta+\lambda_\psi)^2 p}{2\eta - L_\phi + \lambda_\psi}\big[F(\mathbf{x}^0) - \mathbb{E}[F(\mathbf{x}_{\hat{K}+1})]\big] + \frac{L_\phi + \lambda_\psi}{2\eta - L_\phi + \lambda_\psi}\sigma^2 .$$

*Proof.* Let $k \geq 0$ and denote $\mathbf{g}_k := \mathbf{g}^\phi(\mathbf{x}_k;\zeta_k)$. By $(\eta+\lambda_\psi)$-strong convexity of $L_k$, for any $\mathbf{x} \in \mathbb{R}^d$, we have:

$$L_k(\mathbf{x}) \geq L_k(\mathbf{x}_{k+1}) + \frac{\eta+\lambda_\psi}{2}\|\mathbf{x} - \mathbf{x}_{k+1}\|^2 .$$

Substituting $\mathbf{x} = \mathbf{x}_k$, it follows that,

$$F(\mathbf{x}_k) \geq \phi(\mathbf{x}_k) + \langle\mathbf{g}_k,\mathbf{x}_{k+1}-\mathbf{x}_k\rangle + \frac{\eta}{2}\|\mathbf{x}_{k+1}-\mathbf{x}_k\|^2 + \psi(\mathbf{x}_{k+1}) + \frac{\eta+\lambda_\psi}{2}\|\mathbf{x}_{k+1}-\mathbf{x}_k\|^2$$

$$\geq \phi(\mathbf{x}_{k+1}) + \psi(\mathbf{x}_{k+1}) + \frac{2\eta - L_\phi + \lambda_\psi}{2}\|\mathbf{x}_{k+1}-\mathbf{x}_k\|^2 + \langle\mathbf{g}_k - \nabla\phi(\mathbf{x}_k),\mathbf{x}_{k+1}-\mathbf{x}_k\rangle$$

$$= F(\mathbf{x}_{k+1}) + \frac{2\eta - L_\phi + \lambda_\psi}{2}\|\mathbf{x}_{k+1}-\mathbf{x}_k\|^2 + \langle\mathbf{g}_k - \nabla\phi(\mathbf{x}_k),\mathbf{x}_{k+1}-\mathbf{x}_k\rangle .$$

Let $\eta \geq L_\phi$. By the definition of $\mathbf{x}_{k+1}$, we get:

$$\mathbf{g}_k + \eta(\mathbf{x}_{k+1}-\mathbf{x}_k) + \lambda_\psi(\mathbf{x}_{k+1}-\tilde{\mathbf{x}}) = 0 \;\Rightarrow\; \mathbf{x}_{k+1}-\mathbf{x}_k = \frac{1}{\eta+\lambda_\psi}(-\nabla\psi(\mathbf{x}_k) - \mathbf{g}_k) .$$

It follows that:

$$\langle\mathbf{g}_k - \nabla\phi(\mathbf{x}_k),\mathbf{x}_{k+1}-\mathbf{x}_k\rangle = \frac{1}{\eta+\lambda_\psi}\langle\mathbf{g}_k - \nabla\phi(\mathbf{x}_k), -\nabla F(\mathbf{x}_k) - (\mathbf{g}_k - \nabla\phi(\mathbf{x}_k))\rangle ,$$

and that:

$$\|\mathbf{x}_{k+1}-\mathbf{x}_k\|^2 = \frac{\|\mathbf{g}_k - \nabla\phi(\mathbf{x}_k) + \nabla F(\mathbf{x}_k)\|^2}{(\eta+\lambda_\psi)^2} .$$

Substituting these identities into the second display and taking the expectation w.r.t. $\zeta_k$, we have:

$$F(\mathbf{x}_k) \geq \mathbb{E}_{\zeta_k}[F(\mathbf{x}_{k+1})] + \Big(\frac{2\eta - L_\phi + \lambda_\psi}{2(\eta+\lambda_\psi)^2} - \frac{1}{\eta+\lambda_\psi}\Big)\mathbb{E}_{\zeta_k}[\|\mathbf{g}_k - \nabla\phi(\mathbf{x}_k)\|^2] + \frac{2\eta - L_\phi + \lambda_\psi}{2(\eta+\lambda_\psi)^2}\|\nabla F(\mathbf{x}_k)\|^2$$

$$\geq \mathbb{E}_{\zeta_k}[F(\mathbf{x}_{k+1})] + \frac{2\eta - L_\phi + \lambda_\psi}{2(\eta+\lambda_\psi)^2}\|\nabla F(\mathbf{x}_k)\|^2 - \frac{L_\phi + \lambda_\psi}{2(\eta+\lambda_\psi)^2}\sigma^2 .$$

Let $\zeta_{[k]} := (\zeta_0,\ldots,\zeta_k)$. Taking the expectation w.r.t. $\zeta_{[k-1]}$ and rearranging, we get:

$$\mathbb{E}_{\zeta_{[k-1]}}[\|\nabla F(\mathbf{x}_k)\|^2] \leq \frac{2(\eta+\lambda_\psi)^2}{2\eta - L_\phi + \lambda_\psi}\big(\mathbb{E}_{\zeta_{[k-1]}}[F(\mathbf{x}_k)] - \mathbb{E}_{\zeta_{[k]}}[F(\mathbf{x}_{k+1})]\big) + \frac{L_\phi + \lambda_\psi}{2\eta - L_\phi + \lambda_\psi}\sigma^2 .$$

Substituting $k = \hat{K}$ with $\hat{K} \sim \mathrm{Geom}(p)$, taking the expectation w.r.t. $\hat{K}$ and applying Lemma E.3, we have:

$$\mathbb{E}_{\hat{K},\zeta_{[\hat{K}-1]}}[\|\nabla F(\mathbf{x}_{\hat{K}})\|^2] = (1-p)\,\mathbb{E}_{\hat{K},\zeta_{[\hat{K}-1]}}[\|\nabla F(\mathbf{x}_{\hat{K}+1})\|^2] + p\|\nabla F(\mathbf{x}_0)\|^2$$

$$\leq \frac{2(\eta+\lambda_\psi)^2}{2\eta - L_\phi + \lambda_\psi}\big(\mathbb{E}_{\hat{K},\zeta_{[\hat{K}-1]}}[F(\mathbf{x}_{\hat{K}})] - \mathbb{E}_{\hat{K},\zeta_{[\hat{K}]}}[F(\mathbf{x}_{\hat{K}+1})]\big) + \frac{L_\phi + \lambda_\psi}{2\eta - L_\phi + \lambda_\psi}\sigma^2 .$$

$$\leq \frac{2(\eta+\lambda_\psi)^2}{2\eta - L_\phi + \lambda_\psi}\big((1-p)\,\mathbb{E}_{\hat{K},\zeta_{[\hat{K}-1]}}[F(\mathbf{x}_{\hat{K}+1})] + pF(\mathbf{x}^0) - \mathbb{E}_{\hat{K},\zeta_{[\hat{K}]}}[F(\mathbf{x}_{\hat{K}+1})]\big) + \frac{L_\phi + \lambda_\psi}{2\eta - L_\phi + \lambda_\psi}\sigma^2 .$$

Taking the full expectation on both sides, we get the claim. $\qquad\square$

Let us now apply SCGM to solve the subproblem of (I-CGM) with $\phi(\mathbf{x}) = f_1(\mathbf{x})$ and $\psi_t(\mathbf{x}) = \langle \mathbf{g}^t - \nabla f_1(\mathbf{x}^t), \mathbf{x} - \mathbf{x}^t \rangle + \frac{\lambda}{2} \|\mathbf{x} - \mathbf{x}^t\|^2$. For $k = 0, 1, \ldots, K_t - 1$,

$$
\begin{aligned}
\mathbf{y}_{k+1}^t &= \arg\min_{\mathbf{y} \in \mathbb{R}^d} \left\{ \phi(\mathbf{y}_k^t) + \langle \mathbf{g}^\phi(\mathbf{y}_k^t), \mathbf{y} - \mathbf{y}_k^t \rangle + \frac{\eta}{2} \|\mathbf{y} - \mathbf{y}_k^t\|^2 + \psi_t(\mathbf{y}) \right\} \\
&= \frac{1}{\lambda + \eta} \left( \eta \mathbf{y}_k^t + \lambda \mathbf{x}^t + \nabla f_1(\mathbf{x}^t) - \mathbf{g}^t - \mathbf{g}_1(\mathbf{y}_k^t) \right),
\end{aligned}
\tag{H.1}
$$

where $\mathbf{g}_1$ is the unbiased gradient estimator of $\nabla f_1$ with bounded variance $\sigma^2$, $K_t = \hat{K}_t + 1$ where $\hat{K}_t \sim$ Geom$(p)$. The solution is set to be $\mathbf{x}^{t+1} = \mathbf{y}_{K_t}$. We use the notation $\mathbf{x}^{t+1} = \text{SCGM}_{\text{rand}}(\lambda, \eta, \hat{K}_t, \mathbf{x}^t, \mathbf{g}^t)$ for this process.

**Lemma H.2.** *Consider I-CGM with* $\mathbf{x}^{t+1} = \text{SCGM}_{\text{rand}}(\lambda, \eta, \hat{K}_t, \mathbf{x}^t, \mathbf{g}^t)$ *where* $\hat{K}_t \sim$ Geom$(p)$ *under Assumption 2.1 and 2.3. Let* $T \geq 1$ *be the fixed number in condition* (6)*. Then by choosing* $\lambda > \Delta_1$, $\eta = L_1 + \frac{2(\lambda + L_1)\sigma^2}{\varepsilon^2}$, $p = \frac{\lambda - \Delta_1}{2(\eta + \lambda)} < 1$, *the accuracy condition* $\sum_{t=0}^{T-1} \mathbb{E}[e_t^2] \leq \frac{(\lambda + \Delta_1)^2 F^0}{\lambda - \Delta_1} + \sum_{t=0}^{T-1} \Sigma_t^2 + \frac{T\varepsilon^2}{2}$ *is satisfied where* $\varepsilon$ *is the target accuracy for achiving* $\mathbb{E}[\|\nabla f(\bar{\mathbf{x}}^T)\|^2] \leq \varepsilon^2$.

*Proof.* Applying Lemma H.1 and Lemma F.1, we have for any $t \geq 0$:

$$
\begin{aligned}
(1-p)\mathbb{E}[e_t^2] + p\mathbb{E}[e_{t-1}^2] &\leq \frac{2(\eta+\lambda)^2 p}{2\eta - L_1 + \lambda} \mathbb{E}[F_t(\mathbf{x}^t) - F_t(\mathbf{x}^{t+1})] + \frac{L_1 + \lambda}{2\eta - L_1 + \lambda}\sigma^2 \\
&\leq \frac{2(\eta+\lambda)^2 p}{2\eta - L_1 + \lambda} \left( \mathbb{E}[f(\mathbf{x}^t) - f(\mathbf{x}^{t+1})] + \frac{\Sigma_t^2}{2(\lambda - \Delta_1)} \right) + \frac{L_1 + \lambda}{2\eta - L_1 + \lambda}\sigma^2.
\end{aligned}
$$

where $e_{-1}^2 = \|\nabla F_0(\mathbf{x}^0)\|^2$. Summing up from $t = 0$ to $t = T$ and dropping the non-negative $\mathbb{E}[e_T^2]$ and $e_{-1}^2$, we have:

$$
\begin{aligned}
\sum_{t=0}^{T-1} \mathbb{E}[e_t^2] &\leq \frac{2(\eta+\lambda)^2 p}{2\eta - L_1 + \lambda} \left( f(\mathbf{x}^0) - f^\star + \sum_{t=0}^T \frac{\Sigma_t^2}{2(\lambda - \Delta_1)} \right) + \frac{L_1 + \lambda}{2\eta - L_1 + \lambda} T\sigma^2 \\
&\leq 2(\eta+\lambda)p(f(\mathbf{x}^0) - f^\star) + \frac{(\eta+\lambda)p}{\lambda - \Delta_1}\sum_{t=0}^T \Sigma_t^2 + \frac{L_1 + \lambda}{\eta + \lambda} T\sigma^2.
\end{aligned}
$$

By the choice of $\eta$ and $\lambda$, we have:

$$
2(\eta+\lambda)p = \lambda - \Delta_1 \leq \frac{(\lambda + \Delta_1)^2}{\lambda - \Delta_1}, \quad \frac{\eta + \lambda}{\lambda - \Delta_1}p = \frac{1}{2} < 1, \quad \frac{L_1 + \lambda}{\eta + \lambda}T\sigma^2 \leq \frac{L_1 + \lambda}{(L_1 + \lambda)2\sigma^2/\varepsilon^2}T\sigma^2 = T\varepsilon^2/2.
$$
$\square$

To ensure convergence of I-CGM, the expected number of local steps by using stochastic CGM is thus $\frac{1}{p} \simeq \frac{L_1}{\lambda} + \frac{\lambda + L_1}{\lambda\varepsilon^2}\sigma^2$. When $\sigma^2 \to 0$, it recovers the result of deterministic CGM with randomized local steps.

# I  DISCUSSION ON THE SAG ESTIMATOR

SAG is another incremental gradient method (Schmidt et al., 2017). SCAFFOLD has successfully applied it to the FL settings. Specifically, the local update rule of device 1 at outer iteration $t$ (assuming no stochasticity for simplicity) is:

$$
\mathbf{y}_{k+1}^t = \mathbf{y}_k^t - \frac{1}{\eta}\left( \nabla f_1(\mathbf{y}_k^t) + \mathbf{b}^t - \nabla f_1(\mathbf{x}^t) \right).
$$

Compared with the local CGM (8), Scaffold sets $\lambda = 0$ and uses $\mathbf{b}^t$ (10)(SAG) instead of $\mathbf{G}^t$ (SAGA) in the control variate. we next show that the variance of $\mathbf{b}^t$ cannot be controlled by $\delta$. Let $n = 2$, $t = 1$, $\mathbf{b}_1^0 = \nabla f_1(\mathbf{x}^0)$ and $\mathbf{b}_2^0 = \nabla f_2(\mathbf{x}^0)$. Then we get: $\mathbf{b}^1 = \frac{1}{2}(\nabla f_1(\mathbf{x}^1) + \nabla f_2(\mathbf{x}^0))$, if $S_1 = \{1\}$ and $\mathbf{b}^1 = \frac{1}{2}(\nabla f_2(\mathbf{x}^1) + \nabla f_1(\mathbf{x}^0))$, if $S_1 = \{2\}$. Then the variance can be computed as:

$$
\mathbb{E}_{S_1}\left[ \|\mathbf{b}^1 - \nabla f(\mathbf{x}^1)\|^2 \right] = \frac{1}{8}\sum_{i=1}^2 \|\nabla f_i(\mathbf{x}^0) - \nabla f_i(\mathbf{x}^1)\|^2.
$$

While for SAGA, we have:

$$
\mathbb{E}_{S_1}\left[ \|\mathbf{G}^1 - \nabla f(\mathbf{x}^1)\|^2 \right] = \frac{1}{2}\sum_{i=1}^2 \|\nabla h_i(\mathbf{x}^0) - \nabla h_i(\mathbf{x}^1)\|^2,
$$

where $h_i := f - f_i$. Therefore, the SAG estimator cannot fully exploit functional similarity as efficiently as SAGA in the worse case from a theoretical perspective. Nevertheless, SCAFFOLD can still perform well empirically on some problems, as shown in Figure J.7.

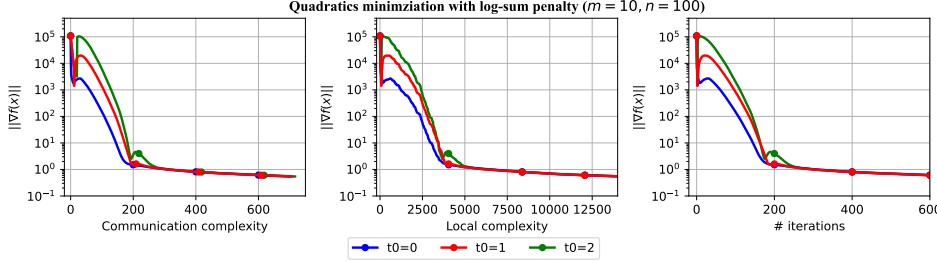

Figure J.1: Comparisons of different initialization strategies of I-CGM-RG-SAGA for solving the quadratic minimization problems with non-convex log-sum penalty.

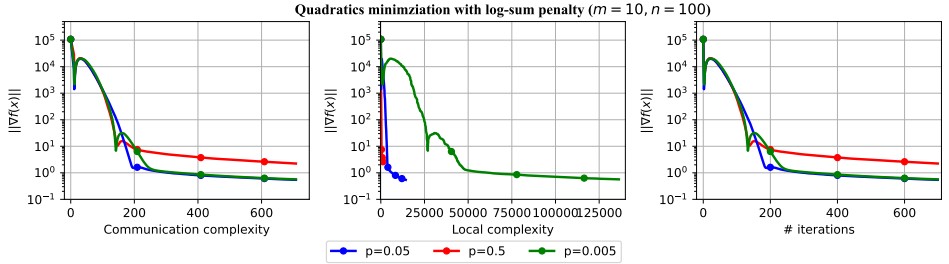

Figure J.2: Comparisons of different $p$ (number of local steps) used in local CGM for I-CGM-RG-SAGA when solving the quadratic minimization problems with non-convex log-sum penalty.

## J    ADDITIONAL DETAILS AND EXPERIMENTS

We simulate the deep learning experiments on one NVIDIA DGX A100. All the other experiments are run on a MacBook Pro laptop.

### J.1    QUADRATIC MINIMIZATION WITH LOG-SUM PENALTY.

Everywhere in the paper, we use the first choice of the control variate for SCAFFOLD (Karimireddy et al., 2020). We set the number of local steps $K$ to be 20 and the local learning rate to be 0.003 (0.005 diverges at the beginning) for FEDAVG and SCAFFOLD. For SABER-FULL, we use the standard gradient method as the local solver and set $K$ to be 20, local learning rate to be 0.005 and the probability for computing the full gradient to be 0.1, matching I-CGM-RG-SVRG. For GD, we run $14000 = 20 * 700$ iterations to match the local gradient computations of other algorithms. Finally, the comparisons of different initialization strategies for I-CGM-RG-SAGA can be found in Figure J.1 ($t_0 = 0, 1, 2$ correspond to computing the full gradient $0, 1, 2$ times at the beginning).

#### J.1.1    ABLATION STUDIES OF I-CGM-RG-SAGA

**Initialization strategies**. The comparisons of different initialization strategies for I-CGM-RG-SAGA can be found in Figure J.1 ($t_0 = 0, 1, 2$ correspond to computing the full gradient $0, 1, 2$ times at the beginning. See Section G for the details). The result shows that the method works well without any full gradient computations.

**Local steps**. We now compare the performance of I-CGM-RG-SAGA under different choices of the parameter $p$, which is defined in Local CGM (8). Theoretically, $p \simeq \frac{\lambda}{\lambda + L_1}$. Since the expected number of local steps per iteration is $\frac{1}{p}$, a smaller $p$ corresponds to more local computations. In the previous experiments, we used the default value $p = \frac{\delta}{L_1} \approx \frac{5}{100} = 0.05$. We now vary $p \in \{0.5, 0.05, 0.005\}$. From Figure J.2, we observe that 1) Large $p = 0.5$ results in worse communication complexity since the local accuracy condition is not fully satisfied; 2) Small $p = 0.005$ achieves similar performance to $p = 0.05$ in terms of communication complexity. This is expected, since communication complexity is determined by the fixed parameter $\lambda$. However, the local complexity becomes worse, as the total number of local steps increase and becomes unnecessarily large.

**Constant $\lambda$**. We now study the impact of the constant $\lambda$ on the performance of I-CGM-RG-SAGA. Note that $\lambda$ directly determines the iteration complexity. Theoretically the best $\lambda \simeq \Delta_1 + \sqrt{n_m}\delta$. In the previous experiments, we used the default value $\lambda = \sqrt{n_m}\delta \approx 15$. We now vary $\lambda \in \{1, 10, 100\}$. From Figure J.3,

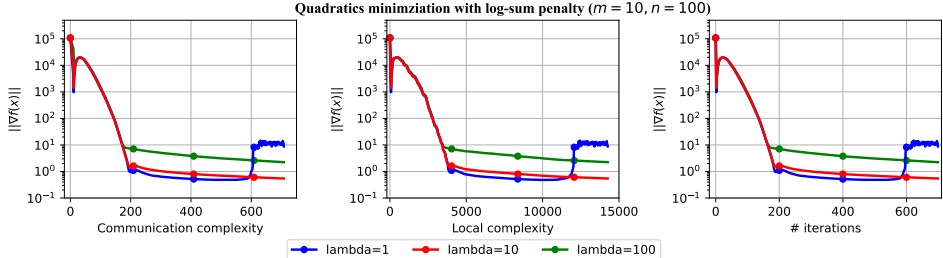

Figure J.3: Comparisons of different $\lambda$ used I-CGM-RG-SAGA for solving the quadratic minimization problems with non-convex log-sum penalty.

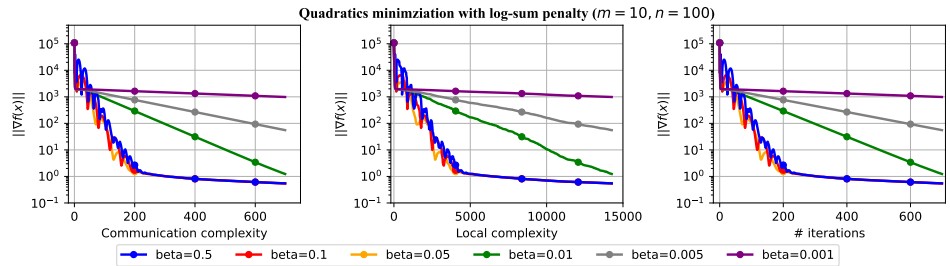

Figure J.4: Comparisons of different $\beta$ used I-CGM-RG-SAGA for solving the quadratic minimization problems with non-convex log-sum penalty.

we observe that: 1) Large $\lambda = 100$ results in worse communication complexity since it does not fully use the similarity structure; 2) Small $\lambda = 1$ does not converge as the theory requires $\lambda \gtrsim \Delta_1 + \sqrt{n_m}\delta_m$, all matching the theory.

**Constant $\beta$.** We now test the effect of $\beta$ used in the RG estimator. Both larger or smaller $\beta$ can theoretically increase the variance bound (Lemma 5.2). Theoretically, the best $\beta \simeq \frac{1}{n_m}$. We now vary $\beta \in \{0.5, 0.1, 0.05, 0.01, 0.005, 0.001\}$. From Figure J.4, we see that $\beta \in [0.05, 0.5]$ results in relatively better performance as $\frac{1}{n_m} = 0.1$ and the values that fall outside this range lead to worse communication complexity.

**Ratio of $\frac{C_A}{C_R}$.** In the main text, we report results under the extreme setting where $C_A = C_R = 1$. Now we test how increasing the ratio $C_A/C_R$ affects the performance. Specifically, we vary $C_A \in \{1, 5, 10, 20\}$ while keeping $C_R = 1$, and repeat the same experiments. From Figure J.5, we observe that the performance of I-CGM-RG-SVRG degrades as $C_A$ increases since each use of A-CSS becomes more costly. In contrast, I-CGM-RG-SAGA remains largely unaffected, as ASS is only used during initialization. This result further confirms the advantage of I-CGM-RG-SAGA in settings where full synchronization is costly.

**Ratio $\frac{n}{m}$.** Finally, we examine how the ratio $\frac{n}{m}$ influences the performance of our method. Theoretically, both the communication and local complexities scale with $\sqrt{n_m}\delta_m F^0/\varepsilon^2$. We fix $m = 1$ and vary $n \in \{10, 100, 1000\}$. The datasets are generated in a consistent manner so that the values of $\delta$ and $L_{\max}$ remain approximately unchanged. We set $\lambda = \sqrt{n_m}\delta \approx 5\sqrt{n_m}$, $\beta = \frac{1}{n_m}$ and $p = \frac{\lambda}{\lambda + L_{\max}} \approx \frac{\lambda}{\lambda + 100}$ with $n_m = n$. From Figure J.6, we observe that increasing $n_m$ indeed leads to higher communication complexity. However, the growth is moderate: the additional cost scales by roughly $\sqrt{100}/\sqrt{10} = \sqrt{1000}/\sqrt{100} \approx 3$ rather than linearly $100/10 = 100/10 = 10$, confirming that the dependence is on $\sqrt{n_m}$ instead of $n_m$.

## J.2 LOGISTIC REGRESSION WITH NONCONVEX REGULARIZER.

For both datasets, we set $p = 0.1$ in Local GD for CGM-RG methods and SCAFFNEW, and use $K = 10$ local steps for the other algorithms. We select the best local learning rate for each method from $\{0.1, 0.2, 0.5, 1.0\}$ for Mushroom and $\{0.002, 0.001, 0.0005\}$ for Duke. For proximal-point methods, we choose the best $\lambda$ from $\{10, 1, 0.1, 0.01\}$ on both datasets. We use $\beta = \frac{m}{n}$ for both I-CGM-RG methods.

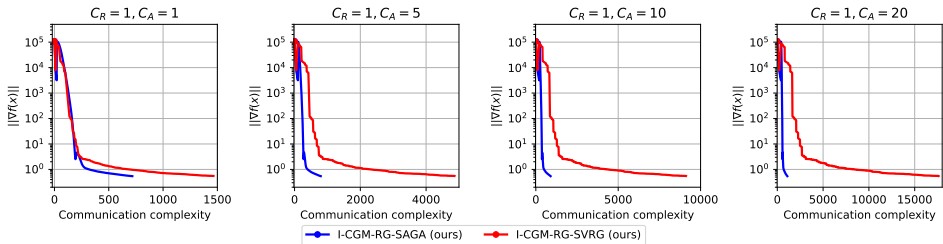

Figure J.5: Comparisons of I-CGM-RG-SAGA against I-CGM-RG-SVRG under different $C_A/C_R$ for solving the quadratic minimization problems with non-convex log-sum penalty.

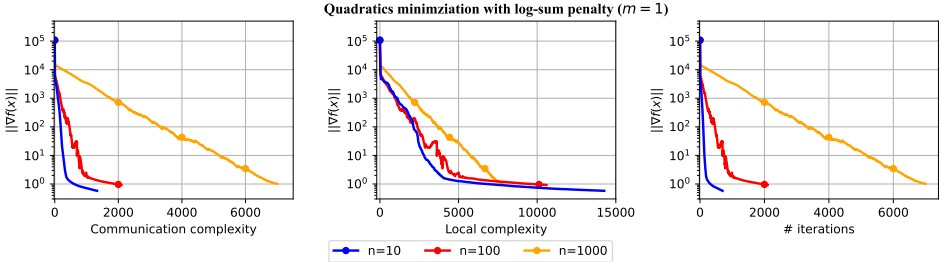

Figure J.6: Comparisons of I-CGM-RG-SAGA under different $n_m$ for solving the quadratic minimization problems with non-convex log-sum penalty.

## J.3 EMNIST WITH RESIDUAL CNN

We now extend our study to neural network training. Specifically, we train a 6-layer Residual CNN on the EMNIST dataset (Cohen et al., 2017), which consists of a collection of 26 letter classes. We use $n = 26$ and $m = 5 \approx \sqrt{n}$, and split the dataset according to the Dirichlet distribution with $\alpha = 0.1$ (the smaller the $\alpha$, the higher the heterogeneity, $\alpha = 0.1$ is highly heterogeneous). We use a batch size of 128 for computing both the local stochastic gradient and the control variates. For all the methods that use control variates, including I-CGM-RG, SCAFFOLD, SABER and SCAFFNEW, we add a damping factor $q$ in front of the control variate to enhance their empirical performance, i.e., on line 5 of Algorithm 8, we use $\mathbf{y}_{k+1}^t = \arg\min_{\mathbf{y} \in \mathbb{R}^d} \left\{ f_1(\mathbf{y}_k^t) + \langle \mathbf{g}_1(\mathbf{y}_k^t) + q(\mathbf{g}^t - \mathbf{g}_1(\mathbf{x}^t)), \mathbf{y} - \mathbf{y}_k^t \rangle + \frac{\eta}{2} \left\| \mathbf{y} - \mathbf{y}_k^t \right\|^2 + \frac{\lambda}{2} \|\mathbf{y} - \mathbf{x}^t\|^2 \right\}$, where $q \in (0, 1]$ is a tuned parameter and $\mathbf{g}_1$ is the stochastic mini-batch gradient of $\nabla f_1$. This approach is suggested by Yin et al. (2025). We report the best local stepsize $\frac{1}{\eta}$ among $\{0.05, 0.02, 0.01, 0.001\}$ and the best $\lambda$ among $\{0.001, 0.01, 0.1, 1\}$. The final choices of the parameters can be found in Table J.1. The convergence behaviours can be found in Figure J.7. The best validation accuracy can be found in Table J.2, where I-CGM-RG-SAGA performs the best.

| optimizers | hyper-parameters used for multi-classification tasks |
|---|---|
| I-CGM-RG-SAGA | $\frac{1}{\eta} = 0.02$, $\lambda = 0.01$, $p = 0.01$, $\beta = 0.2$, $q = 0.001$, $t_0 = 0$ |
| I-CGM-RG-SVRG | $\frac{1}{\eta} = 0.02$, $\lambda = 0.01$, $p = 0.01$, $\beta = 0.2$, $q = 0.001$ |
| SCAFFOLD (Karimireddy et al., 2020) | $\frac{1}{\eta} = 0.02$, $K = 100$, $q = 0.001$ |
| FEDAVG (McMahan et al., 2017) | $\frac{1}{\eta} = 0.02$, $K = 100$ |
| SCAFFNEW (Mishchenko et al., 2022) | $\frac{1}{\eta} = 0.02$, $p = 0.01$, $q = 0.001$ |
| SABER (Mishchenko et al., 2024) | $\frac{1}{\eta} = 0.02$, $\lambda = 0.01$, $p = 0.01$, $\beta = 0.2$, $q = 0.001$ |

Table J.1: Hyper-parameters of the considered optimizers used in the multi-classification task for the EMNIST dataset.

| Optimizers | I-CGM-RG-SAGA | I-CGM-RG-SVRG | SABER-FULL | SCAFFOLD | SCAFFNEW | FEDAVG |
|---|---|---|---|---|---|---|
| Accuracy | **86.2** | 86.0 | 85.3 | 85.9 | 84.9 | 85.6 |

Table J.2: Comparisons of validation accuracy for different optimizers used in the multi-classification task for the EMNIST dataset within 100 outer iterations.

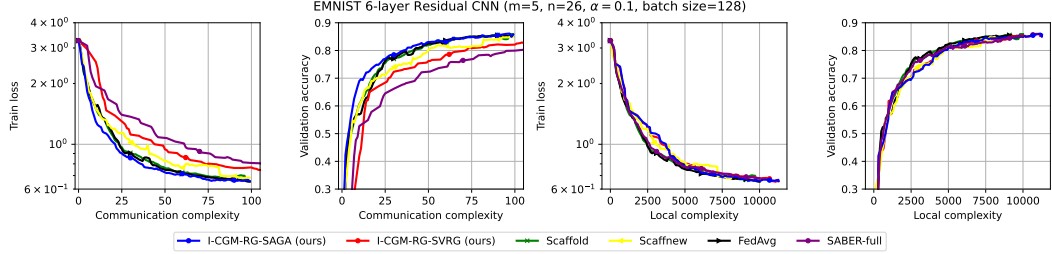

Figure J.7: Comparisons of different algorithms on the EMNIST dataset using a 6-layer residual CNN.

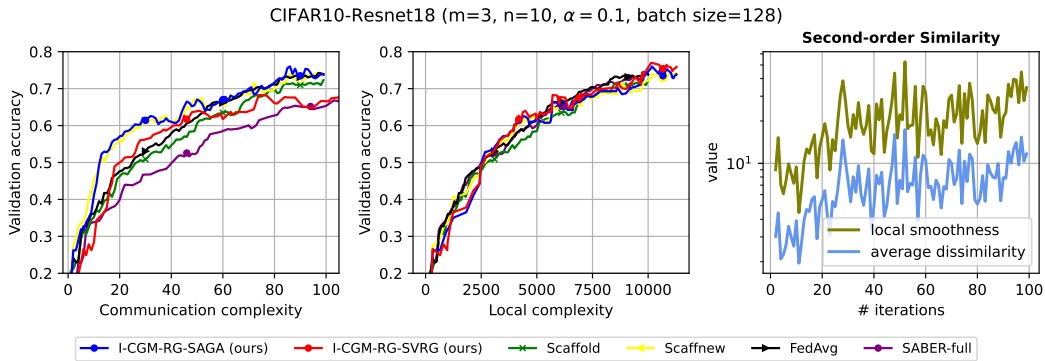

Figure J.8: Comparisons of different algorithms on the CIFAR10 dataset using ResNet18.

## J.4    CIFAR10 WITH RESNET18

We now consider multi-class classification tasks with CIFAR10 (Krizhevsky et al.) using ResNet18 (He et al., 2016). We use $n = 10$ and $m = 3 \approx \sqrt{n}$, and split the dataset according to the Dirichlet distribution with $\alpha = 0.1$ (highly heterogeneous). We use a batch size of 128 for computing both the local stochastic gradient and the control variates $\mathbf{m}$. We report the best local stepsize $\frac{1}{\eta}$ among $\{0.1, 0.05, 0.01, 0.001\}$ and the best $\lambda$ among $\{0.001, 0.01, 0.1, 1\}$. The final choices of the parameters can be found in Table J.3. The convergence behaviours can be found in Figure J.8. The best validation accuracy within 100 outer iterations can be found in Table J.4.

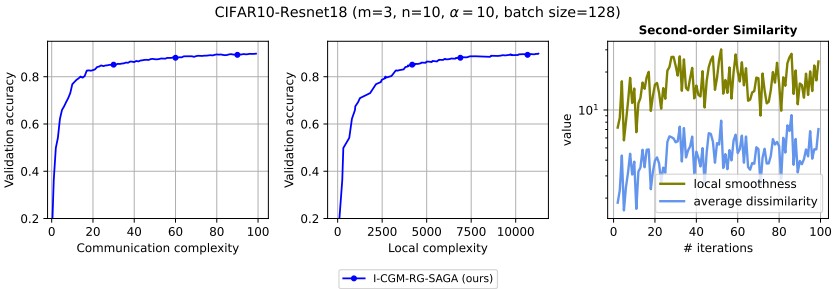

Figure J.9: Performance of I-CGM-RG-SAGA on the CIFAR10 dataset using ResNet18.

| optimizers | hyper-parameters used for multi-classification tasks |
|---|---|
| I-CGM-RG-SAGA | $\frac{1}{\eta} = 0.05$, $\lambda = 0.01$, $p = 0.01$, $\beta = 0.2$, $q = 0.001$, $t_0 = 0$ |
| I-CGM-RG-SVRG | $\frac{1}{\eta} = 0.05$, $\lambda = 0.01$, $p = 0.01$, $\beta = 0.2$, $q = 0.001$ |
| SCAFFOLD (Karimireddy et al., 2020) | $\frac{1}{\eta} = 0.05$, $K = 100$, $q = 0.001$ |
| FEDAVG (McMahan et al., 2017) | $\frac{1}{\eta} = 0.05$, $K = 100$ |
| SCAFFNEW (Mishchenko et al., 2022) | $\frac{1}{\eta} = 0.05$, $p = 0.01$, $q = 0.001$ |
| SABER (Mishchenko et al., 2024) | $\frac{1}{\eta} = 0.05$, $\lambda = 0.01$, $p = 0.01$, $\beta = 0.2$, $q = 0.001$ |

Table J.3: Hyper-parameters of the considered optimizers used in the multi-classification task for the CIFAR10 dataset.

| Optimizers | I-CGM-RG-SAGA | I-CGM-RG-SVRG | SABER-FULL | SCAFFOLD | SCAFFNEW | FEDAVG |
|---|---|---|---|---|---|---|
| **Accuracy** | 76.1 | **77.0** | 74.5 | 72.3 | 74.2 | 74.3 |

Table J.4: Comparisons of validation accuracy for different optimizers used in the multi-classification task for the CIFAR10 dataset within 100 outer iterations.

---

**Algorithm 1** I-CGM-RG-SAGA with $\text{CGM}_{\text{rand}}$

---

1: **Input:** $\mathbf{x}^0 \in \mathbb{R}^d$, $m \in [n]$, $\lambda > 0$, $\beta \in (0,1]$, $p \in (0,1)$, $\eta > 0$, $\mathbf{g}^0 = \nabla f(\mathbf{x}^0)$
2: **for** $t = 0, 1, 2, \ldots$
3: $\quad \hat{K}_t \sim \text{Geom}(p)$
4: $\quad \mathbf{y}_0^t = \mathbf{x}^t$
5: $\quad$ **for** $k = 0, 1, 2, \ldots, \hat{K}_t$
6: $\quad\quad \mathbf{y}_{k+1}^t = \frac{1}{\eta+\lambda}\big(\eta \mathbf{y}_k^t + \lambda \mathbf{x}^t + \nabla f_1(\mathbf{x}^t) - \mathbf{g}^t - \nabla f_1(\mathbf{y}_k^t)\big)$
7: $\quad \mathbf{x}^{t+1} = \mathbf{y}_{\hat{K}_t+1}^t$
8: $\quad$ Sample $S_t \in \binom{[n]}{m}$ uniformly at random without replacement
9: $\quad$ Update $\mathbf{G}^t$ according to (SAGA)
10: $\quad$ Update $\mathbf{g}^{t+1}$ using $\mathbf{G}^t$ according to RG

---

**Limitations and Future Extensions.** 1) In this work, we have assumed that there exists one delegated client that is reliable for communication. If we modify the setting and remove this delegated client, then we can still guarantee similar complexity with minor modifications. Specifically, instead of fixing the index 1 in I-CGM, we can sample $i_t \in [n]$ uniformly at random and define the updates as $\mathbf{x}^{t+1} \approx \arg\min_{\mathbf{x} \in \mathbb{R}^d}\big\{F_t(\mathbf{x}) := f_{i_t}(\mathbf{x}) + h_{i_t}(\mathbf{x}^t) + \langle \mathbf{g}^t - \nabla f_{i_t}(\mathbf{x}^t), \mathbf{x} - \mathbf{x}^t \rangle + \frac{\lambda}{2}\|\mathbf{x} - \mathbf{x}^t\|^2\big\}$. This variant uses R-CSS instead of D-CSS at each iteration. To ensure the convergence rate of $\frac{\lambda F^0}{T}$, we need to choose $\lambda \simeq \Delta_{\max}$ (Jiang et al., 2024a), where $\Delta_{\max} \lesssim \Delta_1$ is defined in D.3. However, suppose there exists more than one delegated client, then it is interesting to check if we can further improve the current complexity. 2) We have shown that the variance of the SAGA estimator is bounded by the function similarity constant $\delta$. An interesting question is whether something similar can be done for another closely related popular gradient estimator, SAG (Schmidt et al., 2017), used in Scaffold (Karimireddy et al., 2020). It turns out that the answer is negative (see Section I). 3) Our analysis focuses on the deterministic first-order oracle $O_{f_i} = O_{FO_i}$. It is interesting to develop efficient algorithms with stochastic, zero-order, or higher-order oracles. 4) The current model does not impose constraints on the size of information that is transmitted between the server and clients. A promising direction is to incorporate communication compression and study how such constraints affect the algorithm design and overall complexity.

