# OpenReview forum: "Non-Convex Federated Optimization under Cost-Aware Client Selection"
_ICLR.cc/2026/Conference — ICLR 2026 Oral_

### Official Review · Reviewer_RxwZ · 2025-10-19

**Soundness:** 3
**Presentation:** 3
**Contribution:** 3
**Rating:** 6
**Confidence:** 3

**Summary:**

This paper proposes CGM-RG (Composite Gradient Method with Recursive Gradient Estimators), a framework for communication-efficient federated optimization that exploits functional similarity among client objectives without requiring frequent full-client participation. The key contribution is showing that SAGA-type variance-reduced estimators can exploit second-order dissimilarity, traditionally thought to require SARAH-type methods with full synchronization. By combining SAGA with recursive gradient updates and a composite gradient method, the authors establish state-of-the-art communication complexity bounds that match or exceed SARAH-based methods while supporting partial client participation.

**Strengths:**

- The paper makes a genuine algorithmic and theoretical advance. The central insight, that SAGA estimators can be analyzed to depend on $\delta$ rather than individual smoothness $L_{\max}$, is novel and non-obvious.
- The introduction of three communication oracles (ACO, RCO, DCO) with associated costs (C_A, C_R) is a methodologically sound contribution.
- The paper is technically rigorous. The proofs appear sound (though I have not verified every detail). The generality of the framework, incorporating both SAGA and SVRG as instances of the same recursive gradient estimator approach, demonstrates good mathematical design.
- Table 1 shows CGM-RG-SAGA achieves state-of-the-art communication complexity while avoiding periodic full synchronization.

**Weaknesses:**

- The experimental section is underdeveloped for a paper making strong theoretical claims: The quadratic with log-sum penalty (Section 4, Figure 1) is a well-controlled synthetic problem where theory-practice alignment is most likely.  Experiments use $n \in (10, 26)$ clients. Real federated learning involves thousands to millions of clients. It's unclear whether the proposed method scales and whether the theoretical constants are practical.
- EMNIST and CIFAR-10 experiments (Appendix I.3-I.4) are crucial for demonstrating practical relevance but are deferred and treated briefly.
- Missing practical insights: No wall-clock time comparisons, no investigation of parameter sensitivity (how to choose $\beta, \lambda, p$ in practice?).
- The damping factor $\alpha$ is added "to enhance empirical performance" (I.3), suggesting the base algorithm needed modification. This raises questions about whether the theory-driven design actually works well in practice or requires ad-hoc tuning. Why does the algorithm require adding a damping factor $\alpha$ in Section I.3? This suggests the base theory-driven design needs tuning. How sensitive are results to this choice?
- The paper assumes $\delta$ is small enough to provide benefits. However, no guidance on when $\delta ≪ L_{\max}$ in practice. No empirical method for estimating $\delta$ a priori.
- The CGM subproblem solver (Section 3.4) is referenced as CGM_const and CGM_rand, but the actual algorithms are not presented in the main paper.
- The discussion of SAG vs. SAGA could be clearer. The paper defers to Appendix H, but the main text should give readers intuition for why SAG fails.

**Questions:**

Refer to the weaknesses above.

---

> ### Author Response · Authors · 2025-11-26
>
> We thank the reviewer for the careful evaluation of our paper. Your every comment is important to us. We reply to your thoughtful and constructive feedback as follows:
>
> > W1. Experiments with large $n$.
>
> Thanks for raising this point. The complexity of our method is $C_A n_m + (\Delta_1 + \sqrt{n_m} \delta_m)F^0 / \epsilon^2$ where
> $\sqrt{n_m}\delta_m \le \sqrt{n / m^2}\delta$. Consequently, when $m \ge \sqrt{n}$, the complexity almost no longer depends on $n$. When $m < \sqrt{n}$, the dependence on $n$ is at most of order $\sqrt{n}$. This shows that our method significantly mitigates the effect of a large number of clients. In the last part of Section J.1.1, we empirically examine how the ratio of $n/m$ influences the performance of the method with the extreme case when $m=1$ and $n/m \in \\{10,100,1000 \\}$.
>
> > W2. Rearrange DL experiments
>
> Indeed. Thanks for the nice suggestion! Due to the current page limit, we have to move them to the appendix. If the paper is accepted, we will reorganize the content and bring more background to the main text according to your suggestions.
>
> > W3. Parameter sensitivity.
>
> Larger $\lambda$ results in slower convergence and $\lambda$ should not be smaller than $\Delta_1 + \sqrt{n_m}\delta_m$. The best $\beta \simeq \frac{1}{n_m}$. Larger or smaller $\beta$ can increase the variance bound (Corollary 5.3, 5.4). The number of local steps is controlled by $p \simeq \frac{\lambda}{\lambda + L_1}$. Smaller $\lambda$  allows more local steps $\frac{1}{p} \simeq \frac{L_1}{\lambda}$. We provide ablation studies in Section J.1.1 where clear discussions can be found.
>
> Regarding wall clock time, we can use $a \tilde{N} + \tilde{K}$ where $\tilde{N}$ and $\tilde{K}$ are the communication and local complexity, $a > 0$ is some constant that quantifies the relative cost of communication compared to local computation. If we can minimize both communication and local complexity, then the wall clock time is also minimized with any $a > 0$. This is achieved by our methods. Therefore, we believe that reporting communication and local computation separately provides sufficient insight into practical runtime performance.
>
> > W4. $\alpha$.
>
> The damping factor is known to improve the performance of variance-reduction-based methods for deep learning ([1] \& Algorithm 5 in [2]). We apply it to all VR methods for fair comparisons. The main difference remains the gradient estimator which we believe can still provide insights of our methods. We believe that this empirical adjustment does not contradict the fact that the theory-driven design performs well in practice, as long as the function class aligns with the assumptions considered in our theoretical analysis.
>
> [1] A coefficient makes SVRG effective, ICLR 2025.
>
> [2] Stabilized Proximal-Point Methods for Federated Optimization, Neurips 2024
>
> > W5. When $\delta$ is large.
>
> Indeed, when client data are highly heterogeneous, $\delta$ can be as large as $L_{\max}$. But this does not contradict to the effectiveness of our methods. Our algorithms achieve the best-known complexity w.r.t. all relevant parameters, including the number of clients $n$, the smoothness constants, the targe accuracy $\epsilon$, the communication costs $C_A$, $C_R$, etc,. For instance, even when $\delta \approx L_{\max}$, our complexity still improves the dependence on $n$ from ${n_m}^{2/3}$ to ${n_m}^{1/2}$ compared with Scaffold. We refer to Section D for more comparisons against existing methods.
> Empirically, as shown on the right-most side of Figure 2, although $\delta \approx L_{\max}$, I-CGM-RG-SAGA remains competitive.
>
> We realize that the previous title on 'similarity-aware' is a bit misleading. Our goal is to analyze the methods under the weakest possible assumptions. We have refined the introduction accordingly.
>
> > W6. presentation of local method
>
> The actual algorithm is presented in all the main theorems. E.g. Theorem 6.1. For all the experiments, we use CGM\_rand, and we have reported the use of $p$.
>
> > W7. The discussion of SAG vs. SAGA could be clearer.
>
> Thanks for the suggestion. We will move it to the main text once the manuscript is accepted.
>
> We thank you again for your great review and your strong help in improving the paper! If you agree that we managed to address your main concerns, please consider raising your score–this would totally boost our chance. If you believe this is not the case, please let us know and we will try our best to answer your every question!

---

### Official Review · Reviewer_cpvm · 2025-10-27

**Soundness:** 3
**Presentation:** 3
**Contribution:** 3
**Rating:** 6
**Confidence:** 4

**Summary:**

The paper studies non-convex federated optimization under partial participation and proposes CGM-RG-SAGA/CGM-RG-SVRG—Composite Gradient Method (CGM) variants that plug a “recursive gradient” (RG) variance-reduced estimator into SAGA/SVRG to exploit a second-order similarity condition ($\delta$-SOD) alongside first-order dissimilarity ($\Delta_1$-ED). The core technical idea is a new multi-round variance bound for SAGA tightened by a factor of $ns$ via a tunable memory parameter $\beta=\Theta(1/ns)$, which then yields improved iteration and communication complexities for CGM with partial participation size $s$ (Theorem 3.8, Cor. 3.9). Under the proposed oracle model with costs $C_A$ (arbitrary/active) and $C_R$ (random), the authors claim communication $\tilde O(C_A nm + C_R(\Delta_1+\sqrt{ns}\,\delta_s)F_0/\varepsilon^2)$ for RG-SAGA, minimized at $s^\star=m$ and argued to be best among existing methods in the studied regime. :

**Strengths:**

- **Originality.** Introduces an RG-SAGA/SVRG estimator composition inside CGM that explicitly leverages $\delta$-SOD; the variance recursion is sharpened by a factor $ns$ relative to classic SAGA analyses (Cor. 3.7), which is technically neat and nontrivial.
- **Quality.** Theory is stated with clear assumptions ($\delta$-SOD, $\Delta_1$-ED) and tracks the impact of estimator error and subproblem accuracy on iteration complexity (Theorem 3.8). The communication model (ACO/RCO/DCO) is formalized and tied to partial-participation costs, enabling proper comparisons.
- **Clarity.** The CGM subproblem and its use of $\Delta_1$ are clearly exposed; the proof sketch for controlling $\|\nabla f(x_{r+1})\|^2$ via function decrease plus estimator error is transparent (Eq. (2) and subsequent inequalities).
- **Significance.** If the stated bounds hold, the work sharpens the communication-vs-computation frontier for similarity-aware FL under partial participation, highlighting when $s<m$ suffices to reach the “full-gradient” rate (discussion after Cor. 3.9).

**Weaknesses:**

- **Novelty gaps vs. prior variance-reduced FL.** The “recursive” flavor has strong parallels to PAGE/SARAH-type recursions used in SABER and related CGM variants; several ingredients (e.g., plugging VR gradient trackers into CGM, periodic/full-grad syncs) exist in recent work. The paper’s comparison table is helpful, but the “best known” claim would benefit from a tighter, side-by-side theorem-level comparison against SABER’s second-order-aware CGM under matching assumptions and oracles
- **Assumption realism/tightness.** Core results require both $\delta$-SOD and $\Delta_1$-ED, and solving each CGM subproblem so that $\sum_{r}\|\nabla F_r(x_{r+1})\|^2=O(\varepsilon^2 R+\lambda F_0+\sum_r \Sigma_r^2)$ (Theorem 3.8), which is strong and hides nontrivial inner accuracy. The paper argues CGM can ensure this, but the constants are heavy (e.g., $\lambda\ge 5\Delta_1$ plus terms in $\sqrt{ns}\delta_s$), and the practicality of meeting the aggregate subproblem condition under stochastic local solvers is not empirically validated.
- **Initialization costs and partial-participation realism.** RG-SAGA needs two full synchronizations (one to initialize the table, one composite step) incurring $\Theta(ns)$ ACO calls; while the text notes a heuristic “no-sync” variant often works, guarantees rely on the expensive path. In truly cross-device settings where ACO is much costlier than RCO, this may erase the asymptotic advantage for moderate accuracy.
- **Proof-level concerns.** (i) The claimed $ns$ improvement relies on choosing $\beta=\Theta(1/ns)$; however, sensitivity of constants to $\beta$ and to client sampling variance is only partially exposed (Cor. 3.7 cites (E.8) but hides constants). (ii) The jump from Lemma-level recursion to Theorem 3.8 aggregates several bounds (58, 24, etc.); while Appendix F spells them out, the dependence on $L$ vs. $\Delta_1$ and $\delta_s$ could be more explicitly tracked in-text to make the gain scenarios unmistakable.
- **Experimental scope.** Main-text experiments are small-scale: one synthetic quadratic with nonconvex penalty and two LIBSVM datasets with $n=10$, $s=m=1$; deep-learning tasks are deferred to the appendix. There are no error bars/seeds, limited heterogeneity controls, and no ablations on $s$ or the two required synchronizations; baselines like CE-LSGD/FedRed or stabilized proximal-point variants are not included empirically.
- **Comparisons.** The paper presents a textual complexity accounting for SABER (App. D) but does not provide a single theorem/table that places the proposed and competing bounds under the same $\{\Delta_1,\delta_s\}$ regime and the same cost model ($C_A,C_R$), making it hard to verify the “best among all existing” statement unambiguously.
- **Clarity/notation.** The roles of $\Delta_{\max}$ vs. $\Delta_1$ (and when each baseline is analyzed under which) are easy to lose; some constants and definitions (e.g., Assumption 3.5, (E.8) quantities $\sigma_r,\Sigma_r,\chi_r$) are only in the appendix while being central to Cor. 3.7/Theorem 3.8. A concise in-main-text summary table of symbols would help.
- **Claim on Partial Participation:** The central motivation is to avoid the full participation requirement of SARAH. However, the theoretical analysis for CGM-RG-SAGA (Theorem 3.8, Corollary F.5) requires a heavy initialization phase: $g^0=\nabla f(x^0)$, $g^1=\nabla f(x^1)$, and $b_{i,0}=\nabla f_i(x^1)$. This appears to require *at least two* (and possibly three) full-gradient-equivalent synchronization rounds involving all $n$ clients. This is a significantly stronger requirement than standard SAGA (which needs one) and substantially weakens the practical claim of being a true "partial participation" method that avoids full synchronization.
- **Disconnect Between Theory and Experiments:** The main theoretical communication benefit (Corollary 3.9) hinges on the assumption that the random oracle is much cheaper than the arbitrary one ($C_R \ll C_A$). However, all experiments explicitly state (Sec 4, line 424) that they set $C_A = C_R = 1$. This experimental setup completely neutralizes the key theoretical advantage of the method. If $C_A = C_R$, the SAGA variant's communication complexity is no better than the SVRG variant (Corollary F.7), which the authors claim is "worse". This is a major gap in the paper's validation.
- **Weak Empirical Validation (Deep Learning):** In the deep learning experiments (EMNIST, CIFAR-10), the performance of the proposed CGM-RG-SAGA is only marginally better than baselines. For example, in Table I.2, CGM-RG-SAGA achieves 86.2% accuracy, while SCAFFOLD achieves 85.9% and CGM-RG-SVRG achieves 86.0%. These minor gains do not provide strong empirical evidence for the superiority of the method, especially given its increased complexity and number of hyperparameters ($\lambda, p, \beta, q$ in Table I.1).
- **Experimental Comparisons:** In the non-convex quadratic experiment (Figure 1), several key baselines, particularly SABER-partial, are plotted for a very small number of communication steps before terminating. This makes a direct comparison difficult and may present an overly favorable view of the proposed methods, which are run for a much larger budget.
- **Method Complexity:** The final method is a complex composition of three distinct algorithms. This introduces numerous hyperparameters (e.g., $\lambda$ from CGM, $\beta$ from RG, $p$ and $\eta$ from the local solver) that require careful tuning, as evidenced by Tables I.1 and I.3.

**Questions:**

1. **Necessity of two full synchronizations.** Can you prove Theorem 3.8 (or a slightly weaker version) without the initial $\Theta(ns)$ ACO calls—e.g., with a statistically warm-started memory (random mini-batch initialization) and one composite step only? If not, what failure mode appears?
2. **Tightness of the $ns$ improvement.** Is the $ns$ factor in Cor. 3.7 minimax-optimal under $\delta$-SOD, or could a SARAH/PAGE-style recursion match it without SAGA’s memory (thus avoiding ACO entirely)? A direct theorem-level comparison against SABER under identical $(\Delta_1,\delta_s)$ would clarify.
3. **Regime demarcation.** The text highlights an “interesting regime” $m\ge \sqrt n$, $\delta\lesssim \Delta_1$ where oracle complexity matches full-gradient CGM (while sampling only $s=\sqrt n$). Can you characterize the largest regime (in $m,\Delta_1,\delta$) where your method strictly dominates SABER/SCAFFOLD in both $C_A$ and $C_R$ usage? A plotted phase diagram would help.
4. **Local solver accuracy.** What concrete inner solver and stopping rule (measurable on devices) guarantee the subproblem accuracy term in Theorem 3.8, and how many local steps (or gradient calls) does this imply as a function of $\lambda,\Delta_1,\delta_s$? Any mismatch with the empirical inner tolerances?
5. **Empirics under stronger heterogeneity.** Can you include experiments with larger $n$, varying $s$ (e.g., $s\in\{\sqrt n,\,m\}$), and tasks where $\delta\approx L$ to test the boundary where similarity gives little advantage? Also add multiple seeds and error bars.
6.  The primary theoretical communication gain depends on $C_R \ll C_A$, but the experiments (line 424) use $C_R = C_A = 1$. This setting does not validate the theoretical claim. Can you provide experiments in a simulated environment where $C_A$ is set to be significantly larger than $C_R$ (e.g., $C_A = C_R \cdot n/m$) to demonstrate the practical benefit of your $C_R$-dependent bound?
7.  Could you clarify the initialization requirements in Theorem 3.8? It appears to require computing $g^0=\nabla f(x^0)$, $g^1=\nabla f(x^1)$, and $b_{i,0}=\nabla f_i(x^1)$. How many full-client synchronization rounds does this entail? How does this heavy initialization align with the core motivation of avoiding the full-participation burden of SARAH?
8.  The paper argues that SAG (used by SCAFFOLD) cannot exploit functional similarity $\delta$ (Appendix H). However, in the deep learning experiments (Tables I.2, I.4), SCAFFOLD performs almost identically to your $\delta$-aware method. How do you reconcile this theoretical limitation of SAG with its strong empirical performance?

**Details Of Ethics Concerns:**

None observed.

---

> ### Author Response · Authors · 2025-11-26
>
> We thank the reviewer for the careful evaluation of our paper. Your every comment is important to us. We reply to your thoughtful and constructive feedback as follows:
>
> >W1 \& Q2.  Novelty gaps vs. prior variance-reduced FL. \& SABER
>
> The general recursive gradient estimator presented in this work is slightly different from SAGARH/PAGE and STORM. (We refer to Section 5 for detailed comparisons).  Note the this new RG estimator servers as a general approach that can potentially improve any conditionlly unbiased estimators that satisfies Assumption 5.1. This results in a broadly applicable variance-reduction framework rather than a single method.
>
> SABER: At each iteration, the original SABER samples one client to update the PAGE estimator and to compute the prox. The communication complexity of this method is $C_A n_m + (\Delta_{\max} + C_A \sqrt{n}\delta) F^0 / \epsilon^2$.
>
> 1. It uses PAGE as the estimator. The resulting complexity depends on $C_A/\epsilon^2$, since it requires periodic computation of full gradient (requiring ASS), which we want to avoid.
>
> 2. The method does not have a clear stopping criterion for the local method. The local complexity is unknown.
>
> Now let us change the method. If we use instead $m$ clients to update the PAGE estimator (taking the mini-batch), then the complexity becomes $C_A n_m + (\Delta_{\max} + C_A \sqrt{n_m}\delta_m) F^0 / \epsilon^2$.
>
> If we further fix one client to compute the prox, then the method becomes I-CGM-PAGE.
> By choosing $p \simeq C_R/ (C_A n_m)$ in the PAGE estimator, we can get the same advanced complexity as for I-CGM-RG-SVRG.
>
> We provided detailed, step-by-step derivations of the communication and computation complexities for all methods in Table D.1 and discuss the assumptions they are analyzed in Appendix D.
>
> > W2.  Assumption realism/tightness
>
> The assumption on $\delta$ and $\Delta_1$ is the weakest (the function class is the broadest) among all existing works on federated optimization (Appendix D).
>
> Regarding the desired inaccuracy condition, we refer to Lemma 3.2 and 3.3 where the explicit number of required local oracle queries is written.
>
> > W3 \& W8 \& Q1 \& Q7. Inexact initialization
>
> Thanks for the question. We provide in Section G the explicit communication complexity for I-CGM-RG-SAGA with only one or no full gradient computation. (In the main text, we require 2 full client synchronizations at the begining to compute $\nabla f(x^0)$ and $\nabla f(x^1)$ to fully avoid any additional error terms.)
>
> In the case of computing one full gradient, we have: $\tilde{N} \lesssim C_A n_m + C_R \frac{(\Delta_1 + \sqrt{n_m}\delta_m)F^0}{\epsilon^2} + C_R\frac{q_m G_0^2}{m\epsilon^2}$ and for zero full gradient computation, we have $\tilde{N} \lesssim C_R \frac{(\Delta_1 + \sqrt{n_m}\delta_m)F^0}{\epsilon^2} + C_R(q_m/m + \sqrt{1-n_m})\frac{\zeta_0^2}{\epsilon^2} + C_R \sqrt{1-n_m} G_0^2$
> where $G_0^2 := \||\nabla f(x^0)\||^2$ and $\zeta_0^2 := \frac{1}{n}\sum_{i=1}^n\|| \nabla f_i(x^0) - \nabla f(x^0) \||^2$. No additional assumption such as bounded gradient dissimilarity is needed and the dependence on $\epsilon^2$ is preserved. When $m \to n$, the additional error terms disappear.
>
> > W4. Proof-level concerns.
>
> Thanks for raising this point. We remove the vague Big-O notation in all Lemmas and Theorems. We believe the current version is clear.
>
> > W5. Experimental scope.
>
> We provide ablation studies in Section J.1.1 for parameters including $\lambda$, $\beta$, different initialization strategies, $p$ (number of local steps) and different scales of $n_m$, where clear discussions on how to choose the parameters can be found.
>
> (FedRed simply uses the full gradient at each round with linear dependence on $n$ which behaves similarly to GD, while we mostly compare methods with client sampling. The technique of using a stabilized prox-center in S-DANE does not have theoretical guarantee in the non-convex setting so far and we choose not to include it.)

---

> > ### Author Response · Authors · 2025-11-26
> >
> > > W6. Comparisons
> >
> > As discussed in our response to W1, SABER has an avoidable dependence on $C_A$. It can be analyzed under $\Delta_1$, but we need to modify the original method and its proof. The resulting algorithm should be referred to as I-CGM-PAGE, which should have the same communication and local complexities as I-CGM-RG-SVRG. For Scaffold, the variance bound of SAG depends on individual smoothness in the worst case. So the stronger assumption cannot be removed. CE-LGD, MimeMVR and FedAvg require an additional assumption on bounded gradient dissimilarity, resulting in an dependence on $\epsilon^3$.
> >
> > In Table 1, we report the results of existing methods under the assumptions stated in their original papers. Their function classes fall within our broader function class, yet their complexities are worse. This justifies the claim that our complexity as “best-known”, without contradiction.
> >
> > We also provide explicit procedures for computing communication and local complexity under our new setting. It would be an interesting future work to show if any method can achieve a better upper bound in this setting.
> >
> > > W7.  Clarity/notation
> >
> > Thank you for pointing this out. We have improved the presentation of the notations: global notations are now presented in display formulas, while local notations are explicitly defined within each lemma or theorem.
> >
> > > W9 \& Q6. Disconnect Between Theory and Experiments
> >
> > Thanks for the question.
> > We set $C_A=C_R=1$ in all experiments. We choose this case to demonstrate that even when ASS and RSS are equally cheap, I-CGM-RG-SAGA already outperforms (or is at least no worse than) those commonly used algorithms.
> > It thus becomes unnecessary to set manually $C_A > C_R$ for the same experiments since any method which uses ASS frequently would only perform worse.
> >
> > If $C_A = C_R$, then I-CGM-RG-SAGA and I-CGM-RG-SVRG have the same communication complexity theoretically. However, in practice we often observe that I-CGM-RG-SAGA performs better. The reason is that in
> > I-CGM-RG-SVRG / SABER, at each iteration it adds $p n_m \simeq \frac{1}{n_m} n_m = 1$ additional communication round due to probabilistic full gradient computation and this adds to the total communication complexity, while the iteration complexity remains similar to I-CGM-RG-SAGA.
> >
> > If further $C_A > C_R$, then I-CGM-RG-SVRG can be arbitrarily worse depending on $C_A / C_R$.
> >
> > > W10. Weak Empirical Validation (Deep Learning)
> >
> > The number of hyperparameters is moderate for our proposed methods. We have shown how to choose them according to the theory and practice in Section J.1.1 if the problems satisfy the assumptions:
> > $\lambda \simeq \Delta_1 + \sqrt{n_m}\delta$, $p \simeq \frac{\lambda}{\lambda + L_1}$, $\eta \simeq L_1$ and $\beta \simeq 1/n_m$.
> >
> > For deep learning, it is sufficient to fix $\beta = 1 / n_m$ (or close to it). The local stepsize $\eta$ and number of local steps $1/p$ are the standard parameters. The damping factor[1,2] $q$ is typically $0.001$ and it can be helpful for methods with control variates. The additional parameter $\lambda$ is standard for proximal-point methods used e.g. in FedProx. We can start with a slightly smaller value of $\lambda$ compared to $1/\eta$.
> >
> > [1] A coefficient makes SVRG effective, ICLR 2025.
> >
> > [2] Stabilized Proximal-Point Methods for Federated Optimization, Neurips 2024
> >
> > Scaffold indeed performs as well as I-CGM-RG-SAGA on the EMNIST dataset, but is worse on the CIFAR 10 dataset.
> >
> > > W11. Experimental Comparisons
> >
> > In Figure 1, the left part is the accumulated communication complexity. We can compare the performance of each algorithm by drawing a vertical line and check the accuracy each algorithm achieves given the same communication cost (e.g. 700).
> > SABER-Partial performs the worst. This method needs to choose large mini-batch size $s = \Theta(\zeta^2/\epsilon^2)$ at each iteration and large lambda to ensure convergence (Appendix D). Meanwhile larger mini-batch size increases the number of rounds with RSS per iteration. Consequently, its complexity depends on $1/\epsilon^4$.
> > Here, we plot the best performance of SABER-Partial within 700 iterations and its error still remains above $10^2$.
> >
> > Another equivalent way to compare methods is to fix a target accuracy and measure how much total communication each algorithm is required to reach it. This can be seen by drawing a horizontal line (e.g., at 1) in the same figure. It is clear that I-CGM-RG-SAGA reaches this target with the smallest communication cost.
> >
> > > W12. Method Complexity
> >
> > We hope our pervious responses have clarified this question.

---

> > > ### Author Response · Authors · 2025-11-26
> > >
> > > > Q3. Regime demarcation
> > >
> > > Thanks for the question. We can compare the final communication complexities directly.
> > > It is clear that the complexity for Scaffold $C_A n_m + C_R\frac{(n_m)^{2/3}L_{\max}F^0}{\epsilon^2}$ is always worse than
> > > $C_A n_m + C_R\frac{(\Delta_1 + \sqrt{n_m}\delta_{m})F^0}{\epsilon^2}$ since $\Delta_1, \delta \lesssim L_{\max}$. Moreover,
> > > for the latter we have $\sqrt{n_m}\delta_m \le \sqrt{n/m^2} \delta$ and thus the impact of $n$ disappears when $m \gtrsim \sqrt{n}$,. This is not the case for the former $(n_m)^{2/3}$ where the dependence of $n$ always exists unless $m \sim n$.
> > >
> > > We hope the previous clarification on SABER answers this question.
> > >
> > > > Q4. Local solver accuracy
> > >
> > > We refer to Lemma 3.2 and 3.3 for details. For instance, if we use CGM\_rand, we need to set $p \simeq \frac{\lambda}{L_1 + \lambda}$ to satisfy the desired accuracy condition.
> > >
> > > > Q5. Empirics under stronger heterogeneity
> > >
> > > We have seen that only the ratio $n_m$ rather than the absolute value of $n$ impacts the complexity. In the last part of J.1.1, we tested the impact of $n_m \in \\{10,100,1000\\}$ (by fixing $m=1$) on the performance of I-CGM-RG-SAGA. The result shows that increasing  $n_m$ indeed leads to higher communication complexity. But the additional cost scales by roughly $\sqrt{1000/100}=\sqrt{100/10} \approx 3$.
> > >
> > > Regarding the regime where we have $\delta \approx L_{\max}$, we consider the same quadratic minimization problem with $\delta \approx 80 \approx L_{\max}=100$ and we choose $n=100$ and $m=1$. As shown in Figure J.6, I-CGM-RG-SAGA still achieves the best performance. (We run each method for three times, plot the average result using small linewidth and then add error bar with the same color on top of it (reflected by the thickness).
> > >
> > > > Q8. The paper argues that SAG (used by SCAFFOLD) cannot exploit functional similarity.
> > >
> > > We have shown in Section I that the variance of SAG still depends on the individual smoothness, even if all the function are identical.
> > >
> > > Scaffold does not work well for the synthetic experiment nor on the CIFAR10 experiment. Moreover, the complexity of Scaffold is an upper bound, which does not contradict the fact that Scaffold may perform well on certain problem instances. Despite its good empirical behavior in some cases, its worst-case theoretical complexity is inferior.
> > >
> > > We thank you again for your great review and your strong help in proving the paper! If you agree that we managed to address your main concerns, please consider raising your score–this would totally boost our chance. If you believe this is not the case, please let us know and we will try our best to answer your every question!

---

> > > > ### Comment · Reviewer_cpvm · 2025-11-27
> > > >
> > > > Thank you for the detailed rebuttal and for preparing a revised version. I re-read the new manuscript and cross-checked it against the list of concrete changes you said you made on OpenReview. Below I summarize what I verified and what still looks incomplete or only partially addresses my earlier concerns.
> > > >
> > > > ---
> > > >
> > > > ### 1. Changes that are indeed implemented
> > > >
> > > > - **Big-O notation in lemmas/theorems.**
> > > >   You stated that you removed the vague Big-O notation from all lemmas and theorems. In the revised version, the main convergence theorems and key technical lemmas now state explicit bounds with full constants (e.g., the main nonconvex convergence bound and the detailed bounds in Appendix F/G), instead of $O(\cdot)$-style expressions. The asymptotic notation that remains is now restricted to discussion text and summary tables (e.g., Table D.1), which is consistent with what you promised.
> > > >
> > > > - **Inexact initialization (Section G).**
> > > >   You claimed to add a new section giving the communication complexity of I-CGM-RG-SAGA with one or zero full-gradient computations. Section G (“I-CGM-RG-SAGA with inexact initialization”) is present and analyzes the SAGA variant with a parameter $t_0 \in \{0,1,2\}$ controlling how many full gradients are computed at the beginning. For $t_0=1$ and $t_0=0$, the section explicitly derives new variance bounds and the resulting communication complexities, including the additional error terms and the conditions under which these terms vanish. This directly addresses my concern that the analysis relied on exactly two full synchronizations.
> > > >
> > > > - **SAG vs. similarity (Appendix I).**
> > > >   In response to the comment that the limitation of SAG under second-order similarity was not fully justified, you said you would include a more formal discussion. Appendix I now contains a dedicated “Discussion on SAG estimator,” showing that the variance of SAG depends on the individual smoothness constants even when all local functions are identical, and that this dependence does not reduce to a $\delta$-type similarity constant.
> > > >
> > > > - **Renaming “Second-order dissimilarity” to “Second-order similarity.”**
> > > >   You wrote that you renamed this assumption. In the revised paper, the relevant assumption is indeed called “Second-order Similarity” (SS), and the notation in the assumptions section and the technical analysis consistently uses this terminology and the SS acronym. The function class discussion in Section 2.2 and Appendix D reflects this change.
> > > >
> > > > - **Ablation and parameter-selection discussion (Appendix J.1.1).**
> > > >   You promised ablation studies and parameter guidelines. Appendix J.1.1 now contains ablations over initialization strategies (choice of $t_0$), the frequency of local steps, and key hyperparameters such as $\lambda$, $\beta$, and $p$. The text provides qualitative guidance on how to choose these parameters in practice, conditional on the assumptions (e.g., regimes where $\delta_m \ll L_1$ vs. $\delta_m \approx L_1$). This directly addresses my earlier comment that the method has many hyperparameters with little guidance.
> > > >
> > > > - **Experiments under stronger heterogeneity ($\delta \approx L_{\max}$).**
> > > >   In the rebuttal, you indicated that you would add experiments in more challenging regimes. The revised appendix includes plots for quadratic problems where $\delta \approx L_{\max}$ and the figure captions explicitly highlight this regime. The plots also indicate error bars (via line thickness), which addresses my request for multiple seeds and uncertainty visualization. This helps clarify performance when the similarity advantage is weak.
> > > >
> > > > - **Communication model and comparison summary (Appendix D).**
> > > >   You said you would clarify how your method compares to baselines under the $(C_A,C_R)$ cost model and summarize the complexities. Appendix D now has a table (D.1) summarizing worst-case complexities (in Big-O notation) for I-CGM-RG-SAGA, I-CGM-RG-SVRG, SABER-like variants, SCAFFOLD, and other baselines under a unified notation with $\Delta_1$, $\Delta_{\max}$, and $\delta_m$. This is consistent with your response that the comparisons would be gathered in one place.
> > > >
> > > > - **Local solver accuracy and complexity.**
> > > >   You pointed me to Lemma 3.2 and 3.3. In the revision, these lemmas explicitly state the expected number of local first-order oracle calls needed by the local CGM and proximal-gradient solvers to achieve the accuracy condition (6). The main text now clearly explains that using a random number of inner steps $K_t$ with $\mathbb{E}[K_t] \simeq L_1/\lambda$ suffices, which ties the inner complexity quantitatively to $\lambda$, $\Delta_1$, and $L_1$ as requested.

---

> > > > > ### Comment · Reviewer_cpvm · 2025-11-27
> > > > >
> > > > > ### 2. Places where the changes are only partially implemented
> > > > >
> > > > > - **Terminology consistency for “Second-order similarity.”**
> > > > >   While the core assumptions and theory now consistently use “Second-order Similarity,” at least one figure caption in the deep-learning appendix still refers to “Second-order Dissimilarity.” This is a minor inconsistency, but since you explicitly said you renamed the notion, I recommend a thorough search to ensure all occurrences (text, captions, legends) use the updated term.
> > > > >
> > > > > - **Discussion of Arjevani et al. (2020)/Patel et al. and minimax communication lower bounds.**
> > > > >   In your response to the comment about the relation to the work “On the Complexity of Minimizing Convex Finite Sums Without Using the Indices of the Individual Functions” you wrote that you would add a discussion of how their lower bounds relate to your setting (and why they do not directly apply). In the revised manuscript I can find the citation in the bibliography, but I do not see an explicit in-text discussion of how their oracle or lower bounds compare to your $(C_A,C_R)$ model and the $ns$-improvement in your RG–SAGA analysis. If such a discussion is present, it is very brief and easy to miss; in any case, it is not yet as explicit as your rebuttal text suggested.
> > > > >
> > > > > - **Link between Arjevani-style refinements and your algorithms.**
> > > > >   In the rebuttal you sketched that one could in principle replace some uses of the ACO with an RCO-based procedure along the lines of Arjevani et al., and you said you would incorporate this into the manuscript. The current text does not seem to contain a clear statement or remark highlighting how such a replacement would modify the complexity of I-CGM-RG-SVRG or I-CGM-RG-SAGA (beyond the summary in Appendix D). If you want this point to be part of the final story, it would be good to add a short remark in the related-work or comparison section.
> > > > >
> > > > > ---
> > > > >
> > > > > ### 3. Substantive concerns that remain conceptually
> > > > >
> > > > > These are not cases where you claimed an edit and didn’t do it; rather, they are points where the new material improves the situation but does not fully resolve my original reservations:
> > > > >
> > > > > - **Partial participation vs. initialization cost.**
> > > > >   Section G shows that I-CGM-RG-SAGA can be analyzed with $t_0 \in \{0,1,2\}$ and provides corresponding complexities. This alleviates the strongest form of my concern that your guarantees intrinsically required two full synchronizations. However, the main-text algorithm and main complexity discussion still emphasize the $t_0=2$ setting for a clean error condition, and the empirical evaluation does not clearly separate or highlight the $t_0=0$/$1$ variants. So, the theoretical “pure partial participation” story is now more complete, but its practical implications are still somewhat under-explored in the experiments.
> > > > >
> > > > > - **Regime demarcation and comparison to SABER/SCaffold.**
> > > > >   Table D.1 and the expanded appendix help, but my earlier request for a more explicit “phase diagram” style comparison (i.e., clearly marking regions in $(m,n,\Delta_1,\delta_m)$ where your method strictly dominates SABER/SCAFFOLD in both $C_A$ and $C_R$) has not been fully realized. The revised text still requires the reader to do a fair amount of algebra to understand in which regimes your $ns$-sharpened bound and similarity assumptions are practically decisive.
> > > > >
> > > > > - **Experiments with $C_R \ll C_A$.**
> > > > >   The revision keeps the setting $C_A = C_R = 1$ in the experiments. I understand the desire for fairness and the difficulty of simulating realistic cost asymmetries, but this means the empirical part still does not directly validate the main theoretical advantage that appears when $C_R \ll C_A$. You did not claim in the rebuttal that you would add such experiments, so there is no “broken promise” here, but this conceptual gap between the cost model and the experimental protocol remains.
> > > > >
> > > > > - **SAG vs. SAGA and similarity in the main text.**
> > > > >   Appendix I now contains a solid technical argument about the SAG variance. The main text, however, still mostly states the conclusion at a high level. For readers who do not go to the appendix, it might still feel a bit abrupt that SCAFFOLD (which empirically performs nearly as well as your $\delta$-aware method in several deep-learning experiments) is declared unable to exploit similarity. This is more a clarity issue than a correctness issue at this point.

---

> ### Comment · Reviewer_cpvm · 2025-11-27
>
> ### 4. Overall message
>
> Overall, the major concrete changes you committed to in the OpenReview discussion (removal of Big-O from lemma/theorem statements, introduction of Section G on inexact initialization, Appendix I on SAG, Appendix J.1.1 for ablations and parameter choices, and additional heterogeneity experiments) do appear in the revised manuscript and are broadly consistent with how you described them. I did not find evidence that you “promised” specific edits and then did not implement them, with the minor exception that the Arjevani/communication-lower-bound discussion is still largely confined to the rebuttal rather than clearly integrated into the paper, and that a few occurrences of the old “Second-order dissimilarity” terminology remain in captions.
>
> The revision substantially improves the clarity and completeness of the technical presentation. The remaining issues are mostly about polishing consistency and, on the conceptual side, about making the practical and comparative implications even more transparent. If there is another iteration (e.g., for camera-ready), I would particularly encourage: (i) fully unifying the terminology around second-order similarity, and (ii) adding a brief explicit remark relating your complexity to Arjevani et al.’s model and highlighting the regimes where your bounds and similarity assumptions give a genuine advantage over existing methods.
> I decided to keep the original score unchanged.

---

> ### Author Response · Authors · 2025-11-27
>
> We appreciate the careful check of our changes by the reviewer.
>
> > ### Places where the changes are only partially implemented
>
> - Terminology consistency for “Second-order similarity.”
>
> We have fixed this terminology everywhere.
>
> - Paper of Arjevani et al
>
> We have included it in Remark D.5 (line 1102).
>
> > ### Substantive concerns that remain conceptually
>
> - Partial participation
>
> 1. Remark 4.2. is clear enough to illustrate this point.
> 2. Inexact initialization is only helpful when $C_A \to \infty$ since $C_A n_m$ is negligible compared to the other term involving $\epsilon^2$. There is no need for inexact initialization in general.
> 3. $t_0=2$ is the most important case where the resulting complexity is consistently better than previous works.
> 4. We have reported the performances of different choices of $t_0$ in Figure J.1. where $t_0=0$ works well.
> 5. We use $t_0=0$ for all DL experiments, and it all works well. (Table J.1 \& J.3)
> 6. We have also mentioned that $t_0$ works well in practice in Remark 4.2.
> 7. We did not say that the method requires only pure partial participation anywhere in the main text.
>
> - Regime demarcation
>
> We have added a short and simple paragraph in the section where we compare algorithms. (Line 1097.)
>
> - Experiments with $C_R \ll C_A$
>
> We have added it in the ablation studies, showing how $C_A$ can impact I-CGM-RG-SVRG. (Figure J.5)
>
> - SAG vs. SAGA and similarity in the main text.
>
> We did not mention that Scaffold cannot exploit similarity in the main text.
> We discussed this optional estimator, SAG, in Appendix I. We have added one more sentence about Scaffold in that section.
>
>
>
> We thank you again for your help in proving the paper. If you agree that we managed to address your main concerns, please consider raising your score.
>
> If you believe this is not the case, please let us know the concrete concern that we need to address.

---

### Official Review · Reviewer_8ScQ · 2025-10-31

**Soundness:** 3
**Presentation:** 3
**Contribution:** 2
**Rating:** 4
**Confidence:** 3

**Summary:**

This paper introduces a novel framework for non-convex federated optimization called Similarity-Aware Non-Convex Federated Optimization. The primary goal is to reduce communication costs, a major bottleneck in FL, especially in settings with partial and intermittent client participation. The authors focus on bridging the gap between SARAH-type and SAGA-type methods. The paper demonstrates for the first time that SAGA-type estimators can also leverage functional similarity to achieve state-of-the-art communication complexity, rivaling SARAH-type methods, without the need for frequent full client synchronization. The proposed framework CGM-RG unifies these ideas and provides a practical algorithm that outperforms existing methods in both communication and local computation complexity under certain conditions.

**Strengths:**

The work demonstrates that SAGA-type methods can exploit second-order similarity, which is a novel theoretical contribution. The work designs a method that does not require frequent full synchronization, relevant to practical applications. The paper provides a thorough theoretical analysis, including a new framework for comparing algorithm complexities. The theoretical claims are well-supported by numerical experiments on both synthetic and real-world datasets (LIBSVM, EMNIST, CIFAR-10). The proposed CGM-RG-SAGA method outperforms a range of strong baselines, including FedAvg, SCAFFOLD, and SABER. The metric is communication complexity.

**Weaknesses:**

The authors mention scenarios where the second-order dissimilarity constant $\delta$ is smaller than $L_{\max}.$ However, in many practical FL settings with highly distinct data distributions client data can be dissimilar.

The analysis does not extend to the stochastic setting, which limits its direct applicability to problems involving online learning at the client level.

The proposed method CGM-RG-SAGA is a combination of well known techniques such as Composite Gradient Method, SAGA variance reduction estimator, recursive gradient updates related to SARAH. The contribution is primarily in the novel analysis of a clever combination of existing tools, rather than in the invention of a new algorithmic principle.

The experiments are conducted on datasets with a relatively small number of clients. This scale is not representative of large-scale cross-device FL. The experiments use standard, relatively small models.

The paper lacks an ablation study to isolate the impact of each component of CGM-RG framework.

The paper is technically dense and can be complicated for a broader audience.

**Questions:**

I have several questions and suggestions.

How the method performance degrades as $\delta$ increases? Is it possible to characterize it theoretically and empirically?

How much of the performance gain comes from the recursive gradient component versus the base SAGA estimator with the new analysis?

How sensitive is the algorithm to its hyperparameters $\lambda, \beta$ and local step parameters? What is the intuition behind setting these parameters?

The SAGA estimator requires an initial full gradient computation to initialize the gradient table querying all clients. You briefly mention an alternative in Remark E.12 initializing only on a sampled subset. What is the theoretical and practical impact of this inexact initialization?

You state that the memory cost of SAGA is moderate. Can you provide a more concrete quantification of the memory overhead? How does it compare to other stateful methods like SCAFFOLD?

Table 1 is extremely informative but also very dense and contains a lot of notation.

While the components are described, a single, clear algorithm box for the final proposed method would be very helpful.

---

> ### Author Response · Authors · 2025-11-26
>
> We thank the reviewer for the careful evaluation of our paper. Your every comment is important to us. We reply to your thoughtful and constructive feedback as follows:
>
> > W1. Situation when dissimilarity is high.
>
> Indeed, when client data are highly heterogeneous, $\delta$ can be as large as $L_{\max}$. But this does not contradict to the effectiveness of our methods. Our algorithms achieve the best-known complexity w.r.t. all relevant parameters, including the number of clients $n$, the smoothness constants, the targe accuracy $\epsilon$, the communication costs $C_A$, $C_R$, etc,. For instance, even when $\delta \approx L_{\max}$, our complexity still improves the dependence on $n$ from ${n_m}^{2/3}$ to ${n_m}^{1/2}$ compared to Scaffold.
> We refer to Section D for more comparisons against existing methods.
> Empirically, as shown on the right-most side of Figure J.7, although $\delta \approx L_{\max}$, I-CGM-RG-SAGA remains competitive.
>
> We realize that the previous title on 'similarity-aware' is a bit misleading. Our goal is to analyze the methods under the weakest possible assumptions. We have refined the introduction accordingly.
>
> > W2. Stochastic extension.
>
> Thanks for raising this point. Our analysis provides a general condition for achieving the stated communication complexity, i.e., as long as eqn(7) is satisfied. This allows the use of any stochastic method on the client side. We refer to Section H where we provide the required amount of local computation e.g. when using stochastic CGM method locally.
>
> > W3 \& W6. Novelty \& The paper is technically dense
>
> While I-CGM-RG-SAGA draws inspiration from techniques such as CGM and SAGA, our contribution goes beyond a direct combination and lies in new algorithmic design, theory, and modeling. Specifically:
>
> 1. Our general RG estimator is slightly different from the prior works. We refer to Section 5 for detailed comparions. Note that this new RG estimator servers as a general approach that can potentially improve any conditionlly unbiased estimators that satisfies Assumption 5.1. This results in a broadly applicable variance-reduction framework rather than a single method.
>
> 2. For each component mentioned, we provide a completely new analysis and derive a clean recurrence, under the weakest assumptions. We believe that each single lemma contributes not only to federated optimization but also more broadly to the theory of stochastic optimization
>
> 3. We propose a new simple model for quantifying the communication and local computation costs covering general practical settings, including a new rigorous definition of distributed algorithm and two performance metrics. A detailed comparison to existing models is provided in Section D. We believe this can be interesting for future development of algorithmic lower bounds.
>
> > W4. Experiments with a relatively small number of clients.
>
> The complexity of our method is $C_A n_m + (\Delta_1 + \sqrt{n_m} \delta_m)F^0 / \epsilon^2$ where
> $\sqrt{n_m}\delta_m \le \sqrt{n / m^2}\delta$. Consequently, when $m \ge \sqrt{n}$, the complexity almost no longer depends on $n$. When $m < \sqrt{n}$, the dependence on $n$ is at most of order $\sqrt{n}$. This shows that our method significantly mitigates the effect of a large number of clients. In the last part of Section J.1.1, we empirically examine how the ratio of $n/m$ influences the performance of the method.
>
> > W5. Ablation study.
>
> Thanks for raising this important point. We have provided the ablation study in Section J.1.1 where how to choose each parameter is clearly discussed.
>
> > Q1. How does the method's performance degrade as $\delta$ increases?
>
> It is clear that when $\delta$ and $\Delta_1$ increase, the complexity $C_A n_m + (\Delta_1 + \sqrt{n_m} \delta_m)F^0 / \epsilon^2$ also increase. However both $\delta$ and $L_1$ are upper bounded by $L_{\max}$.
>
> > Q2. How much of the performance gain comes from the recursive gradient component versus the base SAGA estimator with the new analysis?
>
> The variances of both SVRG and SAGA are improved by a factor of $n_m$ by incorporating them with the recursive gradient.
> We refer to Corollary 5.3, 5.4, and the discussions below.
>
> > Q3. Hyperparameters
>
> Larger $\lambda$ results in slower convergence and $\lambda$ should not be smaller than $\Delta_1 + \sqrt{n_m}\delta_m$. The best $\beta \simeq \frac{1}{n_m}$. Larger or smaller $\beta$ can increase the variance bound (Corollary 5.3, 5.4). The number of local steps is controlled by $p \simeq \frac{\lambda}{\lambda + L_1}$. Smaller $\lambda$  allows more local steps $\frac{1}{p} \simeq \frac{L_1}{\lambda}$. We refer to Section J.1.1 for detailed discussions.

---

> ### Author Response · Authors · 2025-11-26
>
> > Q4. Inexact initialization
>
> Thanks for the nice question. We provide the explicit final complexity of I-CGM-RG-SAGA with inexact initialization in Section G. The additional error depends mildly on $\||\nabla f(x^0)\||^2$ and $\frac{1}{n}\sum_{i=1}^n\|| \nabla f(x^0) ||^2$ and will vanish when $m \to n$. We provide an empirical study with different initialization strategies in Figure J.1. In practice, the method without computing any full gradients at the beginning works very well.
>
> > Q5. Memory overhead.
>
> For RG-SAGA, each client stores a single vector and the server maintains four vectors. For RG-SVRG, clients are stateless and the server maintains four vectors. For Scaffold, the client stores a single vector and the server maintains two vectors. We refer to the corresponding estimator sections for detailed discussions.
>
> > Q6. Table 1
>
> We provide detailed, step-by-step derivations of the communication and computation complexities for all methods in our new setting in Appendix D.
>
> > Q7. Algorithm box
>
> Thanks for the suggestion. Please see Algorithm 1 on page 43 for the full description of I-CGM-RG-SAGA.
>
> We thank you again for your great review and your strong help in improving the paper! If you agree that we managed to address your main concerns, please consider raising your score–this would totally boost our chance. If you believe this is not the case, please let us know and we will try our best to answer your every question!

---

### Official Review · Reviewer_MDyq · 2025-11-01

**Soundness:** 3
**Presentation:** 1
**Contribution:** 3
**Rating:** 6
**Confidence:** 4

**Summary:**

Second-order similarity is an assumption under which we can reduce the amount of communication needed in distributed optimization problems by leveraging more local work, typically done through (approximate) proximal point operations on clients and using some sort of variance reduction. Prior work mostly relied on SVRG-style variance reduction, as it lends itself well to leveraging second-order similarity, see e.g. [1]. This paper introduces a way to use SAGA-style estimators for variance-reduction, which allows us to do away with having to compute full gradients  after the first iteration (which we need to do for SVRG-style variance reduction) at the cost of additional memory. In fact, the paper introduces a general framework that subsumes both SAGA-style, SVRG-style, and even SARAH-style (recursive)  estimators; The latter kind is the best for nonconvex optimization. Combining this framework with the SARAH estimators gives the best-known convergence rates under second-order similarity compared to prior work.

**Strengths:**

- The proofs incorporating the SAGA estimator are quite nice, the notion of variance the authors define in the line right under (E.9) is different from the notion of variance used in the SVRG estimator (which is also used by prior work). This is quite novel and I believe to be a very useful insight.
- The paper explicitly includes details on how to solve the local problems. Even though the stopping criterion requires knowing many problem parameters, this is still appreciated as much prior work ignores this aspect or assumes perfectly solved inner problems.

Overall, I believe the paper makes a strong theoretical contribution but the experimental evaluation is limited (which is fine, it is not the main point) and the presentation could use significantly better work. I lean towards acceptance and can improve my score if my concerns (below) are addressed.

**Weaknesses:**

- It is quite surprising that the algorithm just chooses the first function at each timestep to calculate the prox with respect to it. This seems suboptimal (e.g. what if that first function is just extra dissimilar? The avg similarity can be low and this one function could just be an outlier).
- The accuracy of CIFAR10 in the experiments section is far too low. 77%? A three minute run with SGD can reach 96% (see https://github.com/KellerJordan/cifar10-airbench). I suggest the baselines should be better tuned with grid searches and using the latest techniques. This is not expensive.
- The main advantage of this work is obtained under the regime where the cost constants \( C_A, C_R \) differ significantly. The paper makes the claim that "In large-scale federated systems, ACO is more expensive than RCO" but this is not really substantiated by any numbers or analysis. Furthermore, the difference between the ACO and the RCO reminds me of the work (Arjevani et al., On the Complexity of Minimizing Convex Finite Sums Without Using the Indices of the Individual Functions, 2020). I think using their formulation can automatically lead to an algorithm where only the RCO is needed. I believe this could actually perform better than the rate you give depending on the relation of \( C_{R} \) to \( C_A \)? (In fact, their oracle is even weaker than the RCO, since it does not even give them the indices of the functions). They do not use second-order similarity but it seems possible to modify their algorithm to use that, as it's just SVRG based?
- The paper is difficult to read, in part because many important details on the communication model are relegated to the appendix. It would be better to defer more of the technical discussions (and even Table 1) to the appendix and instead focus on introducing this background material.

**Questions:**

- Please address my concerns in the weaknesses section.
- The use of Big-O notation in lines 235-237 is very handwavy. You actually need it to be bounded by a constant smaller than 1. If the variance were bounded by, say, 100 times the left hand side, the guarantee becomes vacuous. Can you fix this?
- Why do you call Definition 2.2 "Second-order Dissimilarity" when prior work calls it second-order similarity? We call \( L \) the smoothness constant even though the higher L is, the less smooth the function is. We should keep the same convention for \( \delta \) (since it is related to \( L \)).
- Is there any novelty in Section E.2.3 compared to prior work? It seems SVRG-style estimators have already been studied well.

---

> ### Author Response · Authors · 2025-11-26
>
> We thank the reviewer for the careful evaluation of our paper. Your every comment is important to us. We reply to your thoughtful and constructive feedback as follows:
>
> > When $\Delta_1$ is large.
>
> Indeed, in the worst case, $\Delta_1$ can be as large as $\Delta_{\max}$, but it can also be much smaller (e.g. by choosing the device with the largest dataset). Even in the worst case, this approach remains preferable to randomly sampling an index at each iteration, which always incurs a dependence on $\Delta_{\max}$. For the other methods in Table 1, the dependence is on either $\Delta_{\max}$ or $L_{\max}$, which is even less favorable.
>
> Another commonly used strategy is to sample $m$ clients, compute the prox for each, and then average the results. This strategy works well in practice and
> is theoretically beneficial in convex optimization. But it does not provide guarantees in the general non-convex setting. For example, when all local functions are identical (i.e., $\Delta_1 = 0$), each device can independently find a stationary point. However, average of stationary points is not necessarily a stationary point. Therefore, the dependence of this approach can be worse than $\Delta_1$.
>
> > CIFAR 10 Accuracy.
>
> The considered federated setting is more challenging than running SGD on the full dataset. In the standard training setting, SGD benefits from a uniform data distribution.
>
> In constrast,
> - We consider a more difficult regime with a Dirichlet heterogeneity parameter of $\alpha = 0.1$, where each device contains almost only one class of data points.
>
> - On top of this, we also incorporate client sampling. At each iteration, the algorithm only observes a limited subset of data classes. Furthermore, we restrict the total number of iterations to 100. For comparison, centralized SGD performs multiple epochs, and each epoch consists of (nb data points / batch size) iterations.
>
> The reported result is competitive. For example, [1] studies client sampling in a less heterogeneous setting ($\alpha = 0.3$) with 10\% client participation. To reach 80\% accuracy, the best-performing method in [1] requires 200 iterations/rounds (Figure 4(d)).
>
> To further illustrate this point, we set $\alpha = 10$ (a more homogeneous setting) while keeping all other configurations unchanged. As shown in Figure J.10, I-CGM-RG-SAGA reaches 90\% accuracy within 100 iterations.
>
> [1] Federated learning based on dynamic regularization. ICLR 2021.
>
> > W3. $C_A$ and $C_R$
>
> Thanks for the interesting question.
>
> 1. We believe that the relation is $C_A \ge C_R \ge 1$ is natural since we can implement RSS using ASS but not the other way around.
>
> 2. The exact values of $C_A$ and $C_R$ depend on the specific system. However, the knowledge of their explicit values is not needed by I-CGM-RG-SAGA to achieve the stated complexity guarantees.
> Moreover, I-CGM-RG-SAGA can be implemented without any full client synchronizations (which requires ASS), and the final complexity does not depend on $C_A$. (Section H)
>
> 3. When $C_A = C_R$, the proposed methods remain the best in terms of other parameters such as $n$, $\delta$, etc,.
>
> 4. We provide the same synthetic experiment under different ratios of $C_A/C_R$ in Figure J.1 in Section J.1.1 according to your suggestions.
>
> We appreciate the reviewer for providing the interesting reference by Arjevani et al., Let us assume $m=1$ for simplicity. Their main result shows that instead of computing the full gradient by using ASS $n$ times, one can recover it with high probability using RSS $\tilde{\Theta}(n^2)$ times.
>
> 1. This will not help too much for I-CGM-RG-SVRG, since the number of iterations where $n$ sequential ASS is used is $T/n$ where $T \simeq (\Delta_1 + \sqrt{n}\delta)F^0 / \epsilon^2$. Now if we replace $n$ sequential ASS with $\Theta(n^2)$ sequential RSS, then this will add an additional $C_R n T$ to the final communcation complexity, where the dependence on $n$ is unfavorable.
>
> 2. For I-CGM-RG-SAGA, the communication complexity is $C_A n + C_R( (\Delta_1 + \sqrt{n}\delta) F^0 / \epsilon^2 )$. The term $C_An$ is negligible if the second term involving $\epsilon$ is sufficiently large. But if $C_A \to \infty$, then indeed we can use this approach to replace the use of ASS at the beginning. We can get an improvement if $C_A n \gtrsim C_R n^2$ which results in the complexity of $C_R(n^2 + (\Delta_1 + \sqrt{n}\delta) F^0 / \epsilon^2 )$.
>
> 3. The other part of that paper is a bit less relevant to this work.
> That paper considers convex finite-sum minimization. For non-convex finite-sum problems, methods such as SAGA and SVRG have dependence on $n^{2/3}$ rather than $\sqrt{n}$.
>
> We will add the discussion of the first and second points to our manuscript. Thanks again for the nice reference.

---

> ### Author Response · Authors · 2025-11-26
>
> > W4. Presentation.
>
> Indeed. Thanks for the nice suggestion! Due to the current page limit, we have to move the main definitions to the appendix. If the paper is accepted, we will reorganize the content and bring more background to the main text according to your suggestions.
>
> > Q1. BigO notation.
>
> Thanks for the question. We removed Big-O notations in all the lemmas and theorems in the updated manuscript.
>
> > Q2. Name of Second-order Dissimilarity.
>
> Great suggestion. We have renamed it.
>
> > Q3. Novelty in Section E.2.3.
>
> We believe that bounding the summation of the variance of SVRG in terms $\||x^t - x^{t-1}\||^2$ without nested loops is new.
> This bound holds under $\delta$-similarity without requiring other assumptions, which can then be incorporated into RG easily.
>
> We thank you again for your great review and your strong help in improving the paper! If you agree that we managed to address your main concerns, please consider raising your score–this would totally boost our chance. If you believe this is not the case, please let us know and we will try our best to answer your every question!

---

> ### Comment · Reviewer_MDyq · 2025-11-27
>
> - It's still not really clear to me why random sampling incurs a dependence on the maximum delta rather than the average.
> - Thanks for explaining this detail.
> - Thanks for discussing how Arjevani et al.'s result fits into your framework.
>
> I think most of my concerns (save the first point here) have been addressed and recommend this paper for acceptance. I cannot raise my score directly (not sure why) but to the AC: please consider my score raised to accept.

---

> > ### Author Response · Authors · 2025-11-30
> >
> > > Random sampling \& maximum delta
> >
> > Suppose that each subproblem is solved exactly and the approximation error is zero. Let $i_t$ be the index sampled at each iteration. Then we have: $\|| \nabla f(x^{t+1}) \||^2 \lesssim \frac{ ( \lambda +\Delta_{i_t} )^2}{\lambda - \Delta_{i_t}}\bigl(
> > f(x^t) - f(x^{t+1}) \bigr) $. Since both $\Delta_{i_t}$ and $x^{t+1}$ depends on $i_t$, the simplest way is to first upper bound $\Delta_{i_t} \le \Delta_{\max}$ and then taking the expectation w.r.t. $i_t$ on both sides.
> >
> > Otherwise, it is an interesting idea to explore in the future how sampling might help improve the constant dependence (e.g., using importance sampling). But at the moment, we do not see how uniform sampling would help.
> >
> > We sincerely appreciate the support, and thank the consistent and strong help in improving the paper from the reviewer.

---

### Author Response · Authors · 2025-11-26
**Global response**

We thank all reviewers for their constructive evaluations of our manuscript and we appreciate all the help from the reviewers for improving the paper.

In this work, we aim to develop optimization algorithms that are efficient in both
communication and local computation in the setting where client-selection strategies might incur different
costs. Specifically,

- We propose a new model that associates the non-uniform costs with different client selection strategies. We provide rigorous yet simple definitions for distributed algorithms and two information-based notions of complexity metrics. This allows fair comparisons across algorithms and can also help establish formal lower bounds, even if all the costs are the same.

- We propose a new algorithm that achieves the best-known communication and local complexities among existing methods for non-convex optimization, under the weakest assumptions. This method is based on the inexact composite gradient method (I-CGM) with gradient estimators constructed using recursive gradient and SAGA.

During the rebuttal, we made the following changes.

- We largely improve the proof structure and clearly illustrate how each component of the method contributes to the final complexity. We remove the vague Big-O notation in all Lemmas and Theorems (Reviewer MDyq, 8ScQ, cpvm).
- We simplify and improve the definitions of communication and local complexities. (Section C)
- We provide the explicit complexity of I-CGM-RG-SAGA with one or zero full client synchronizations (inexact initialization). The latter allows removing the dependence on $C_A$ completely without requiring additional assumptions (Section G) (Reviewer cpvm, 8ScQ)
- We add the ablation study of I-CGM-RG-SAGA and illustrate how each parameter impacts its performance empirically and how it connects to the theory. (Section J.1.1) (Reviewer 8ScQ, cpvm) We also compare the performances under different ratios of $n/m$. (Reviewer MDyq, 8ScQ, RxwZ,cpvm)
- We add the amount of local computation required to achieve the desired communication complexity when using the local stochastic CGM method. (Section H) (Reviewer 8ScQ)
- We improve the introduction and abstract. After submission, we believe a more appropriate title is "Non-Convex Federated Optimization under Cost-Aware Client Selection".

We anticipate an interactive discussion with you, and we will be most happy to answer any remaining questions.

---

### Meta-Review · Area_Chair_ftdr · 2026-01-05

**Summary:**

Major concerns raised by reviewers lie in experiments (for example, strange baselines, missing ablation studies, missing practical insights), and presentation. After reading the author rebuttal, I find most concerns have been addressed and one reviewer has accordingly raised their score. The next version of this paper can benefit by incorporating all changes made in author rebuttal.

Importantly, all reviewers highly appreciate the technical novelty, quality of theoretical analysis, and significance of this paper, so the final reviews could be a unanimous decision toward accept if all reviewers could have actively engaged in the discussion with authors. Specifically, this work presents a new variance bound of the SAGA estimator that depends on the similarity constant $\delta$ rather than the individual smoothness $L_\max$ which can be significantly larger and has been widely used in existing literature. Given this distinctive and impactful technical contribution, I recommend Accept (oral).

**Reviewer Concerns:**

Major concerns raised by reviewers lie in experiments (for example, strange baselines, missing ablation studies, and missing practical insights), and presentation. After reading the author rebuttal, I find most concerns have been addressed.

**Reviewer Scores:**

Reviewer MDyq raised the score from 6 to 8 after reading the author rebuttal. Reviewer cpvm kept the score at 6 after reading the author rebuttal. After reading the author rebuttal, I find it can address most of concerns raised by Reviewer 8ScQ and Reviewer RxwZ, so they may raise their scores.

---

### Decision · Program_Chairs · 2026-01-26

Accept (Oral)